# Formation of False Context Fear Memory Is Regulated by Hypothalamic Corticotropin-Releasing Factor in Mice

**DOI:** 10.3390/ijms23116286

**Published:** 2022-06-03

**Authors:** Emi Kasama, Miho Moriya, Ryuma Kamimura, Tohru Matsuki, Kenjiro Seki

**Affiliations:** 1Department of Pharmacology, School of Pharmaceutical Science, Ohu University, 31-1 Misumido, Tomitamachi, Koriyama 963-8611, Fukushima, Japan; 716014@ohu-u.jp (E.K.); 715076@ohu-u.jp (M.M.); 713037@ohu-u.jp (R.K.); 2Department of Cellular Pathology, Institute for Developmental Research, Aichi Developmental Disability Center, Kasugai 480-0392, Aichi, Japan

**Keywords:** traumatic stress, false context fear memory, hypothalamic corticotropin-releasing factor, adeno-associated virus, shRNA

## Abstract

Traumatic events frequently produce false fear memories. We investigated the effect of hypothalamic corticotropin-releasing factor (CRF) knockdown (Hy-*Crf*-KD) or overexpression (Hy-CRF-OE) on contextual fear memory, as fear stress-released CRF and hypothalamic–pituitary–adrenal axis activation affects the memory system. Mice were placed in a chamber with an electric footshock as a conditioning stimulus (CS) in Context A, then exposed to a novel chamber without CS, as Context B, at 3 h (B-3h) or 24 h (B-24h). The freezing response in B-3h was intensified in the experimental mice, compared to control mice not exposed to CS, indicating that a false fear memory was formed at 3 h. The within-group freezing level at B-24h was higher than that at B-3h, indicating that false context fear memory was enhanced at B-24h. The difference in freezing levels between B-3h and B-24h in Hy-*Crf*-KD mice was larger than that of controls. In Hy-CRF-OE mice, the freezing level at B-3h was higher than that of control and Hy-*Crf*-KD mice, while the freezing level in B-24h was similar to that in B-3h. Locomotor activity before CS and freezing level during CS were similar among the groups. Therefore, we hypothesized that Hy-*Crf*-KD potentiates the induction of false context fear memory, while Hy-CRF-OE enhances the onset of false fear memory formation.

## 1. Introduction

Excessive fear stress influences cognitive function, the long-term storage of acquired information, and memory retrieval, contributing to the development of stress-related disorders [1]. The emergence of pathological fear memories or maladaptive hypermnesia is often observed in post-traumatic stress disorder (PTSD) [2]. The rapid failure of experienced fear memory, called “false memory”, can be formed within 3 h after the fear exposure [3]. A study in mice has demonstrated that memory reconsolidation after retrieval required fear-related protein synthesis within 6 h [4]. Conversely, it has been suggested that memory failure due to confusion regarding the source of a traumatic event can lead to fear generalization [2,5,6]. Indeed, this increase in generalization is due to a loss of detailed information about the context, and not fear incubation [7]. However, in contrast to false context fear memory, inducing of “fear generalization” across environments requires at least a week after exposure to a traumatic event in rodents, and it has been demonstrated that one month is required to induce freezing in novel environments [8]. According to the hypothesis that a shared neural ensemble linking distinct memories that encode close in time due to a temporary increase in neuronal excitability, a subsequent memory to the neuronal ensemble encodes the first memory. It has been hypothesized that a shared neural ensemble links distinct memories which are encoded within a close timeframe, due to a temporary increase in neuronal excitability. Accordingly, in rodents, it has been demonstrated that the first memory can strengthen a second memory within a day, but not across a week [9]. Although, a false context fear memory of the traumatic event may form and subsequently lead to fear generalization, the neuronal mechanism of fear generalization may differ from the concept of first memory neuronal ensemble encoding, which may lead to the false fear memories. 

Cortisol modulates various learning and memory processes, depending on the particular timing of cortisol increases relative to encoding, consolidation and retrieval [10]. Long-term dysregulation of cortisol systems after activation of the HPA axis has been suggested to have lasting effects on the vulnerable areas of the hippocampus, amygdala, and medial prefrontal cortex [11]. The acquisition of fear associative memory requires various HPA axis-related brain processes involving coordinated neural activity within the amygdala [10,12,13], prefrontal cortex (PFC) [14,15], and hippocampus [14,15,16,17]. The amygdala plays a key role in the acquisition of fear learning, while the PFC and hippocampus are two other crucial neural structures that contribute to this process, together representing the neural network of fear conditioning [18]. The ventral part of the medial PFC may also play a major role in fear conditioning [19,20,21]. Traumatic events within a few hours increase corticotropin-releasing factor (CRF) levels and activation of the HPA axis, which may affect memory consolidation. However, how the stress-induced activation of the HPA axis contributes to the formation of false context fear memory remains unclear. Cortisol in humans and corticosterone in rodents are both secreted due to excessive activation of the HPA axis, which is driven by CRF secreted from the hypothalamus. Shortly after a traumatic experience, secretion of either cortisol or corticosterone is typically enhanced, which suppresses HPA axis activation to produce CRF in the hypothalamus [22]. Moreover, HPA axis activity links the various symptom regarding autonomic function, anxiety, locomotor activity, or cognitive functions [23]. Therefore, the CRF secretion and HPA axis activation might contribute to the formation of false context fear memory and, so it is important to investigate how CRF contributes to the formation of the false context fear memories. 

For this study, we endeavored to understand the mechanism of formation of false fear memories after traumatic stress events. In the present study, we focus on the cause of spatial memory failure due to source confusion in inducing false fear memory, using two different contexts within 3 hours after conditional stimulation (CS). Then, we investigated the role of hypothalamic CRF on the induction of false context fear memory using mice with hypothalamic *Crf* knockdown (Hy-*Crf*-KD) and hypothalamic CRF overexpression (Hy-CRF-OE). Adeno-associated virus (AAV)-mediated mouse *Crf* small hairpin RNA (shRNA) was used to knockdown *Crf* expression, while an AAV carrying mouse *Crf* cDNA was constructed for the overexpression of CRF.

## 2. Results

### 2.1. Exposure of Novel Context in 3 h after Fear Conditioning Formed False Fear Memory That Was Further Enhanced at 24 h after Conditioning

We randomly assigned the mice to either the control group without the CS (electric footshock: 1.0 mA for 2 s) during Context A in Box A (CS (−)) or the treatment group with CS during Context A in Box A (CS (+)); see Figure 1C. In addition, mice were divided into two different contextual configurations for conditioning, resulting in two different sub-groups. One of these sub-groups was exposed to Context A, followed by Context B at 3 h after the CS and re-exposed to Context B at 24 h (ABB mice), in order to investigate whether experiencing Context B at 3 h consequently leads to the spatial memory failure between Box A and Box B at 24 h after the CS, resulting in the mice exhibiting a fear response in Box B. The other sub-group was exposed to Context A, followed by Context B at 24 h (A-B mice) without the exposure to Box B at 3 h after the CS in order to compare the freezing level in Box B at 24 h between ABB and A-B mice, and to determine whether experiencing the Box B at 3 h affects the freezing response in Box B at 24 h (Figure 1C). 

A similar freezing level during Context A in Box A was observed in A-B CS (+) and ABB CS (+) mice (Figure 2A). In contrast, freezing behavior was not observed in A-B CS (−) and ABB CS (−) mice (Figure 2A). Two-way repeated measures analysis of variance (ANOVA) revealed that the electric shocks and the time affected the freezing level (*F*(3, 37) = 15.01, *p* < 0.01, Figure 2B, Table 1). We observed that the false fear memory was formed in Box B at 3 h, as the freezing level in ABB CS (+) mice 3 h after the CS was significantly higher than that in Box B 3 h in ABB CS (−) mice that were not conditioned by electrical shocks (*F*(5, 52) = 7.09, *p* < 0.01; Tukey’s multiple comparison test: average of freezing level in ABB CS (+) 3 h after CS, *p* < 0.05, vs. average of freezing level in ABB CS (−) 3 h after CS, Figure 2B, Table 1). The freezing level in novel Box B (Box B) 24 h after the CS was higher in A-B CS (+) mice than that of A-B CS (−) mice (Tukey’s multiple comparison test: A-B CS (+) 24 h, *p* < 0.01 vs. A-B CS (−) 24 h, Figure 2B, Table 1), indicating that the false fear memory was induced at 24 h after the CS. The freezing level in Box B at 3 h in ABB CS (+) mice was similar to that of A-B CS (+) mice in Box B at 24 h (Tukey’s multiple comparison test: A-B CS (+) 24 h, *p* = 0.01 vs. ABB CS (+) 24 h, Figure 2B, Table 1), suggesting that the false fear memory of Context A was formed within 3 h after the CS. In addition, experiencing Box B 3 h after the CS potentiated the freezing level in Box B at 24 h compared to that at 3 h (Tukey’s multiple comparison test: freezing level in ABB CS (+) mice 24 h: *p* < 0.05, vs. ABB CS (+) mice 3 h; ABB CS (+) mice 3 h, *p* = 0.833 vs. A-B CS (+) 3 h; Figure 2B, Table 1). These results suggest that the false fear memory of Context A was induced within 3 h after the CS, and the false fear memory was potentiated in Box B at 24 h if the mice had experienced a similar environment to Context A with the CS after formation of the false fear memory. 

We confirmed that the mice exhibited a heightened freezing behavior in Box A 24 h after the CS in Box A (Welch’s *t*-test, *t* = 6.735, *df* = 7, *p* < 0.01, Appendix A). Two-way (CS × time) repeated measures ANOVA revealed that the contextual fear conditioning significantly affected the freezing level (*F*(1, 17) = 5.626, *p* < 0.05, Appendix A). The paired *t*-test indicated that the freezing level of ABA CS (−) mice in Box A at 24 h was similar to that in Box B at 3 h (*t* = 2.613, *df* = 4, *p* = 0.059 vs. ABA CS (−) mice 24 h after CS; Appendix A). In ABA CS (+) mice, the freezing level during Context A (24 h) in Box A after the CS was significantly larger than that in Box B at 3 h, designated as Context B (3 h) (paired *t*-test, *t* = 3.663, *df* = 13, *p* < 0.01; Appendix A). The freezing level in Box B at 3 h for ABA CS (+) mice after the CS was significantly higher than that of ABA CS (−) mice in Box B 3 h after the CS (Tukey’s multiple comparison test: ABA CS (+) 3 h: *p* < 0.01 vs. ABA CS (−) 3 h; Appendix A), indicating that the false context fear memory was also formed in ABA CS (+) mice within 3 h after the CS.

### 2.2. Hy-Crf-KD Enhanced False Fear Memory Level in 24 h and Hy-CRF-OE Potentiated the False Fear Memory Level within 3 h after Fear Conditioning

To investigate the effects of HPA axis activity on false fear memory, we induced mice with Hy-Crf-KD or Hy-CRF-OE through AAV-PHP.eB-produced virus injection into the hypothalamus, for comparison with mice injected with AAV-PHP.eB GFP (green fluorescence protein) into the hypothalamus as a control (Figure 3 and Appendix A). 

It has been reported that hypothalamus-specific *Crf* knockout mice showed a normal glucocorticoid diurnal rhythm [24]. Therefore, we measured the plasma corticosterone concentration at 12:00–14:00 h, a time range corresponding to the half-maximal concentration of daily plasma corticosterone in mice [25]. As we expected, the plasma corticosterone concentration in Hy-*Crf*-KD mice was significantly lower than that of control mice (one-way ANOVA, *F*(2, 12) = 31.68, *p* < 0.01, Fisher’s LSD, Hy-*Crf*-KD: *p* < 0.05 vs. control; Figure 4A). On the contrary, the plasma corticosterone concentration in Hy-CRF-OE mice was significantly higher than in control mice (Fisher’s LSD, Hy-CRF-OE: *p* < 0.01 vs. control; Hy-*Crf*-KD: *p* < 0.01 vs. Hy-CRF-OE, Figure 4A). The freezing level was similar in the three groups of ABB-control, ABB-KD, and ABB-OE mice during the three electric shocks in Box A as a conditioning stimulus (Figure 4B). One-way ANOVA revealed that the AAV injections affected the locomotor activity in Box A for 3 min just before the CS was delivered (*F*(2, 35) = 4.787, *p* < 0.05, Figure 4C). Fisher’s LSD test indicated that the locomotor activity in Hy-*Crf*-KD mice was higher than that of control mice and Hy-CRF-OE mice (control: *p* < 0.01 vs. Hy-*Crf*-KD; Hy-CRF-OE: *p* < 0.05 vs. Hy-*Crf*-KD, *p* = 0.277 vs. control, Figure 4C). As the mice were habituated to Box A for 15 min at 60 min prior to the test, due to the distance travelled in Box A, there is the possibility that the locomotor activity—but not anxiety level—was higher in Hy-*Crf*-KD mice than in control and Hy-CRF-OE mice. Tukey’s multiple comparison test indicated that the freezing level in Box B 3 h after the CS in either control, Hy-*Crf*-KD, or Hy-CRF-OE mice was higher than that of mice not exposed to the CS during Context A (control CS (+) 3 h: *p* < 0.05 vs. control CS (−) 3 h; Hy-*Crf*-KD CS (+) 3 h: *p* < 0.05 vs. Hy-*Crf*-KD CS (−) 3 h; Hy-CRF-OE CS (+) 3 h: *p* < 0.01 vs. Hy-CRF-OE CS (−) 3 h; Figure 4D, Table 2), indicating that the false fear memory was formed in control, Hy-*Crf*-KD, and Hy-CRF-OE mice in Box B at 3 h. On the other hand, a paired *t*-test also indicated that the freezing levels of control and Hy-*Crf*-KD mice, but not Hy-CRF-OE mice, at 24 h after the CS were higher than those at 3 h (control, 24 h: *t* = 3.022, *df* = 12, *p* < 0.05 vs. 3 h; Hy-*Crf*-KD mice, 24 h: *t* = 4.467, *df* = 10, *p* < 0.01 vs. 3 h; Figure 4D, Table 2), indicating that the false fear memory was potentiated in Box B at 24 h in control and Hy-*Crf*-KD mouse groups when the mice experienced Box B 3 h after the CS. In contrast, the freezing level in Box B at 24 h of Hy-CRF-OE mice was similar to their own freezing level at 3 h (Hy-CRF-OE, 24 h: *t* = 0.828, *df* = 14, *p* = 0.422 vs. 3 h; Figure 4D, Table 2). Interestingly, the freezing level in Hy-CRF-OE CS (+) at 3 h was significantly higher than those of control CS (+) at 3 h and Hy-*Crf*-KD CS (+) at 3 h (Hy-CRF-OE CS (+) 3 h: *p* < 0.05 vs. control CS (+) 3 h; *p* < 0.01 vs. Hy-*Crf*-KD CS (+) 3 h; Figure 4D, Table 2), suggesting that the onset of the false context fear memory was facilitated by Hy-CRF-OE. To evaluate whether Hy-*Crf*-KD affected the contextual fear generalization, we compared the difference between the freezing levels of Hy-*Crf*-KD mice at 3 and 24 h in Box B with those of control mice. Two-way ANOVA (CRF × CS) indicated that the interaction between CRF and CS did not affect the difference between 3 and 24 h in Box B (*F*(2, 51) = 2.4094, *p* = 0.100, Figure 4E); however, the CS factor significantly affected the difference between 3 and 24 h in Box B (*F*(1, 51) = 6.074, *p* < 0.05, Figure 4E). The enhancement of freezing level in Hy-*Crf*-KD mice was significantly larger than in control mice and Hy-CRF-OE mice (Tukey’s multiple comparison test: difference between 3 and 24 h in Hy-*Crf*-KD: *p* < 0.05 vs. control; *p* < 0.01 vs. Hy-CRF-OE; Figure 4E). In addition, the slope of the regression line for the correlation between the freezing level of the Hy-*Crf*-KD mice at 24 h against that at 3 h was approximately twice that of control mice, although both groups shared a similar *y*-intercept (control: Pearson’s correlation, *t* = 3.0346, *df* = 11, *p* < 0.01, *y* = 0.92*x* + 18.4, R^2^ = 0.456; Hy-*Crf*-KD: Pearson’s correlation, *t* = 2.632, *df* = 9, *p* < 0.05, *y* = 1.97*x* + 19.2, R^2^ = 0.435; Figure 4F). However, the correlation between the freezing level at 3 and 24 h disappeared in Hy-CRF-OE mice (Hy-CRF-OE: Pearson’s correlation, *t* = 1.5106, *df* = 13, *p* = 0.0774, *y* = 0.37*x* + 31.6, R^2^ = 0.149; Figure 4F). 

## 3. Discussion

In the present study, after fear conditioning in Context A, we investigated whether exposure to Context B at 3 h—a novel chamber with the absence of footshock—affected fear memory in Context B at 24 h. At the beginning of this study, we expected that the mice would be confused at 24 h when subjected to footshock, as Contexts A and B were similar in their shape, size, and floor material, both having a stainless-steel floor; although the mice could stand on a polypropylene board in Context B, while they could only stand on stainless steel in Context A, as shown in Figure 1A,B). However, contrary to our expectations, a freezing response was observed in Context B 3 h after the conditioning training during Context A, where the duration of this response in Context B was longer than that of mice that had not been subjected to conditioning in Context A. It has been demonstrated that the mice were habituated to exposure to a novel and highly dissimilar context with a conditioning chamber, exhibiting a low level of freezing in a dissimilar chamber on day 0 [26]. In our model of induction of false memory, we did not use the cue for conditioning. Therefore, false fear memory might not be able to form 3 h after the CS in the present study. It may be considered that a false context fear memory was produced within 3 h after the fear conditioning. However, the freezing level of mice in Context B 24 h after conditioning was higher than that in Context B at 3 h, suggesting that the mice may not be able to discriminate the chamber which delivered the footshock due to false memory enhancement, based on the suggestion that a traumatic fear event may precede the rapid failure of memory, leading to source confusion of a traumatic event [2,5,6], and the rapid failure of the experienced fear memory may be produced by exposure to different environments [3]. Therefore, we discussed the relationship between false context fear memory formation and hypothalamic CRF.

It has been suggested that memories are never completely precise, with novel situations being partially generalized against similar previous experiences [27]. The concept of spatial memory failure of a traumatic event in patients with PTSD is similar to the production of false context fear memories [28]. Fear generalization to innocuous stimuli acting as reminders of the trauma, even in a safe place or environment, is one of the central problema and a hallmark of PTSD [29]. It has been further posited that the generalization will be enhanced if the new context is similar, but not completely different, from the training context, due to memory source confusion [30]. False memories can be spontaneously produced after an extreme fear experience but, classically, the term “false memory” applies to memory formed without actual experience of the event. If “fear generalization” is defined as the actual memory of a fear experience that transfer a conditioned response to stimuli that perceptually differ from the original conditioned stimulus [30], fear generalization may also be the same as false fear memory. In rodents, fear generalization has been shown to be induced when mice were exposed to contexts soon after training [26]. From these reports, the definition of “memory generalization” is never only with respect to a remote memory. In the present study, we applied environments 3 h and 24 h after the CS, which the mice may not have been able to precisely discriminate from the box in which the conditioning was carried out. As such, this paradigm is not a test of the remote memories, but, instead, experiencing the novel box 3 h after the CS led to the source confusion of the traumatic event and potentiated the induction of false context fear memory. In the present study, exposing the mice to the novel box (i.e., Context B in Box B) 3 h after the CS (i.e., Context A in Box A), caused the freezing level of ABA mice in Box A at 24 h after the CS to decrease, compared with that of mice that were not exposed to Box B at 3 h (i.e., A-A mice). However, we did not perform these experiments at the same time (i.e., not in the same batch), so we cannot statistically compare the difference in freezing level between ABA and A-A mice. If the freezing level of ABA mice in Box A at 24 h was decreased, compared to that of the A-A mice, two hypotheses can be posed. One is memory extinction, and the other is that the experience in the novel Box B at 3 h interfered with the memory consolidation regarding Box A, which delivered the electrical shocks. The widely used paradigm of memory extinction is generally that, after the CS, the mice are exposed to the same chamber several times without the CS; however, we used a novel box after the CS. On the other hand, the memory of Box A, with the CS, might have become less precise by interfering with the perceptual memory though exposure to Box B at 3 h. This result can also be considered in terms of the hypothesis that a shared neural ensemble may link distinct memories that are encoded within a close timeframe due to a temporary increase in neuronal excitability. 

According to the DSM-V, patients with PTSD generally show clinical symptoms at a certain time after a traumatic event [31]. Contextual fear memory is suggested to be time-dependently generalized by the diminishing precision of remote memory recall [8]. For example, it has been suggested that the context of a fear memory becomes less specific with time, with an increase in fear generalization due to a loss of detailed information about the context, including generalization across environments [8]. Re-experiencing the remote fear memory is a remarkable characteristic of PTSD. It has been found that, when separated by a week, independent populations of neurons encoded two distinct contexts; meanwhile, while the two contexts were separated only within a day, shared neuronal ensembles between the two contexts overlapped in the CA1 region of the hippocampus [9]. Neuronal excitability can lead to increases in memory strength, and neural ensemble sharing can strengthen the memory for a secondary context within 5 h [9]. Therefore, the neuronal mechanism of generalized remote fear memories in patients with PTSD is different from false memories produced within a day. In the present study, freezing behavior was observed in Context B 3 h after the conditioning, and the freezing response at 24 h was facilitated. Therefore, our observation that experiencing a novel environment enhanced false context fear memory 24 h after a traumatic fear event may be explained by the hypothesis of shared neuronal ensembles between Box A and Box B strengthening the false memory within 24 h. 

Many papers have suggested that false fear memory is mainly dependent on the hippocampus [2,32,33,34], and it is widely known that hippocampal neurons are responsible for spatial memory [35]. False context fear memories have been postulated to be attributed to the imprecision of spatial memory [8,36]. A study has demonstrated and hypothesized that the intracerebroventricular administration of corticosterone significantly suppressed long-term synaptic potentiation (LTP) in the CA1 region of the hippocampus within 30 min in vivo [37]. LTP in the CA1 region of the hippocampus was facilitated by mineralocorticoid receptor (MR) activation [38,39]. Furthermore, reduced hippocampal MR expression has been associated with impaired synaptic plasticity and spatial memory deficit in mice [40]. In Hy-*Crf*-KD mice, the difference between the freezing level at 3 and 24 h after fear conditioning was significantly longer than that of control and Hy-CRF-OE mice. In addition, the freezing level at 3 h was similar between Hy-*Crf*-KD mice and control mice. Thus, it is conceivable that MR activation may be diminished in Hy-*Crf*-KD mice, indicating that the spatial memory of Context A and Context B was impaired in Hy-*Crf*-KD mice, who exhibited the freezing response in Context B at 24 h. According to this result, if the MR activation in Hy-*Crf*-KD mice is not enough to induce LTP in the hippocampus, Hy-*Crf*-KD mice cannot discriminate between Contexts A and B and, thus, exhibit a higher freezing level at 24 h after the conditioning. By contrast, glucocorticoid receptor (GR) activation has been reported to impair LTP induction in the CA1 region [38,41]. Moreover, cortisol-induced GR activation blocks perceptual learning in humans [42]. In Hy-CRF-OE mice, the GR may be continuously occupied by corticosterone throughout tests. Therefore, it can be supposed that GR was strongly activated by high concentrations of corticosterone in Hy-CRF-OE mice. In that case, Hy-CRF-OE mice might not be able to discriminate between Box A and Box B, even at only 3 h after the fear conditioning, due to suppressed memory consolidation. Although lower levels of cortisol in saliva [43], urine [44,45], and hair [46] in patients with PTSD, compared to those without PTSD, have been associated with more persistent traumatic memories and increased PTSD symptoms, the high-risk genotype of the GR gene against PTSD has been associated with increased GR signaling under stress [47], indicating that the GR is activated in patients with PTSD, even those with a low cortisol level. Although it has been shown that either cortisol or corticosterone secretion is typically enhanced after the traumatic event, and is followed by suppression of HPA axis activity [22], most individuals with PTSD show low cortisol excretion [43,45,46]. In the present study, plasma corticosterone in Hy-*Crf*-KD mice was lower than that of controls, and experiencing Context B at 3 h in Hy-*Crf*-KD mice led to the potentiation of false context fear memory 24 h after the CS. Therefore, both the lower corticosterone levels and the potentiation of context fear memory in Hy-*Crf*-KD mice were similar to observations in patients with PTSD. 

Functional alterations of the neural network underlying fear conditioning might contribute to the etiology of fear-related psychiatric disease, including PTSD [18]. Fear-associative memory acquisition of fear learning requires coordinated neural activity within the amygdala, prefrontal cortex (PFC), and hippocampus [17,18,19,20,21,48,49]. In addition, cortisol exerts a critical impact on the amygdala–hippocampus–ventromedial PFC network, which underpins fear and memory extinction [17]. Dysregulation of negative feedback, through cortisol suppressing the release of CRF after the long-term activation of the HPA axis, affects the hippocampus, amygdala, and medial PFC [11]. Inactivation of prefrontal inputs into the nucleus reuniens or direct silencing of nucleus reuniens projections enhances fear memory generalization [27]. Xu and Südhof have demonstrated the generalization of memory attributes for a particular context by processing information from the medial PFC en route to the hippocampus within a day after conditional training [27]. Decreased connectivity between the amygdala and medial PFC has been shown to be related to memory intrusion and the re-experiencing of traumatic events [50]. Interestingly, repetitive transcranial magnetic stimulation of the right dorsolateral PFC appears to have a positive effect in reducing core symptoms in patients with PTSD [50]. Therefore, it is of crucial importance to investigate the role of CRF and cortisol in various neuronal networks, including the amygdala–hippocampus–ventromedial PFC, with respect to their contribution to both the acquisition of fear memories and the consolidation of imprecise fear memories, although there are still limitations to the elucidation of neuronal mechanisms when using rodent models, as humans and animals differ in terms of the functional neuroarchitecture of the PFC. 

## 4. Materials and Methods

### 4.1. Animal Ethics Approval

All animal experiments were approved by the Institutional Animal Care and Use Committee of Ohu University, in compliance with the criteria mandated by the Japanese Law for the Humane (No. 2018-29, 2019-39, and 2020-17). Animals were supplied by Charles River Laboratories (Yokohama, Japan) and CLEA Japan, Inc. (Tokyo, Japan) and were housed at 25 ± 2 °C under a 12-h light (08:00–20:00 h)/12-h dark (20:00–08:00 h) cycle with ad libitum access to food and water. The three Rs principles (Replacement, Reduction, and Refinement) were implemented throughout the present study by adopting the principles of laboratory animal care to minimize distress and utilizing the minimum number of required animals for all experiments. This study was carried out in compliance with the ARRIVE guidelines. In total, 15 mice were used for measuring the plasma corticosterone concentration (5 mice for each group). For the behavioral test, in the non-AAV infected mouse test (Figure 2 and Appendix A), 54 mice (24 mice for Figure 2 and 30 mice for Appendix A) and mice were used and in the AAV infected mouse test (Figure 4), 57 mice were used for behavioral tests. In total, 126 mice were used in the present study. In the test using a group of A-B (CS−) mice, A-B (CS+) mice, ABB (CS−) mice, and ABB (CS+) mice in Figure 2 were performed at the same time. Additionally, all of the AAV injected mice with CS group (Hy-Control (CS+), Hy-Crf-KD (CS+), and Hy-CRF-OE (CS+)) were tested at the same time and all of the AAV injected mice without the CS group (Hy-Control (CS−), Hy-Crf-KD (CS−) and Hy-CRF-OE (CS−)) were tested at the same time. Every day, 3 to 4 mice were tested and these tests were repeated until the number of mice tested was sufficient for analysis.

### 4.2. Behavioral Tests

Adult male C57BL/6J mice (8–10 weeks old) were randomly assigned to the test group used for the behavioral test. All behavioral tests were performed between 10:00 and 16:00 h and were conducted and analyzed by two investigators blinded to the group assignments. All behavioral tests were recorded using a web camera installed with the ANY-maze software (Stoelting Co., Wood Dale, IL, USA). 

### 4.3. Apparatus for the Contextual Fear Conditioning Test

To measure contextual fear memory, it has been suggested that the animals are fear-conditioned and then exposed to a different context that had not been paired with a shock [51,52,53,54]. Therefore, for the present study, we modified the contextual fear conditioning methods in mice reported by Fujinaka et al. [26]. A communication box (CBX-303M, Muromachi Kikai Co., Ltd., Tokyo, Japan) was used for contextual fear conditioning in Context A but without the isolation plates. Box A (W: 330 mm, D: 330 mm, H: 454 mm) for Context A had four walls made of light-gray polyvinyl chloride and a stainless-steel grid floor (Figure 1A) with 24 stainless-steel bars (3 mm in diameter) spaced 11 mm apart to allow for the delivery of electric shocks. The stainless-steel grid floor was separated (55-mm gap) from the bottom, which was covered with a paper towel. Therefore, the mice stayed on the stainless-steel grid during the Context A test. The light was covered with white paper, in order to avoid direct illumination and kept at a constant brightness (i.e., not used as a fear conditioning cue). The center of the grid floor of the box was illuminated at 270 lux throughout the Context A test. Box B (W: 345 mm, D: 345 mm, H: 295 mm) for Context B was a similar shape to Box A, except that the four walls of Box B were made of dark-brown wood (Figure 1B). Furthermore, the bottom of Box B contained a paper towel and stainless-steel grid placed directly on a thin polypropylene board. The illuminance at the center of the floor was adjusted to 80 lux, darker than that of Box A. The stainless-steel grid had 17 stainless-steel bars (4 mm in diameter) spaced 17.8 mm apart, was not connected to the electric shock generator, and there was no space at the bottom. Therefore, mice could stand on the paper towel to avoid the stainless-steel grid.

### 4.4. Contextual Fear Conditioning Test

The mice were handled for 1 min for three consecutive days before the behavioral test. All mice were habituated to Box A 60 min before Context A for 15 min. Mice were placed in Box A again for 3 min, and then electrical shocks (1.0 mA for 2 s, delivered three times at 100 s intervals) in Box A were administered 3 min after the mice were placed on the grid floor, designated as Context A. After exposure to Context A for fear conditioning in Box A, all mice were returned to their home cage until the next context. At 3 h after CS in Box A, mice were exposed to Box B without electric shock for 150 s (context B), followed by re-exposure to Box B without electric shock for 150 s at 24 h (Context B: ABB-CS (+); Figure 1C). We measured the freezing level, the time spent in the Box, and the distance traveled throughout both the arena in the box and the central zone in the box during the first 3 min before CS in Context A and for 150 s during Context B at 3 h and 24 h. As a control group against the ABB-CS (+), we placed the mice in Box A as Context A for 430 s (3 min + 250 s) without any electric shock. Then, 3 h after Context A, the mice were exposed to Box B for 150 s, as Context B, followed by re-exposure to Box B at 24 h for 150 s as Context B (ABB CS (−); Figure 1C). To confirm the effect of fear conditioning on the freezing level in Box B at 3 and 24 h, we subjected the mice to Context A with CS (+) or without CS (−), followed by Context B 3 and 24 h after Context A. In addition, we investigated the effect of exposure to Box B at 3 h on the freezing level in Box B at 24 h by comparing the freezing level in Box B 24 h in the ABB CS (+) and ABB CS (−) groups with that in Box B 24 h after Context A with or without fear conditioning, but not followed with exposure to Context B at 3 h (A-B CS (+) and A-B CS (−); Figure 1C). In all tests in the present study, the chambers were thoroughly cleaned with a 70% ethanol solution between exposure to mice every time. Minimum freezing detection using the ANY-maze software was 1 s in both boxes. The freezing level was represented by the percentage of freezing duration relative to test duration (% freezing level). The central zone was defined as the middle area (172.5 × 172.5 mm) of the arena in Box B. The time spent in and the distance traveled in the central zone were measured by using the ANY-maze software.

### 4.5. Producing the AAV for Knockdown and Overexpression of CRF

All recombinant DNA experiments were performed in accordance with the guidelines of the Recombinant DNA Experiments Committee of Ohu University (No. 2019004) and Aichi Developmental Disability Center (No. 19-6), in compliance with the criteria mandated by the Act on the Conservation and Sustainable Use of Biological Diversity through Regulations on the Use of Living Modified Organisms in Japan. The AAV plasmid to express shRNAs or CRF-T2A-RFP (red fluorescent protein) was developed by modifying AAV-shRNA-ctrl (#85741, Addgene, Watertown, MA, USA). We substituted the RNA polymerase III-driven U6 promoter with EYFP (enhanced yellow fluorescent protein) encoding the synthesized DNA fragment encoding the U6 promoter, CAG promoter, EGFP (enhanced green fluorescent protein), or CRF-T2A-RFP (Biomatik, Cambridge, ON, Canada). Another plasmid, pCAGGS-FLAG-CRF, was used for shRNA validation. The shRNA sequences used to knockdown *Crf* were based on a previous report [55], the Sigma MISSION shRNA library by the RNAi Consortium (Boston, MA, USA): (#1) AGATTATCGGGAAATGAAA [55], TRCN0000414479 (#2) TTAGCTCAGCAAGCTCACAG, TRCN0000436997: and (#3): ATCTCTCTGGATCTCACCTTC. We assessed the knockdown efficiencies of shRNA candidates by co-transfection of each shRNA-carrying AAV plasmid and FLAG-CRF-encoding pCAGGS plasmid into HEK293FT cells. CRF expression was analyzed by immunoblot with anti-FLAG-HRP antibody (#015-22391, FUJIFILM Wako Pure Chemical Corp., Tokyo, Japan) (Appendix A). AAV was produced by following the protocols reported by Challis et al. [56] with minor modifications. HEK293FT cells were passaged and cultured in fifteen 15 cm diameter dishes in DMEM supplemented with 10% FBS and penicillin streptomycin. Cells were transfected with the plasmids pUCmini-iCAP-PHP.eB (Addgene #103005), pAdDeltaF6 (#112867, Addgene), and the plasmid-carrying rAAV genome (pAAV control). To collect AAV particles from cells and culture media, cell pellets were dissociated with 1 mg/mL of DNase I solution and subjected to six freeze–thaw cycles. The AAV-containing solution from cell pellets was collected and placed at 4 °C. The culture medium was mixed with 40% PEG8000, followed by incubation on ice for 2 h. AAV particles were precipitated by centrifugation at 6000× *g*, 4 °C for 30 min, and dissociated with DNase I solution. The AAV-containing solution from cell pellets and the culture medium were mixed and ultracentrifuged at 160,000× *g* with Opti Prep gradients. Purified AAV was concentrated and buffer-exchanged to PBS (−). The virus was titrated using THUNDERBIRD SYBR qPCR Mix (Toyobo, Osaka, Japan) and a CFX96 real-time PCR system (Bio-Rad Laboratories, Hercules, CA, USA), with the following primer sets targeting the WPRE sequence: GGCTGTTGGGCACTGACAAT and CCGAAGGGACGTAGCAGAAG.

### 4.6. AAV Injection into the Hypothalamus

Mice were anesthetized with a mixture of medetomidine hydrochloride, butorphanol tartrate (0.3 and 5.0 mg/kg, respectively; Wako Pure Chemical Corp.) and midazolam (4.0 mg/kg; Sandoz Ltd., Yamagata, Japan) to ensure the loss of sensation, including loss of pain sensation and immobilization during procedures. After anesthetization, the mice were placed in a stereotactic frame (#68045, RWD Life Science, Guangdong, China), and four holes were made in the skull using a dentist’s drill. Two holes were used for positioning the needle during injection, and the other two holes served to anchor the stabilizing screws for dental cement. Next, a Hamilton neurosyringe (32-gauge, 7000.5 Neuros Syringe, #65457-02, Hamilton Co. Japan K.K., Tokyo, Japan) was used for bilateral AAV microinjection into the bilateral hypothalamus, which was carried out by infusing 0.05 μL AAV into each side (0.1 μL/min). The titers of AAV-GFP (as a control), AAV-shRNA-*Crf*-GFP (for knockdown of *Crf*), and AAV-CRF-RFP (for overexpression of CRF) were 1.08 × 10^12^, 1.22 × 10^11^, and 7.22 × 10^11^ vg/mL, respectively. The stereotaxic coordinates (mm), according to the Paxinos mouse brain atlas [57,58], were as follows. For hypothalamic injection from the anterior side for the right hemisphere, anteroposterior (AP), −0.7 mm, lateral (L), 0.25 mm from bregma; depth and (DV), −4.6 mm (just above the paraventricular nucleus of the hypothalamus). After the AAV injection, the neurosyringe was left in place for at least 5 min to minimize the spread and leakage of the drug along the injection track [59]. Next, both holes were filled with dental cement (GC Unifast II, GC Dental Products Corp., Tokyo, Japan) attached to the stabilizing screw. After waking from anesthesia, mice were kept in their home cages for 14 days before behavioral tests.

### 4.7. Verification of AAV Infection and CRF Expression Levels

Mice were anesthetized with a mixture of medetomidine hydrochloride, butorphanol tartrate (0.3 and 5.0 mg/kg, respectively), and midazolam (4.0 mg/kg). Transcardial perfusion was performed with ice-cold 0.1 M PBS, followed by 4% ice-cold PFA in 0.1 M PBS. The brains were removed and post-fixed in OTC compound (Sakura Finetek Japan Co., Ltd., Tokyo, Japan) and frozen in a deep freezer (−80 °C) for 30 min. Next, 20 μm-thick sections were cut from the frozen brain block using a cryostat at −20 °C (Leica CM1100, Leica Microsystems, Wetzlar, Germany) and were then mounted on poly-L-lysine-coated glass slides (#S7441, Matsunami Glass Ind., Ltd., Osaka, Japan). Each section was incubated in HistoVT One (#06380-05, Nacalai Tesque, Inc., Kyoto, Japan) at 70 °C for 20 min to enhance the antigen–antibody reaction. After further incubation for 30 min in PBS containing 0.2% Tween 20 and 10% normal donkey serum (#565-73631, FUJIFILM Wako Pure Chemical Corp.) at room temperature, the sections were incubated overnight at 4 °C in double-antibody solutions containing the following combinations: (1) To detect GFP and CRF, mouse monoclonal antibody for GFP (#M048-3, MBL Life Science, Tokyo, Japan) (1:200) and rabbit polyclonal antibody against CRF (#H-019-06, Phoenix Pharmaceuticals, Burlingame, CA, USA) (1:100); and (2) to detect RFP and CRF, mouse monoclonal antibody for RFP (#M155-3, MBL Life Science) (1:200) and rabbit polyclonal antibody against CRF (#H-019-06) (1:100) as primary antibodies in PBS containing 0.2% Tween 20 and 10% donkey serum. Next, the primary antibodies were removed by washing the sections three times with PBS containing 0.2% Tween 20. Then, the sections were incubated for 2 h in appropriate combinations to detect GFP and CRF: an Alexa Fluor 488, donkey anti-mouse IgG (H&L) secondary antibody for GFP (#ab150105, Abcam, Cambridge, UK) (1:200), and an Alexa 555, donkey anti-rabbit IgG H&L (#ab150106, Abcam) (1:200) for CRF, or to detect RFP and CRF, an Alexa Fluor 555, a donkey anti-mouse IgG H&L (#ab150106, Abcam) (1:200) for RFP, and an Alexa 488, a donkey anti-rabbit IgG H&L (#ab150073, Abcam) (1:200) for CRF. Next, the sections were washed with PBS containing 0.2% Tween 20, mounted on slides, and covered with coverslips. A confocal scanning laser microscope (LSM 510, Carl Zeiss, Oberkochen, Germany) was used to detect and acquire the fluorescence imaging data. Immunofluorescence was quantified using the Image-Pro Plus imaging software (Media Cybernetics, Silver Springs, MD, USA). When the GFP or RFP signal was not observed in the hypothalamus, the data obtained from the behavioral tests were excluded.

### 4.8. Measurement of Plasma Corticosterone

The mice were anesthetized with pentobarbital sodium hydrochloride (70 mg/kg), and blood samples were obtained by cardiac puncture. These mice were not subjected to any behavioral tests and this experiment was performed by an investigator who was blinded to the group assignments. Plasma was collected in tubes containing 1.0% citrate, 1.5 mg/mL EDTA-2Na, and 12 U/mL heparin as an anticoagulant. Immediately after blood collection, the samples were centrifuged at 1200× *g*, 4 °C for 15 min to recover the supernatants, which were immediately stored at −80 °C until further use. Plasma concentration of corticosterone (rodent glucocorticoid) was measured using an ELISA kit, according to the manufacturer’s protocol (#YK240, Yanaihara Inst., Inc., Shizuoka, Japan). The detection limit of the ELISA kit was 0.21 ng/mL. The serum samples were analyzed in duplicate against standard curves of a known dilution and positive and negative controls as appropriate, and color development was measured using a microplate reader (BioTek Instruments, Inc., Winooski, VT, USA), according to the manufacturer’s guidelines. The intra-assay coefficient of variation (%CV) of the ELISA was 2.5–4.7%.

### 4.9. Statistical Analysis

Statistical analyses were performed using the EZR (Easy R) software [60] (version 1.38; Saitama Medical Center, Jichi Medical University, Saitama, Japan) for Pearson’s correlation test and the BellCurve software (version 3.20, Social Survey Research Information Co., Ltd., Tokyo, Japan) [61]. One-way ANOVA was used for Figure 2B and Figure 4A,C,E). Two-way repeated measures ANOVA was performed for Figure 2A and Figure 4D). Tukey’s multiple comparison test was used to compare groups in Figure 2B and Figure 4E. Fisher’s LSD test was performed after one-way ANOVA when the number of groups was three, such as in Figure 4A,C. Correlation analyses were performed by Pearson’s correlation test in Figure 4F. All data presented as bars indicate the mean ± standard error of the mean (SEM). All analyses were set at *p* < 0.01 (**) or *p* < 0.05 (*).

## 5. Conclusions

In the present study, we found that exposure to novel but similar contexts in the environment of traumatic fear experience within a few hours induced the formation of false fear memories, which were contextually enhanced within a day after the traumatic fear event. These results suggest that exposure to a novel context within a few hours after a traumatic fear event may interfere with the consolidation of precise fear memory regarding a traumatic event, thus enhancing false context fear memory formation within 24 h. The knockdown of hypothalamic Crf increased the freezing level and potentiated the false context fear memory at 24 h after fear conditioning. On the other hand, the overexpression of hypothalamic CRF enhanced the onset of false fear memory formation within a few hours after the traumatic fear event. Cortisol has been suggested to exert a critical impact on the amygdala–hippocampus–ventromedial PFC network, thus underpinning the acquisition of fear memories, as well as the consolidation of precise fear memories and fear generalization. Therefore, it is important to understand how the HPA axis contributes to the formation and potentiation of false context fear memories, as well as to explore the associated clinical implications for the treatment of psychiatric diseases, such as PTSD and advance knowledge related to CRF-related cognitive impairment, such as post-stroke depression [62], in the future.

## Figures and Tables

**Figure 1 ijms-23-06286-f001:**
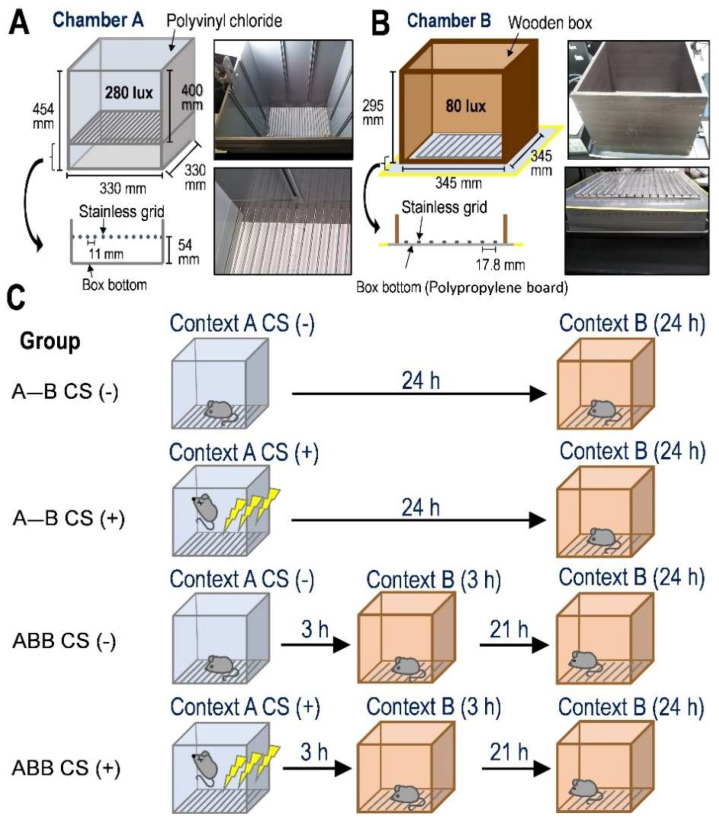
The two boxes used in the present study and the experimental paradigm for the contextual fear memory test. (**A**) Illustration of the schematic representation of Box A for contextual fear conditioning. The right two panels are photographs of Box A. Upper panel represents Box A from the diagonal above Box A, and the lower panels show the grid floor in detail. There is a 54-mm spacing from the grid to the bottom. A paper towel covered the bottom. The brightness in Box A was kept at 280 lux. (**B**) Illustration of the schematic representation of Box B for Context B. The four walls were made of brown wood, and the brightness was kept at 80 lux. The stainless-steel grid was sheeted directly on the bottom with no space between the grid and the bottom. The right two panels are photographs of Box B. Upper panel represents Box B from the diagonal above Box B, and the lower panel shows the grid floor in detail when the four walls are removed. (**C**) Experimental paradigms for contextual fear memory for A-B CS (−), A-B CS (+), ABB CS (−), and ABB CS (+) groups. Electric shocks (1.0 mA for 2 s × 3 times at 100 s intervals) were delivered to the mice for CS (+).

**Figure 2 ijms-23-06286-f002:**
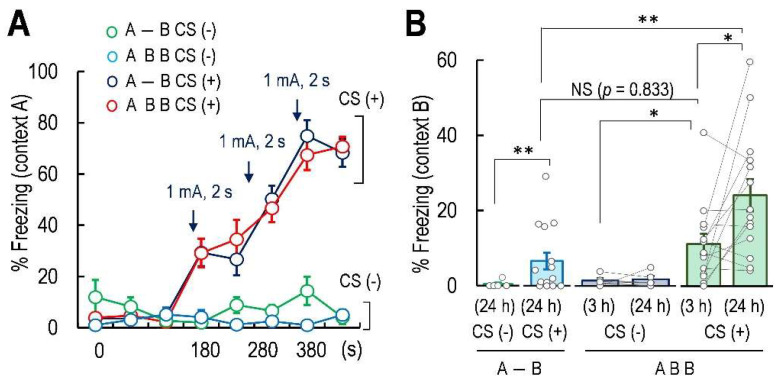
Freezing level in the contextual fear conditioning. (**A**) Percentage (%) of time spent in freezing during Context A, when the electric shocks (CS (+)) were delivered at 100 s after the mice were placed in the box. Arrows represent electric shock delivery at 180, 280, and 380 s after the mouse was put in the box. Freezing time was recorded every 1 min. (**B**) Bar graphs showing the percentage of freezing level in Box B 24 h in A-B CS (−) (n = 5), Box B at 3 and 24 h in ABB CS (−) (n = 5), Box B 24 h in A-B CS (+) (n = 15), and Box B 3 and 24 h in ABB CS (+) mice (n = 14). Data are represented by mean ± SEM. NS represents no significant difference. * *p* < 0.05; ** *p* < 0.01.

**Figure 3 ijms-23-06286-f003:**
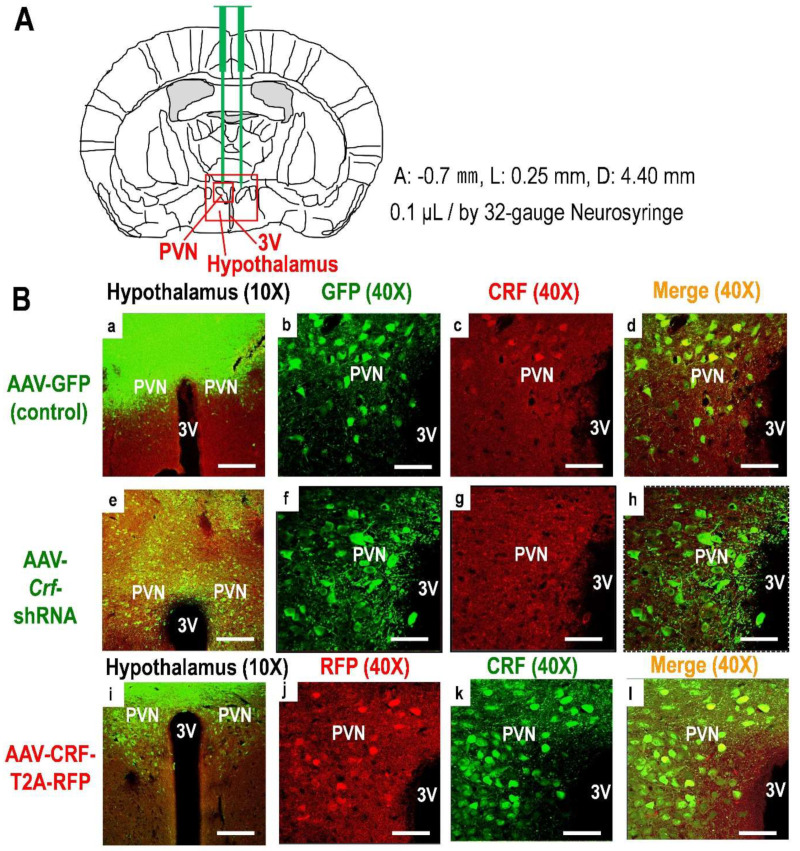
(**A**) AAV-PHP.eB virus injection maps according to the Paxinos mouse brain atlas (anterior: −0.7 mm, lateral: 0.25 mm, depth: 4.40 mm). Green bars around the center of the brain map indicate virus injection needles and the location of virus injections into the hypothalamus, including the PVN (0.1 μL of virus injected by 32-gauge neurosyringe). In the images, “3V” represents the third ventricle and “PVN” represents the paraventricular nucleus of hypothalamus in the red square, in the hypothalamus. (**B**) Transduction of mouse hypothalamus with AAV-PHP.eB vector expressing GFP or RFP and detection of mouse CRF by immunofluorescence in control ((**a**–**d**); upper lane), knockdown ((**e**–**h**); middle lane; Hy-*Crf*-KD), and overexpression ((**i**–**l**); lower lane, Hy-CRF-OE) mice. In the images, “3V” represents the third ventricle and “PVN” represents the paraventricular nucleus of the hypothalamus. Scale bar = 200 μm (×10 magnification) for (**a**,**e**,**i**) and 50 μm for (**b**–**d**,**f**–**h**,**j**–**l**) (×40 magnification).

**Figure 4 ijms-23-06286-f004:**
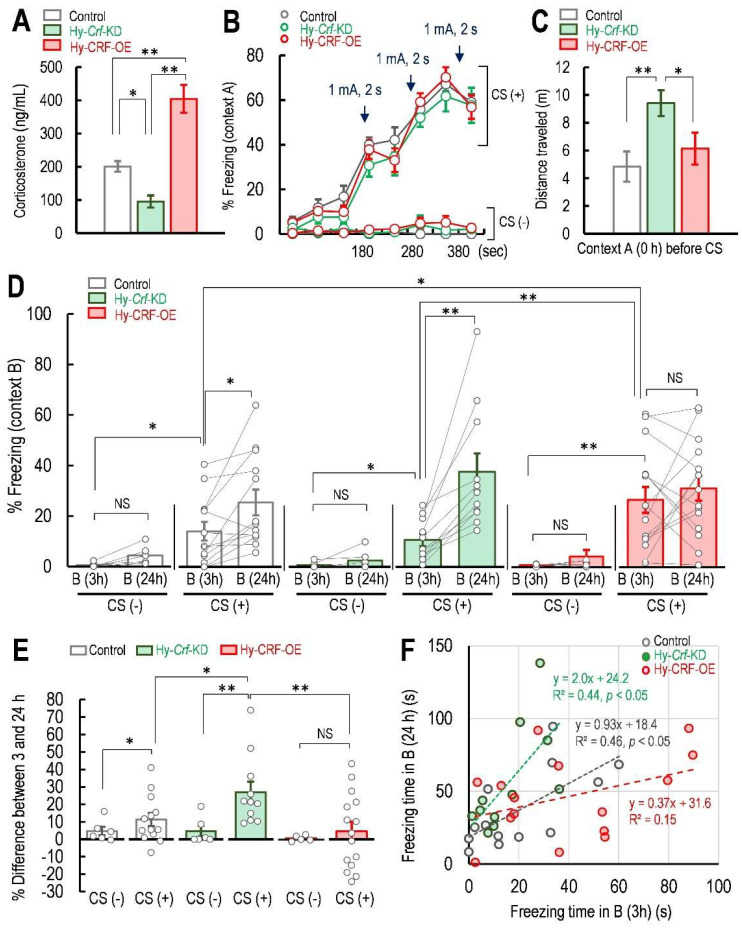
Effect of hypothalamic *Crf* knockdown or overexpression on freezing level during and after contextual fear conditioning. (**A**) Concentration of plasma corticosterone in control (n = 5), Hy-*Crf*-KD (n = 5), and Hy-CRF-OE (n = 5) mice during 12:00–14:00 h in the light phase. (**B**) Freezing level during Context A with (+) or without (−) the conditioning stimulations (CS). Arrows represent the electric shocks delivered at 180, 280, and 380 s after the mouse was placed in the center of the bottom of Box A. (**C**) Effect of hypothalamic *Crf* knockdown or CRF overexpression on locomotor activity. Distance (m) traveled in Box A before CS in ABB CS (+) mice group with Hy-*Crf*-control (n = 13), Hy-*Crf*-KD ABB (n = 11), and Hy-CRF-OE (n = 15). (**D**) Percentage (%) of freezing level in Context B at 3 and 24 h for ABB CS (+) mice with Hy-*Crf*-control (n = 13), Hy-*Crf*-KD ABB (n = 11), and Hy-CRF-OE (n = 15) and ABB CS (−) and with Hy-*Crf*-control (n = 5), Hy-*Crf*-KD ABB (n = 7), and Hy-CRF-OE (n = 5) at 3 and 24 h after CS. (**E**) Percentage (%) of freezing level differences between 3 and 24 h in ABB CS (+) with the Hy-*Crf*-control (n = 13), Hy-*Crf*-KD (n = 11), and Hy-CRF-OE mice (n = 15) and ABB CS (−) mice with the Hy-*Crf*-control (n = 6), Hy-*Crf*-KD (n = 6), and Hy-CRF-OE mice (n = 6). (**F**) Correlation between freezing level at 3 h vs. 24 h in control (n = 13), Hy-*Crf*-KD (n = 11), and Hy-CRF-OE mice (n = 15). R^2^ is the correlation coefficient. Data are represented as mean ± SEM. * *p* < 0.05; ** *p* < 0.01.

**Table 1 ijms-23-06286-t001:** Statistical data for Figure 2B.

**Two-Way RM ANOVA**		**Sum Sq**		** *Df* **	** *F* **		** *p* **	**Significance**
Group		4040.128		5	7.0933		*p* = 0.0000402	***
Time × CS		9963.637		57				
**Group**	**n**	**Mean**	**SEM**	** *t* **	** *Df* **	**Cohen’s d**	**Welch** **Test**	**Significance**
A-B (B-24h) CS−	5	0.440	0.440	2.566	18	1.147	*p* = 0.0274	*
A-B (B-24h) CS+	15	6.436	2.207					
**Group**	**n**	**Mean**	**SEM**	** *t* **	** *Df* **	**Cohen’s d**	**Paired** ***t*-Test**	**Significance**
ABB (B-3h) CS−	5	1.373	0.621	0.2730	4	0.2070	*p* = 0.798	N.S.
ABB (B-24h) CS−	5	1.680	0.843					
ABB (B-3h) CS+	14	23.805	4.373	2.7170	13	0.970	*p* = 0.0176	*
ABB (B-24h) CS+	14	11.0095	2.767					
**Group**		**n**	**Mean**		** *t* **		**Tukey’s** **Test**	**Significance**
A-B in B 24 h (CS+)		15	6.436					
vs. ABB in B 3 h (CS+)		14	11.00952		1.153		*p* = 0.833	N.S.
vs. ABB in B 24 h (CS+)		14	23.805		4.379		*p* = 0.000688	***
ABB in B 3 h (CS−)		5	1.373		2.7330		*p* = 0.0478	*
ABB in B 3 h (CS+)		14	11.00952					
ABB in B 24 h (CS−)		5	1.680		3.9789		*p* = 0.00245	**
ABB in B 24 h (CS+)		14	23.805					

* *p* < 0.05, ** *p* < 0.01, *** *p* < 0.001.

**Table 2 ijms-23-06286-t002:** Statistical data for Figure 4D.

**Two-Way RM ANOVA**	**Sum Sq**		** *Df* **	***F* Value**		***p* Value**		**Significance**
Group	12,587.407		5	7.602		*p* = 0.0000212		***
Time × CS	40,989.471		113					
**Group**	**n**	**Mean**	**SEM**	** *t* **	** *Df* **	**Cohen’s d** **Test**	**Paired** ***t*-Test**	**Significance**
Control CS (−) B-3h	6	0.567	0.394	2.260	5	1.161	*p* = 0.733	N.S.
Control CS (−) B-24h	6	3.683	1.651					
Control CS (+) B-3h	13	13.964	3.661	3.0222	12	0.737	*p* = 0.0106	*
Control CS (+) B-24h	13	25.185	5.019					
Hy-*Crf*-KD CS (−) B-3h	6	0.700	0.492	1.510	5	0.951	*p* = 0.191	N.S.
Hy-*Crf*-KD CS (−) B-24h	6	5.367	3.064					
Hy-*Crf*-KD CS (+) B-3h	11	10.509	2.397	4.467	10	1.557	*p* = 0.0012	**
Hy-*Crf*-KD CS (+) B-24h	11	37.182	7.274					
Hy-CRF-OE CS (−) B-3h	6	0.807	0.175	1.207	5	0.767	*p* = 0.282	N.S.
Hy-CRF-OE CS (−) B-24h	6	4.183	2.776					
Hy-CRF-OE CS (+) B-3h	15	26.280	5.058	0.827	14	0.245	*p* = 0.422	N.S.
Hy-CRF-OE CS (+) B-24h	15	30.818	4.848					
**Group**	**n**		**Mean**	** *t* **		**Tukey’s** **Test**		**Significance**
Control CS (−) B-3h	6		0.189	2.536		*p* = 0.0316		*
Control CS (+) B-3h	13		13.964					
Hy-*Crf*-KD CS (−) B-3h	6		0.7000	1.806		*p* = 0.0446		*
Hy-*Crf*-KD CS (+) B-3h	11		10.509					
Hy-CRF-OE CS (−) B-3h	6		0.807	4.926		*p* = 0.0000565		***
Hy-CRF-OE CS (+) B-3h	15		26.280					
Control CS (−) B-24h	6		4.350	3.944		*p* = 0.00204		**
Control CS (+) B-24h	13		25.185					
Hy-*Crf*-KD CS (−) B-24h	6		5.367	5.856		*p* = 0.00000206		***
Hy-*Crf*-KD CS (+) B-24h	11		37.182					
Hy-CRF-OE CS (−) B-24h	6		4.183	5.151		*p* = 0.0000240		***
Hy-CRF-OE CS (+) B-24h	15		30.818					
Control CS (+) B-3h	13		13.964	0.788		*p* = 0.965		NS
Hy-*Crf*-KD CS (+) B-3h	11		10.509					
Control CS (+) B-3h	13		13.964	3.0364		*p* = 0.0333		*
Hy-CRF-OE CS (+) B-3h	15		26.280					
Hy-*Crf*-KD CS (+) B-3h	11		10.509	3.712		*p* = 0.00441		**
Hy-CRF-OE CS (+) B-3h	15		26.280					
Control CS (+) B-24h	13		25.185	2.736		*p* = 0.0726		NS
Hy-*Crf*-KD CS (+) B-24h	11		37.182					
Control CS (+) B-24h	13		25.185	1.389		*p* = 0.715		NS
Hy-CRF-OE CS (+) B-24h	15		30.818					
Hy-*Crf*-KD CS (+) B-24h	11		37.182	1.498		*p* = 0.646		NS
Hy-CRF-OE CS (+) B-24h	15		30.818					

* *p* < 0.05, ** *p* < 0.01, *** *p* < 0.001.

## Data Availability

Data are available from the corresponding author upon a reasonable request and with the permission of Tohru Matsuki and Kenjiro Seki.

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
