# Peer review of "Formation of False Context Fear Memory Is Regulated by Hypothalamic Corticotropin-Releasing Factor in Mice"

_ijms, 2022, doi:10.3390/ijms23116286_

Round 1
Reviewer 1 Report
In this study, Kasama et al. study the consequences of different conditioning paradigms and CRF expression on freezing response in mice, that they explain in the context of false contextual memory formation and generalization. The idea is stimulating, the experiments are technically well executed and in the end the manuscript provides some interesting observations. This is the case for the reported increase in freezing in context B after pre-exposure to the same context 3h after conditioning. However, we have some major problems with both the interpretation of the results and the fitting with previous conceptions of, for instance, what fear generalization actually is. This, together with a rather poor English grammar, makes the paper difficult to read and be put in context with previous work.
One major concern is the concept of “generalization”. Although the authors dedicate a big part of the introduction to present these terms, the difference between “false memory” and “generalization” (and in the discussion, “memory confusion”) is not clear. Is it a matter of time? Amount of detected freezing? Are there different neurobiological mechanisms? Basically, the authors make interpretations in the discussion (page10 of 17, line 312) that previously they have assumed in their entire manuscript, and only when reading the discussion one understands what they refer to with generalization and false memory. They acknowledge the definition of generalization provided by others (Wiltgen and Silva), but disregarding that this definition is applied to memories more remote than 24h. This is quite confusing, and makes the first reading of the paper complicated for someone used to the general definition of generalization.
Importantly, the authors do not cite key work related to their experiments and addressing the neurobiological mechanisms underlying memory linking (probably the equivalent to their “false memory”) such as Cai et al., 2016. In fact, most of what is discussed in the manuscript can also be interpreted under the light of the two memories (context A and B) being linked, rather than “confused” or “difficult to discriminate” as the authors suggest.
Having said this, my biggest suggestion is that the authors come with a different name for what they refer to with “generalization”, that then can be for the first time linked to false memory formation and CRF levels. This could improve the paper.
Other comments:
In the results section, sub-titles should reflect statements related to the most important finding (“manipulation A results in B”) rather than a generic sentence.
Fig 1
The two contexts have the same shape. How do the authors think their results would look if the two contexts were more dissimilar, e.g., with a different shape?
Fig 2
In our opinion, the false memory vs. generalization dichotomy brings problems in fig 2: Here, the authors say that “false fear memory (in context B) was induced 24h after the CS” and, below, “formed within 3h after CS” (page4, line112), after comparing freezing in A-B 24h vs. A-B 3h. We understand from this is to suggest that elevated levels of freezing both at 3h and 24h after conditioning are interpreted by the authors as similar false memory. However, 3h is similar to the time required for memory linking (Cai et al., 2016), whereas 24h could be, although still too short, more widely associated with the idea of “generalization”. In the concluding sentence, they state that, on the one hand, false memory was induced within 3h and, on the other hand, that generalization was produced if the mice had experienced the intermediate exposure to context B. Looking at the data, it is clear that “generalization” is produced regardless of the intermediate exposure (as evidenced by increased freezing in A-B 24h). What remains interesting is the fact that prompt formation of false memory (3h) is necessary for having increased freezing levels at 24h compared to those without 3h exposure to context B. This is probably the most interesting observation, that in our opinion becomes diluted in the false memory vs. generalization terminology/conceptual discussion. Calling this something different, such as “false memory enhancement” would eliminate this problem.
In comparing freezing after 24h in context A in A-A vs. A-B-A mice (fig. suppl. 1C vs. D), it seems like exposure to context B 3h after conditioning weakened memory expression in the context A. First, the experiment ABA comes a little bit out of the blue and should be more properly introduced. Second, this observation might be linked to the same concept as before, in the sense that intermediate exposure to different contexts affect memory expression 24h later, which is very interesting. How do the authors interpret this?
Fig 3
Subfigures A-D can be moved to supplementary materials. The entire maps of the viral vectors are not necessary, a schematic representation is enough. In B, do the authors have in vitro evidence that their overexpression constructs worked? In D, what does “Nat Neurosci” and the different code numbers refer to? In F, the distance indicated by the scale bars should be indicated in the figure legend. Furthermore, a DAPI stain would be an added plus, as well as a zoomed out image of the hypothalamus. Finally, the figure legend contains a panel “M?
Fig S2
Where the Flag and actin WB carried out on the same membrane? The way the supp figure is presented, this is not clear.
Fig 4
4C is referenced in the text before 4B.
Variability of freezing is high among injected animals. Was an outlier test carried out? Were injection sites checked and animals with poor injection discarded from the test?
Were all the animals conditioned and tested at the same time? Otherwise, inter-group comparisons of total freezing levels might be subject to batch effects that are difficult to account for. In any case, the difference in freezing levels between 3h and 24h is interesting and adequate in trying to normalize freezing per group even if this was obtained from different batches.
Page 6, line 189. This was the first time, together with the conclusion of the first part of the results, we understood that the authors use “fear generalization” for the increase in freezing observed at 24h after the pre-exposure to context at 3h post-conditioning… This is problematic because this is only clearly stated in the discussion and reflects and interpretation of the authors, it is difficult to follow the paper if this is not defined before. Again, we would suggest that the authors come with a new name for the phenomenon they observe, to distinguish it from the more traditional idea of generalization. This might help the reader to not get distracted in terminology issues.
Fig 5
We appreciate the effort in trying to link anxiety leading to “false memory” to freezing the day after, but do not quite understand the logic behind this analysis and what additional information different from freezing this brings… it is obvious that the longer a mouse will spend freezing, the less distance it will move. Correlation of 3h distance vs. 24h freezing is lost in genetically manipulated mice, but correlation is not causation (statements as in page10 of 17, line 358 should be weakened). This only reinforces our opinion that it is extremely difficult to extract any conclusion about anxiety levels from assessments that are no de-coupled from fear conditioning. Assessing distance travelled in context A before CS, as done in fig4C, is the closest and most appropriate way.
Other comments:
- Line 30, evidence is not plural
- Line 64: The statement “in patients with PTSD” is grossly misleading, as mice are being studied here.
- Line 143-147. This is not a proper sentence, grammatically speaking.
Author Response
To the Editors Fukushima, Japan
International Journal of Molecular Sciences May 22nd, 2022
Subject: Revision for Manuscript ID: ijms-1718457
With this letter we would like to submit a revised version of the article entitled “Formation of false context fear memory and fear generalization are respectively regulated by distinct expression levels of hypothalamic corticotropin-releasing factor in mice” by Kenjiro Seki submitted for publication to the International Journal of Molecular Sciences.
At first, we would like to thank the reviewers for their valuable critical comments and excellent suggestions, which have further contributed to the present improvement of the manuscript. We essentially followed the reviewer’s comments and arguments and address all points that had been raised. The manuscript was changed accordingly. We have highlighted all the changes made in the original manuscript below in response to reviewer. According to reviewer 1 suggestions we have also changed the title of the manuscript “Formation of false context fear memory is regulated by hypothalamic corticotropin-releasing factor in mice”.
The Abstract, Introduction, Results, Discussion and Methods section were modified according to the reviewer comments/suggestions and additional changes to make reading easier and clearer. References are updated accordingly.
For reference, the reviewer’s comments are shown in quotations. Please note that the referring page and line number shown in each our response to the reviewer comment corresponds to the line number of each page in the untracked-changed version of revised manuscript. We also submitted tracking version after the untracking version of template manuscript as one file.
Best Regards,
Kenjiro Seki, Ph.D.
Responses to Reviewer 1:
Reviewer 1 comment 1: “~~ However, we have some major problems with both the interpretation of the results and the fitting with previous conceptions of, for instance, what fear generalization actually is. This, together with a rather poor English grammar, makes the paper difficult to read and be put in context with previous work.”
We apologize that our poor English language makes difficult for understanding our works in the first submitted manuscript. We acknowledge that the English in our manuscript should be improved and we asked our revised version of manuscript to the Wordvice English Editing Service and MDPA English Editing Service.
Reviewer 1 comment 2: “One major concern is the concept of “generalization”. Although the authors dedicate a big part of the introduction to present these terms, the difference between “false memory” and “generalization” (and in the discussion, “memory confusion”) is not clear. Is it a matter of time? Amount of detected freezing? Are there different neurobiological mechanisms? Basically, the authors make interpretations in the discussion (page10 of 17, line 312) that previously they have assumed in their entire manuscript, and only when reading the discussion one understands what they refer to with generalization and false memory. They acknowledge the definition of generalization provided by others (Wiltgen and Silva), but disregarding that this definition is applied to memories more remote than 24h. This is quite confusing, and makes the first reading of the paper complicated for someone used to the general definition of generalization.”
We acknowledge that our interpretation against the word “Fear generalization” of first submitted manuscript was not adequate for our results according to general definition of published many papers. Particularly, the paper by Wiltgen an Silva, and Cai et al 2016 were very helpful to understand the definition of “generalization”. We strongly agreed the Reviewer 1 suggestion and the interpretation of the word “generalization” in our results of first submitted manuscript was changed to the “False fear memory” as follow:
Title (page 1, line 2-4):
Formation of false context fear memory is regulated by hypothalamic corticotropin-releasing factor in mice
Abstract (page 12, line ):
Traumatic events frequently produce false fear memories.
Abstract (page 12, line ):
The within-group freezing level in B-24h was higher than that in B-3h, indicating that false context fear memories were enhanced in B-24h when the mice experienced B-3h.
Abstract (page 12, line ):
Therefore, we hypothesized that Hy-Crf-KD enhances the false context fear memory in a day, while Hy-CRF-OE potentiates the false fear memory within a few hours.
Keywords (page 12, line ):
Traumatic stress, false context fear memory; hypothalamic corticotropin-releasing factor, Adeno-associated virus, shRNA
Introduction (page 4, line 1-page 5, line 3):
Indeed, the increase in generalization was due to a loss of detailed information about the context and not fear incubation [7]. However, in contrast to false context fear memory, inducing “fear generalization” across environments requires at least a week after exposure to traumatic events in rodents and it has been demonstrated that one month is required to induce freezing in novel environments [8]. According to the hypothesis that a shared neural ensemble linking distinct memories that encode close in time due to a temporary increase in neuronal excitability, a subsequent memory to the neuronal ensemble encodes the first memory. In rodents, it has been demonstrated that the first memory can strengthen the second memory within a day but not across a week [9]. Although, a false context fear memory of a traumatic event would be formed and subsequently lead to the formation of fear generalization, the considerable neuronal mechanism of fear generalization might not be the same as the concept of neuronal ensemble encoding the first memory that can lead to false fear memories.
Introduction (page 4, line 1-page 5, line 3):
The present study endeavored to understand the formation mechanism of false fear memories after traumatic stress events based on the concept of close in time encoding due to a temporary increase in neuronal excitability of subsequent memories to the neuronal ensemble after encoding an initial memory. Therefore, the present study focuses on the cause of spatial memory failure by source confusion after inducing false fear memory using two different contexts within three hours after conditional stimulation (SC). Then, we investigated the role of hypothalamic CRF on the induction of false context fear memories using mice with hypothalamic Crf knockdown (Hy-Crf-KD) and hypothalamic CRF overexpression (Hy-CRF-OE), respectively. Adeno-associated virus (AAV)-mediated mouse Crf small hairpin RNA (shRNA) was used to knock down Crf expression, and an AAV carrying mouse Crf cDNA was constructed for the overexpression of CRF.
Reviewer 1 comment 3: “Importantly, the authors do not cite key work related to their experiments and addressing the neurobiological mechanisms underlying memory linking (probably the equivalent to their “false memory”) such as Cai et al., 2016. In fact, most of what is discussed in the manuscript can also be interpreted under the light of the two memories (context A and B) being linked, rather than “confused” or “difficult to discriminate” as the authors suggest.
Having said this, my biggest suggestion is that the authors come with a different name for what they refer to with “generalization”, that then can be for the first time linked to false memory formation and CRF levels. This could improve the paper.”
As we mentioned above in our reply for Reviewer 1 comment 2, we acknowledge that our interpretation against the word “Fear generalization” of first submitted manuscript was not adequate for our results according to general definition of published many papers. We carefully read the paper by Cai et al 2016 many times and we agreed that the concept of the overlap between the neuronal ensembles representing two separate contextual memories within a day and over a week. Therefore, we extensively revised our claim throughout the manuscript as follow:
Introduction (page 4, line 1-page 5, line 3):
Indeed, the increase in generalization was due to a loss of detailed information about the context and not fear incubation [7]. However, in contrast to false context fear memory, inducing “fear generalization” across environments requires at least a week after exposure to traumatic events in rodents and it has been demonstrated that one month is required to induce freezing in novel environments [8]. According to the hypothesis that a shared neural ensemble linking distinct memories that encode close in time due to a temporary increase in neuronal excitability, a subsequent memory to the neuronal ensemble encodes the first memory. In rodents, it has been demonstrated that the first memory can strengthen the second memory within a day but not across a week [9]. Although, a false context fear memory of a traumatic event would be formed and subsequently lead to the formation of fear generalization, the considerable neuronal mechanism of fear generalization might not be the same as the concept of neuronal ensemble encoding the first memory that can lead to false fear memories.
Introduction (page 4, line 1-page 5, line 3):
The present study endeavored to understand the formation mechanism of false fear memories after traumatic stress events based on the concept of close in time encoding due to a temporary increase in neuronal excitability of subsequent memories to the neuronal ensemble after encoding an initial memory. Therefore, the present study focuses on the cause of spatial memory failure by source confusion after inducing false fear memory using two different contexts within three hours after conditional stimulation (SC). Then, we investigated the role of hypothalamic CRF on the induction of false context fear memories using mice with hypothalamic Crf knockdown (Hy-Crf-KD) and hypothalamic CRF overexpression (Hy-CRF-OE), respectively. Adeno-associated virus (AAV)-mediated mouse Crf small hairpin RNA (shRNA) was used to knock down Crf expression, and an AAV carrying mouse Crf cDNA was constructed for the overexpression of CRF.
Reviewer 1 comment 4: “In the results section, sub-titles should reflect statements related to the most important finding (“manipulation A results in B”) rather than a generic sentence.”
We acknowledge that the subtitles did not including our important finding. We have changed the subtitles in the Results section as follow:
Results (page 4, line 1-page 5, line 3):
Exposure of novel context in 3 h after fear conditioning formed false fear memory that was further enhanced at 24 hours after conditioning.
Results (page 4, line 1-page 5, line 3):
Hy-Crf-KD enhanced false fear memory level in 24 h and Hy-CRF-OE potentiate the false fear memory level in 3 h after fear conditioning.
Reviewer 1 comment 5: “Fig 1. The two contexts have the same shape. How do the authors think their results would look if the two contexts were more dissimilar, e.g., with a different shape?”
We acknowledge that we should discussed the possibility whether the mice were exposed to dissimilar chamber as a novel box represents the false fear memory. Therefore, we cite the adequate reference that dissimilar box after the conditioning did not lead the freezing behaviors in 3 hours after the conditioning in Discussion section as follow:
Discussion (page 4, line 1-page 5, line 3):
It has been demonstrated that the mice were habituated to exposed to a novel and highly dissimilar context with conditioning chamber exhibited a low level of freezing in dissimilar chamber on day 0 [25]. Our model of induction of false memory, we did not use the cue for conditioning. Therefore, false fear memory might not be able to form 3 h after the CS in the present study.
Reference:
[25] Fujinaka, A.; Li, R.; Hayashi, M.; Kumar, D.; Changarathil, G.; Naito, K.; Miki, K.; Nishiyama, T.; Lazarus, M.; Sakurai, T.; et al. Effect of context exposure after fear learning on memory generalization in mice. Mol Brain 2016, 9, 2, doi:10.1186/s13041-015-0184-0.
Reviewer 1 comment 6: “Fig 2. In our opinion, the false memory vs. generalization dichotomy brings problems in fig 2: Here, the authors say that “false fear memory (in context B) was induced 24h after the CS” and, below, “formed within 3h after CS” (page4, line112), after comparing freezing in A-B 24h vs. A-B 3h. We understand from this is to suggest that elevated levels of freezing both at 3h and 24h after conditioning are interpreted by the authors as similar false memory. However, 3h is similar to the time required for memory linking (Cai et al., 2016), whereas 24h could be, although still too short, more widely associated with the idea of “generalization”. In the concluding sentence, they state that, on the one hand, false memory was induced within 3h and, on the other hand, that generalization was produced if the mice had experienced the intermediate exposure to context B. Looking at the data, it is clear that “generalization” is produced regardless of the intermediate exposure (as evidenced by increased freezing in A-B 24h). What remains interesting is the fact that prompt formation of false memory (3h) is necessary for having increased freezing levels at 24h compared to those without 3h exposure to context B. This is probably the most interesting observation, that in our opinion becomes diluted in the false memory vs. generalization terminology/conceptual discussion. Calling this something different, such as “false memory enhancement” would eliminate this problem.
As related suggestion by Reviewer 1 in the comment 2 and 3, we have performed extensive revised throughout the manuscript, Title, Abstract, Introduction, Results, Discussion and Conclusion as follow:
Introduction (page 4, line 1-page 5, line 3):
Indeed, the increase in generalization was due to a loss of detailed information about the context and not fear incubation [7]. However, in contrast to false context fear memory, inducing “fear generalization” across environments requires at least a week after exposure to traumatic events in rodents and it has been demonstrated that one month is required to induce freezing in novel environments [8]. According to the hypothesis that a shared neural ensemble linking distinct memories that encode close in time due to a temporary increase in neuronal excitability, a subsequent memory to the neuronal ensemble encodes the first memory. In rodents, it has been demonstrated that the first memory can strengthen the second memory within a day but not across a week [9]. Although, a false context fear memory of a traumatic event would be formed and subsequently lead to the formation of fear generalization, the considerable neuronal mechanism of fear generalization might not be the same as the concept of neuronal ensemble encoding the first memory that can lead to false fear memories.
Introduction (page 4, line 1-page 5, line 3):
The present study endeavored to understand the formation mechanism of false fear memories after traumatic stress events based on the concept of close in time encoding due to a temporary increase in neuronal excitability of subsequent memories to the neuronal ensemble after encoding an initial memory. Therefore, the present study focuses on the cause of spatial memory failure by source confusion after inducing false fear memory using two different contexts within three hours after conditional stimulation (SC). Then, we investigated the role of hypothalamic CRF on the induction of false context fear memories using mice with hypothalamic Crf knockdown (Hy-Crf-KD) and hypothalamic CRF overexpression (Hy-CRF-OE), respectively. Adeno-associated virus (AAV)-mediated mouse Crf small hairpin RNA (shRNA) was used to knock down Crf expression, and an AAV carrying mouse Crf cDNA was constructed for the overexpression of CRF.
Reviewer 1 comment 7: “In comparing freezing after 24h in context A in A-A vs. A-B-A mice (fig. suppl. 1C vs. D), it seems like exposure to context B 3h after conditioning weakened memory expression in the context A. First, the experiment ABA comes a little bit out of the blue and should be more properly introduced. Second, this observation might be linked to the same concept as before, in the sense that intermediate exposure to different contexts affect memory expression 24h later, which is very interesting. How do the authors interpret this?”
We acknowledge that the results of freezing level 24 h in ABA mice looked decreased, compared to that in A-A mice. However, we did not these two experiments were not performed at the same time. Therefore, we cannot represent these graphs as one graph and perform the statistical analysis. We discussed the possibility that the shorter freezing level 24 h in ABA mice than that in A-A mice in Discussion section as follow:
Discussion (page 4, line 1-page 5, line 3):
In the present study, exposing mice to the novel box during Context B in Box B 3 h after the CS during Context A in Box A, the freezing level in Box A 24 h after the CS in ABA mice seemed decreased compared to the result of freezing level in A-A mice that were not exposed to Box B at 3 h. However, we did not perform these experiments simultaneously (not the same batch), so we cannot statistically compare the difference in the freezing level between ABA and A-A mice. If the freezing level in Box A at 24 h for ABA mice was decreased compared to that of A-A mice, two hypotheses are raised. One is memory extinction and another is that experience of Box B, at 3 h interfered with the memory consolidation of Box A that delivered the electrical shocks. The widely used paradigm of memory extinction is that after the CS, the mice should be exposed to the same chamber several times without the CS. However, we used a novel box after the CS. However, the memory of Box A with the CS might become less precise after interfering with the perceptual memory of Box B at 3 h. This result is also considered by the hypothesis that the shared neural ensemble leads to a subsequent memory with the neuronal ensemble encoding of the first memory during a temporary increase in neuronal excitability in which two distinct memories are encoded the close together in time.
Reviewer 1 comment 8: “Fig 3. Subfigures A-D can be moved to supplementary materials. The entire maps of the viral vectors are not necessary, a schematic representation is enough. In B, do the authors have in vitro evidence that their overexpression constructs worked? In D, what does “Nat Neurosci” and the different code numbers refer to? In F, the distance indicated by the scale bars should be indicated in the figure legend. Furthermore, a DAPI stain would be an added plus, as well as a zoomed out image of the hypothalamus. Finally, the figure legend contains a panel “M?”
We acknowledge that“Nat Neurosci” and the different code numbers in the figure are not explained in legend of Figure 3. We changed the indication of each band of western blotting and explained in figure legend refer to the“Nat Neurosci” and the code numbers. In addition, we wrote the scale bar in the figure legend and delete “M”. As for counterstaining with DAPI for CRF and fluorescence tag (GFP or RFP), we do not have a laser for detecting the DAPI, so we tried to purchase the TO-PRO-3 which can be detected by Ex 642. Unfortunately, we have to import from the Sigma-Aldrich or Thermo Fisher Scientific in the USA, so we have to wait over three weeks to get this. Therefore, we cannot try to stain the nuclear in the hypothalamic sections by deadline during revision.
Subfigures A-D have moved to the Supplementary Figure S2 and changed the Figure 3. Also, we improved the Methods section and the legend of Figure 3 and separate it to the supplementary Figure S2 as follow:
Methods (page , line ):
The shRNA sequences to knockdown Crf were based on a previous report [55], the Sigma MISSION shRNA library by the RNAi Consortium (Boston, MA, USA); (#1) AGATTATCGGGAAATGAAA [55], TRCN0000414479 (#2) TTAGCTCAGCAAGCTCACAG, TRCN0000436997 (#3): ATCTCTCTGGATCTCACCTTC. We assessed the knockdown efficiencies of shRNA candidates by co-transfection of each shRNA carrying AAV plasmid and FLAG-CRF encoding pCAGGS plasmid into HEK293FT cells. CRF expression was analyzed by immunoblot with anti-FLAG-HRP antibody (#015-22391, FUJIFILM Wako Pure Chemical Corp., Tokyo, Japan) (Fig. S2C, S2D and Fig. S3).
Figure 3 (page , line ):
Fig. 3. (A) AAV-PHP.eB virus injection maps according to the Paxinos mouse brain atlas. Green bars around the center of the brain map indicate virus injection needles and the location of virus injections. (B) Confocal images in the transduction of mouse hypothalamus with AAV-PHP.eB vector expressing GFP, RFP and detecting the mouse CRF by immunofluorescence in control (aï€d), knockdown (eï€h), and overexpression (iï€l). Scale bar = 200 μm (×10 magnification) for a, e, I and 50 μm for b, c, d, f, g, h, j, k, l (×40 magnification).
Supplementary Figure S2 (Page , line ):
Fig. S2. Construction and characterization of adeno-associated virus (AAV) PHP.eB vectors for knockdown of mouse Crf by shRNA system and overexpression of mouse CRF. (A) Illustration of the AAV vectors expressing Crf shRNAs under the control of U6 promoter. The vectors express GFP as a reporter gene under the control of Chicken β-Actin promoter (CBA pro).(B) Illustration of the AAV vector carrying CRF*-FLAG-T2A-RFP. CRF* represents the shRNA#2 resistant version. (C) Western blotting for expressing CRF*-FLAG-T2A-RFP carrying AAV plasmid that was transfected to HEK293 cells. (D) The knockdown validation of the effective shRNA candidates. #2 shRNA was the most effective and used this shRNA in further experiments.
Reviewer 1 comment 9: “Fig 4. 4C is referenced in the text before 4B.”
We acknowledge that the describing the result of 4B should be before the 4C. Therefore, we have changed the sentences as follow:
Results (Page , line ):
Freezing level was similar in the three groups of ABB-control, ABB-KD, and ABB-OE mice during the three electric shocks in Box A as a conditioning electric shock (Fig. 4B). One-way ANOVA revealed that the AAV injections affected the locomotor activity in Box A for 3 min just before the CS was delivered (F(2, 35) = 4.787, p < 0.05, Fig. 4C). Fisher's LSD test indicated that the locomotor activity in Hy-Crf-KD mice was higher than that in control mice and Hy-CRF-OE mice (control: p < 0.01 vs. Hy-Crf-KD; Hy-CRF-OE: p < 0.05 vs. Hy-Crf-KD, p = 0.277 vs. control, Fig. 4C).
Reviewer 1 comment 10: “Fig 4. Variability of freezing is high among injected animals. Was an outlier test carried out? Were injection sites checked and animals with poor injection discarded from the test? Were all the animals conditioned and tested at the same time? Otherwise, inter-group comparisons of total freezing levels might be subject to batch effects that are difficult to account for. In any case, the difference in freezing levels between 3h and 24h is interesting and adequate in trying to normalize freezing per group even if this was obtained from different batches.”
As Reviewer 2 suggested to summarize the statistics, we represented all averages and SEMs in the Table 1 and 2 refer to the Figure 2 and 4. The range of SEM in 3 h after the conditioning in non-AAV injected mice were from 1.373 to 11.01%, while the range of SEM in 3 h after the conditioning in AAV-injected mice were from 0.394 to 5.058. Therefore, we do not think the variability of freezing is not higher among AAV injected mice than non-injected mice. Please see the Table 1 and 2.
Reviewer 1 comment 11: “Page 6, line 189. This was the first time, together with the conclusion of the first part of the results, we understood that the authors use “fear generalization” for the increase in freezing observed at 24h after the pre-exposure to context at 3h post-conditioning… This is problematic because this is only clearly stated in the discussion and reflects and interpretation of the authors, it is difficult to follow the paper if this is not defined before. Again, we would suggest that the authors come with a new name for the phenomenon they observe, to distinguish it from the more traditional idea of generalization. This might help the reader to not get distracted in terminology issues.”
As related suggestion by Reviewer 1 in the comment 2 and 3, we have performed extensive revised throughout the manuscript, Title, Abstract, Introduction, Results, Discussion and Conclusion as follow:
Introduction (page 4, line 1-page 5, line 3):
Indeed, the increase in generalization was due to a loss of detailed information about the context and not fear incubation [7]. However, in contrast to false context fear memory, inducing “fear generalization” across environments requires at least a week after exposure to traumatic events in rodents and it has been demonstrated that one month is required to induce freezing in novel environments [8]. According to the hypothesis that a shared neural ensemble linking distinct memories that encode close in time due to a temporary increase in neuronal excitability, a subsequent memory to the neuronal ensemble encodes the first memory. In rodents, it has been demonstrated that the first memory can strengthen the second memory within a day but not across a week [9]. Although, a false context fear memory of a traumatic event would be formed and subsequently lead to the formation of fear generalization, the considerable neuronal mechanism of fear generalization might not be the same as the concept of neuronal ensemble encoding the first memory that can lead to false fear memories.
Introduction (page 4, line 1-page 5, line 3):
The present study endeavored to understand the formation mechanism of false fear memories after traumatic stress events based on the concept of close in time encoding due to a temporary increase in neuronal excitability of subsequent memories to the neuronal ensemble after encoding an initial memory. Therefore, the present study focuses on the cause of spatial memory failure by source confusion after inducing false fear memory using two different contexts within three hours after conditional stimulation (SC). Then, we investigated the role of hypothalamic CRF on the induction of false context fear memories using mice with hypothalamic Crf knockdown (Hy-Crf-KD) and hypothalamic CRF overexpression (Hy-CRF-OE), respectively. Adeno-associated virus (AAV)-mediated mouse Crf small hairpin RNA (shRNA) was used to knock down Crf expression, and an AAV carrying mouse Crf cDNA was constructed for the overexpression of CRF.
Reviewer 1 comment 12: “Fig 5. We appreciate the effort in trying to link anxiety leading to “false memory” to freezing the day after, but do not quite understand the logic behind this analysis and what additional information different from freezing this brings… it is obvious that the longer a mouse will spend freezing, the less distance it will move. Correlation of 3h distance vs. 24h freezing is lost in genetically manipulated mice, but correlation is not causation (statements as in page10 of 17, line 358 should be weakened). This only reinforces our opinion that it is extremely difficult to extract any conclusion about anxiety levels from assessments that are no de-coupled from fear conditioning. Assessing distance travelled in context A before CS, as done in fig4C, is the closest and most appropriate way.”
We acknowledge that this part is not adequate and awkward phrase. Also, we should not discuss the relationship from the result between the center preference in the box and the anxiety level. However, the findings the central preference in the Box B 3h and the enhancement of false fear memory in 24 h after the conditioning is very important. We are currently the possibility that the center preference is a kind of active coping against novel environment at 3 h and decrease the enhancement of false fear memory in 24 h. Therefore, we changed the sentences in Results and Discussion sections as follow:
Results (page 4, line 1-page 5, line 3):
Therefore, there is a possibility that the both of the facilitation of false fear memory formation and the potentiated false context fear memory level were due to the hypothalamic CRF expression level rather than adapting ability to the novel environment. In the next, we investigate whether the preference for the central zon in the Box B 3 h after the CS during the Context B affect to the false context fear memory level in 24 h. The freezing time in 24 h was negatively correlated with the distance traveled by control mice in the central zone in 3 h (control mice: Pearson’s correlation, t = -2.574, df = 11, p < 0.05, R2 = 0.376, Fig. 5C), indicating that exploratory as an active coping behavior into the central zone of a novel chamber during Context B 3 h might affect the freezing level in Context B 24 h.
Discussion (page 4, line 1-page 5, line 3):
Finally, in the present study, we found that the preference for the central zone in Box B at 3 h—as a novel environment after the CS—affected the freezing level during Context B 24 h in control mice. While hypothalamic CRF did not affect the preference for the central zone during Context B 3 h, the significant correlation between the preference for the central zone in a novel chamber (Box B 3 h) and the freezing level during Context B 24 h was abolished in Hy-Crf-KD and Hy-CRF-OE mice. Therefore, we also suggest that adequate active coping behavior when in a novel environment may decrease and subsequently induce false context fear memory formation.
Reviewer 1 comment 12: “- Line 30, evidence is not plural”
We thank your suggestion for our grammatical error in the manuscript. We corrected from “influence” to “influences”.
Reviewer 1 comment 13: “- Line 64: The statement “in patients with PTSD” is grossly misleading, as mice are being studied here.”
We acknowledge that the statement “in patients with PTSD” is not adequate in our mouse study. According to the correction of our focus in this study, we focus on the false fear memory, but not fear generalization. Therefore, we excluded this phrase “in patients with PTSD”.
Reviewer 1 comment 14: “- Line 143-147. This is not a proper sentence, grammatically speaking.”
We acknowledge that this part of our English was very poor. Therefore, we corrected this sentence in the Results section as follow:
Results (page 4, line 1-page 5, line 3):
To investigate the effects of HPA-axis activity on the formation of false fear memory using mice that were injected with AAV-PHP.eB-produced virus which can knockdown the Crf gene by the shRNA system and the overexpression of CRF in the hypothalamus (Fig. 3 and Fig. S2).
Responses to Reviewer 2:
Reviewer 2 comment 1: “Abstract: Please rephrase the results and conclusion to make them clear for readers to understand.”
We acknowledge that our abstract contained many results and lack of the background of the present study. Also, due to the limitation of number of words, it was not clear what our conclusion is refer to. As Reviewer 1 suggested that the our interpretation of definition of “fear generalization” is wrong and the “fear generalization” we claimed in first our draft should be “enhancement of false memory” according to the classical definition of “fear generalization” which are used in the symptom of PTSD. We agreed the Reviewer 1 suggestion and extensively corrected our claim throughout the manuscript. So, we also improved the abstract to make it clear for reader to understand by adding the sentence to describe the background before the results and mention the conclusion more clearly as Reviewer 2 suggestion in the Abstract section as follow:
Abstract (page 4, line 1-page 5, line 3):
Traumatic events frequently produce false fear memories. We investigated the effect of hypothalamic corticotropin-releasing factor (CRF) knockdown (Hy-Crf-KD) or overexpression (Hy-CRF-OE) on contextual fear memory, because fear stress releases CRF and hypothamic–pituitry–adrenal axis activation affects the memory system. Mice were placed in a chamber with a footshock as a conditioning stimulation (CS) during Context A and then exposed to a novel chamber without CS as Context B 3 h (B-3h) and 24 h (B-24h). The freezing response in B-3h was intensified in these mice compared to control mice that were not exposed to CS, indicating that false fear memory was formed after 3 h. The within-group freezing level in B-24h was higher than that in B-3h, indicating that false context fear memories were enhanced in B-24h when the mice experienced B-3h. The difference in freezing levels between B-3h and B-24h in Hy-Crf-KD mice was larger than that in the control. In Hy-CRF-OE mice, the freezing level in B-3h was higher than that in control and Hy-Crf-KD mice, while the freezing level in B-24h was similar to that in B-3h. Locomotor activity before CS and freezing level during CS were similar among the groups. Therefore, we hypothesized that Hy-Crf-KD enhances the false context fear memory in a day, while Hy-CRF-OE potentiates the false fear memory within a few hours.
Reviewer 2 comment 2: “In general, I recommend authors to use more evidence to back their claims, especially in the Introduction of the article, which I believe is currently lacking. Thus, I recommend the authors to attempt to deepen the subject of their manuscript, as the bibliography is too concise: nonetheless, in my opinion, less than 60/70 articles for a research paper are insufficient. Indeed, currently, authors cite only 50 papers, and they are too low. Therefore, I suggest the authors to focus their efforts on researching more relevant literature: I believe that adding more studies and reviews will help them to provide better and more accurate background to this study.”
We acknowledge that our first manuscript lacked the background of our study and add many reference papers as the Reviewer 2 suggested. Please see our reply to the Reviewer 2 comment 3 what and where we changed the sentences as a background according to the suggestions of Reviewer 2 comment 3 in the Introduction section.
Reviewer 2 comment 3: “Introduction: The authors decided to focus specifically on animal models to discuss the role of hypothalamic CRF in the induction of false context fear memory and fear generalization, and, in my opinion, this is too limiting. Thus, I suggest reshaping the Introduction section, which seems not enough extensive and it does not seem to consider, in most cases, all the available studies in the literature that have acknowledged inhomogeneous and dispersive. I think that more information about the ability to use contextual information to modulate the expression of fear would provide a better background here. I suggest the authors to make such effort to provide a brief overview of the pertinent published literature that offer a perspective on altered brain circuits underlying aberrant fear learning process in PTSD, because as it stands, this information is not highlighted in the text. In this regard, I would recommend focusing on the role that the ventromedial prefrontal cortex and hippocampus network have in context-dependent fear learning, addressing how impairments in these brain regions affect fear memories: evidence from a recent theoretical review (https://doi.org/10.1038/s41380-021-01326-4) that focused on neurobiology of fear conditioning, analyzed the role of the ventromedial prefrontal cortex (vmPFC) was analyzed in the processing of safety-threat information and their relative value, and how this region is fundamental for the evaluation and representation of stimulus-outcome’s value needed to produce sustained physiological responses. Also, I believe that a recent yet relevant perspective manuscript (https://doi.org/10.17219/acem/146756) might be of interest: here the focus was on providing a deeper understanding of human learning neural networks, particularly on human PFC crucial role, that might also contribute to the advancement of alternative, more precise and individualized treatments for psychiatric disorders.”
We acknowledge that our first manuscript lacked the background of our study as Reviewer 2 suggested. Therefore, we added many reference papers as Reviewer 2 suggested and extensively improved the Introduction section as follow:
Introduction (page 4, line 1-page 5, line 3):
Cortisol modulates various learning and memory processes depending on the particular timing of cortisol increases relative to encoding, consolidation and retrieval [10]. Long-term dysregulation of cortisol systems after activation of the HPA-axis has been suggested to have lasting effects on vulnerable areas of the hippocampus, amygdala, and medial prefrontal cortex [11]. The acquisition of fear-associative memory requires various HPA-axis-related brain processes of coordinated neural activity within the amygdala [10,12,13], prefrontal cortex (PFC) [14,15], and hippocampus [14-17]. The amygdala plays a key role in the acquisition of fear learning, while the PFC and hippocampus are two other crucial neural structures that contribute to this process, collectively representing the neural network of fear conditioning [18]. The ventral part of the medial PFC may also play a major role in fear conditioning [19-21]. Subsequent traumatic events within a few hours raise the corticotropin-releasing factor (CRF) and the activation of the HPA-axis which might affect to the memory consolidation. However, how the stress-induced activation of HPA-axis, contributes to the formation of false context fear memories remains unclear.
References:
- Merz, C.J.; Hermann, A.; Stark, R.; Wolf, O.T. Cortisol modifies extinction learning of recently acquired fear in men. Social cognitive and affective neuroscience 2014, 9, 1426-1434, doi:10.1093/scan/nst137.
- Bremner, J.D. Traumatic stress: effects on the brain. Dialogues in clinical neuroscience 2006, 8, 445-461, doi:10.31887/DCNS.2006.8.4/jbremner.
- Skórzewska, A.; Lehner, M.; WisÅ‚owska-Stanek, A.; TurzyÅ„ska, D.; Sobolewska, A.; KrzÄ…Å›cik, P.; Szyndler, J.; Maciejak, P.; Chmielewska, N.; KoÅ‚osowska, K.; et al. Individual susceptibility or resistance to posttraumatic stress disorder-like behaviours. Behavioural brain research 2020, 386, 112591, doi:10.1016/j.bbr.2020.112591.
- Li, G.; Wang, G.; Shi, J.; Xie, X.; Fei, N.; Chen, L.; Liu, N.; Yang, M.; Pan, J.; Huang, W.; et al. trans-Resveratrol ameliorates anxiety-like behaviors and fear memory deficits in a rat model of post-traumatic stress disorder. Neuropharmacology 2018, 133, 181-188, doi:10.1016/j.neuropharm.2017.12.035.
- Perrine, S.A.; Eagle, A.L.; George, S.A.; Mulo, K.; Kohler, R.J.; Gerard, J.; Harutyunyan, A.; Hool, S.M.; Susick, L.L.; Schneider, B.L.; et al. Severe, multimodal stress exposure induces PTSD-like characteristics in a mouse model of single prolonged stress. Behavioural brain research 2016, 303, 228-237, doi:10.1016/j.bbr.2016.01.056.
- Ribeiro, T.O.; Bueno-de-Camargo, L.M.; Waltrick, A.P.F.; de Oliveira, A.R.; Brandão, M.L.; Munhoz, C.D.; Zanoveli, J.M. Activation of mineralocorticoid receptors facilitate the acquisition of fear memory extinction and impair the generalization of fear memory in diabetic animals. Psychopharmacology 2020, 237, 529-542, doi:10.1007/s00213-019-05388-9.
- Sarabdjitsingh, R.A.; Zhou, M.; Yau, J.L.; Webster, S.P.; Walker, B.R.; Seckl, J.R.; Joëls, M.; Krugers, H.J. Inhibiting 11β-hydroxysteroid dehydrogenase type 1 prevents stress effects on hippocampal synaptic plasticity and impairs contextual fear conditioning. Neuropharmacology 2014, 81, 231-236, doi:10.1016/j.neuropharm.2014.01.042.
- Merz, C.J.; Hamacher-Dang, T.C.; Stark, R.; Wolf, O.T.; Hermann, A. Neural Underpinnings of Cortisol Effects on Fear Extinction. Neuropsychopharmacology : official publication of the American College of Neuropsychopharmacology 2018, 43, 384-392, doi:10.1038/npp.2017.227.
- Battaglia, S. Neurobiological advances of learned fear in humans. Advances in clinical and experimental medicine : official organ Wroclaw Medical University 2022, 31, 217-221, doi:10.17219/acem/146756.
- Battaglia, S.; Garofalo, S.; di Pellegrino, G.; Starita, F. Revaluing the Role of vmPFC in the Acquisition of Pavlovian Threat Conditioning in Humans. The Journal of neuroscience : the official journal of the Society for Neuroscience 2020, 40, 8491-8500, doi:10.1523/jneurosci.0304-20.2020.
- Fullana, M.A.; Harrison, B.J.; Soriano-Mas, C.; Vervliet, B.; Cardoner, N.; Àvila-Parcet, A.; Radua, J. Neural signatures of human fear conditioning: an updated and extended meta-analysis of fMRI studies. Molecular psychiatry 2016, 21, 500-508, doi:10.1038/mp.2015.88.
- Battaglia, S.; Harrison, B.J.; Fullana, M.A. Does the human ventromedial prefrontal cortex support fear learning, fear extinction or both? A commentary on subregional contributions. Molecular psychiatry 2022, 27, 784-786, doi:10.1038/s41380-021-01326-4.
Reviewer 2 comment 4: “Introduction: Following the first point raised, I would also recommend another recent opinion manuscript which provided novel ‘functional interplay between central and autonomic nervous systems in human fear conditioning’, and highlights the crucial role of the prefrontal cortex. Finally, authors also might to consider some studies that have focused on this topic (https://doi.org/10.3390/biomedicines10010076; https://doi.org/10.3390/biomedicines9050517).”
We acknowledge that our first manuscript lacked the background of HPA-axis activity related symptom. Therefore, we added the following sentence in the Introduction section as Reviewer 2 suggested;
Introduction (page 4, line 1-page 5, line 3):
Moreover, HPA-axis activity links the varius symptom regarding the autonomic function, anxiety, locomotor activity or cognitive functions [23].
References:
- Tanaka, M.; Vécsei, L. Editorial of Special Issue "Crosstalk between Depression, Anxiety, and Dementia: Comorbidity in Behavioral Neurology and Neuropsychiatry". Biomedicines 2021, 9, doi:10.3390/biomedicines9050517.
Reviewer 2 comment 5: “Behavioral tests: Could the authors provide the specific number of mice that were used in the experiments?”
We acknowledge that our first manuscript lacked the total number of mice were used in the Methods section as follow: The number of mice we used in each group of each experiment was shown in Figure legends. Also, as Reviewer suggested, we put two summary Table1 and 2 in the revised version.
Methods (page 4, line 1-page 5, line 3):
Total 15 mice were used for measuring the plasma corticosterone concentration (5 mice for each group). For the behavioral test, in non-AAV infected mouse test (Figure 2 and Figure S2), 54 mice (24 mice for Figure 2 and 30 mice for Figure 1S) and mice were used and AAV infected mouse test (Figure 4), 57 mice were used for behavioral tests. Total 126 mice were used in the present study. In the test using a group of A-B (CS-) mice, A-B (CS+) mice, ABB (CS-) mice, ABB (CS+) mice in Figure 2 were performed at the same time respectively. Also, all of AAV injected mice with CS group (Hy-Control (CS+), Hy-Crf-KD (CS+) and Hy-CRF-OE (CS+)) were tested at the same time and all of AAV injected mice without the CS group (Hy-Control (CS-), Hy-Crf-KD (CS-) and Hy-CRF-OE (CS-)) were tested at the same time. Every day, 3 to 4 mice were tested and repeated these tests until the number of mice were enough to analyzing.
Reviewer 2 comment 6: “Contextual fear conditioning test: This paragraph that explains how mice were fear conditioned is the most important part of the study and should clearly describe all the experimental sessions in detail; therefore, this section might be improved by including further explanations, allowing the effective communication of experimental procedures.”
We acknowledge that our first manuscript lacked the detail of the method of conditioning test in the present study. Because we described some sentences about the conditioning test in the first paragraph of our first submission in the “Contextual fear conditioning” which also describing the apparatus and it might make a confusion for reader. Therefore, we separate the part of “Apparatus for the contextual fear conditioning test” and “Contextual fear conditioning” in the Methods section as follow:
Methods (page 4, line 1-page 5, line 3):
The mice were handled for 1 min for three consecutive days before the behavioral test. All mice were habituated to Box A 60 min before Context A for 15 min. Mice were placed in Box A again for 3 min, and then electrical shocks (1.0 mA for 2 s was delivered three times at 100-s intervals) were delivered 3 times at 100-s intervals as a fear CS (Context A) in Box A 3 min after the mice were placed on the grid floor, designated as Context A. All mice were returned to their home cage until the next context after Context A for fear conditioning in Box A.
Methods (page 4, line 1-page 5, line 3):
In all test in the present study, the chambers were thoroughly cleaned with a 70% ethanol solution between mice every times.
Reviewer 2 comment 7: “Results: In my opinion, this section is well organized, but it illustrates findings in an excessively broad way, without really providing full statistical details, to ensure in-depth understanding and replicability of the findings. Also, in my opinion, it is necessary for the authors to present their findings using summary tables.”
We acknowledge that the statistics were not sufficient for describing all results in our first manuscript. As Reviewer 2 suggested, we put two summary Table 1 and 2 for detail information about the results of Figure 2 and 4 including statistics in our revised version of our manuscript. Please see the two tables in the template manuscript.
Reviewer 2 comment 8: “Discussion: In this final section, the authors described the results and their argumentation and captured the state of the art well; however, I would have liked to see some views on a way forward. I believe that the authors should make an effort, trying to explain the theoretical implication as well as the translational application of this research article, to adequately convey what they believe is the take-home message of their study. Discussion of theoretical and methodological avenues in need of refinement is necessary, as well as suggestions of a path forward in understanding the role of hypothalamic CRF on the induction of false context fear memory and fear generalization. In this regard, recent evidence suggests that the application of new methods in Neuroinflammatory disorders’ treatment, such as the Non-invasive brain stimulation techniques (NIBS), have shown promising results in humans (https://doi.org/10.1038/s41398-020-0851-5). Importantly, I recommend referring to recent studies that revealed that the application of NIBS induces long-lasting effects, noninvasively modulating the cortical excitability, and modulates a variety of cognitive functions altered in patients that suffer from PTSD: for example, a recent review (https://doi.org/10.1016/j.neubiorev.2021.04.036) on the potential and effectiveness of non-invasive brain stimulation (NIBS) to interfere and modulate the abnormal activity of neural circuits (i.e., amygdala-mPFC-hippocampus) involved in the acquisition and consolidation of fear memories, which are altered in many mood psychiatric disorders (i.e., anxiety disorder, specific phobias, post-traumatic stress disorder or depression), would be of interest. Accordingly, another recent manuscript (https://doi.org/10.1016/j.jad.2021.02.076) focused on the same topic, illustrating the therapeutic potential of NIBS as a valid alternative in the treatment of untypically persistent memories that characterized those patients that do not respond to psychotherapy and/or drug treatments. In addition to the previously mentioned literature, authors might also see these additional studies that have focused on the efficacy of NIBS and IBS (https://doi.org/10.3389/fpsyt.2018.00201; https://doi.org/10.3389/fnagi.2020.578339).”
We appreciate the Reviewer 2 suggestion to discuss the important background and possibility raised from our results in the present study. As related to the Reviewer 2 comment 2 and 3, we discussed about the amygdala-mPFC-hippocampus in Discussion section as follow:
Discussion (page 4, line 1-page 5, line 3):
Functional alterations of the neural network underlying fear conditioning might contribute to the etiology of fear-related psychiatric disease, including PTSD [18]. Fear associative memory acquisition of fear learning requires coordinated neural activity within the amygdala, prefrontal cortex (PFC), and hippocampus [17-21,48,49]. In addition, cortisol exerted a critical impact on the amygdala–hippocampus–ventromedial PFC network underlying fear and extinction memories [17]. Dysregulation of negative feedback by cortisol suppressing the CRF release after the long-term activation of activation of the HPA-axis affects the hippocampus, amygdala, and medial PFC [11]. Inactivation of prefrontal inputs into the nucleus reuniens or direct silencing of nucleus reuniens projections enhances fear memory generalization [27]. Xu and Südhof demonstrated the generalization of memory attributes for a particular context by processing information from the medial PFC enroute to the hippocampus within a day after conditional training [27]. Decreased connectivity between the amygdala and medial PFC were shown to be related to memory intrusion and the re-experiencing of traumatic events [50]. Interestingly, repetitive transcranial magnetic stimulation of the right dorsolateral PFC appears to have a positive effect of reducing core symptoms in patients with PTSD [50]. Therefore, it is crucially important to investigate the role of CRF and cortisol on the neuronal networks including the amygdala–hippocampus–ventromedial PFC for their contribution to both the acquisition of fear memories and the consolidation of imprecise fear memories, although there remains a limitation for elucidating the neuronal mechanisms using the rodent model, because humans and animals have dissimilar functional PFC neuroarchitecture.
References:
- Bremner, J.D. Traumatic stress: effects on the brain. Dialogues in clinical neuroscience 2006, 8, 445-461, doi:10.31887/DCNS.2006.8.4/jbremner.
- Merz, C.J.; Hamacher-Dang, T.C.; Stark, R.; Wolf, O.T.; Hermann, A. Neural Underpinnings of Cortisol Effects on Fear Extinction. Neuropsychopharmacology : official publication of the American College of Neuropsychopharmacology 2018, 43, 384-392, doi:10.1038/npp.2017.227.
- Battaglia, S. Neurobiological advances of learned fear in humans. Advances in clinical and experimental medicine : official organ Wroclaw Medical University 2022, 31, 217-221, doi:10.17219/acem/146756.
- Battaglia, S.; Garofalo, S.; di Pellegrino, G.; Starita, F. Revaluing the Role of vmPFC in the Acquisition of Pavlovian Threat Conditioning in Humans. The Journal of neuroscience : the official journal of the Society for Neuroscience 2020, 40, 8491-8500, doi:10.1523/jneurosci.0304-20.2020.
- Fullana, M.A.; Harrison, B.J.; Soriano-Mas, C.; Vervliet, B.; Cardoner, N.; Àvila-Parcet, A.; Radua, J. Neural signatures of human fear conditioning: an updated and extended meta-analysis of fMRI studies. Molecular psychiatry 2016, 21, 500-508, doi:10.1038/mp.2015.88.
- Battaglia, S.; Harrison, B.J.; Fullana, M.A. Does the human ventromedial prefrontal cortex support fear learning, fear extinction or both? A commentary on subregional contributions. Molecular psychiatry 2022, 27, 784-786, doi:10.1038/s41380-021-01326-4.
- Xu, W.; Südhof, T.C. A neural circuit for memory specificity and generalization. Science (New York, N.Y.) 2013, 339, 1290-1295, doi:10.1126/science.1229534.
- Borgomaneri, S.; Battaglia, S.; Sciamanna, G.; Tortora, F.; Laricchiuta, D. Memories are not written in stone: Re-writing fear memories by means of non-invasive brain stimulation and optogenetic manipulations. Neuroscience and biobehavioral reviews 2021, 127, 334-352, doi:10.1016/j.neubiorev.2021.04.036.
- Stark, R.; Wolf, O.T.; Tabbert, K.; Kagerer, S.; Zimmermann, M.; Kirsch, P.; Schienle, A.; Vaitl, D. Influence of the stress hormone cortisol on fear conditioning in humans: evidence for sex differences in the response of the prefrontal cortex. NeuroImage 2006, 32, 1290-1298, doi:10.1016/j.neuroimage.2006.05.046.
- Kan, R.L.D.; Zhang, B.B.B.; Zhang, J.J.Q.; Kranz, G.S. Non-invasive brain stimulation for posttraumatic stress disorder: a systematic review and meta-analysis. Translational psychiatry 2020, 10, 168, doi:10.1038/s41398-020-0851-5.
Reviewer 2 comment 9: “Even though it is not mandatory, I believe that the ‘Conclusions’ section would be useful to adequately indicate convey what the authors believe is the take-home message of their study, and therefore provide a synthesis of the data presented in the paper as well as possible keys to advancing research and understanding of the prevalence of depression in post-stroke patients.”
We acknowledge that the Conclusion section should be provided for the reader of our study. Therefore, we put the “Conclusion” section after the Discussion as follow:
Conclusion (page 4, line 1-page 5, line 3):
In the present study, we found that the exposure to novel but similar contexts with the environment of traumatic fear experience within a few hours forms a false fear memory that is contextually enhanced within a day after the traumatic fear event. These results suggest that exposure to a novel context within a few hours after a traumatic fear event might interfere in the consolidation of precise fear memories of traumatic events and enhance false context fear memory within 24 h. Knockdown of hypothalamic Crf increases the freezing level and potentiates the false context fear memory at 24 h after fear conditioning. Meanwhile, overexpression of hypothalamic CRF enhances the onset of false fear memory formation within a few hours after a traumatic fear event. Cortisol has been suggested to exert a critical impact on the amygdala–hippocampus–ventromedial PFC network underlying the acquisition of fear memory, consolidating precise fear memories and fear generalization. Therefore, it is important that how the HPA-axis contributes to the formation and the potentiation of false context fear memories and to explore the clinical implications such as the treatment of psychiatric diseases such as PTSD and advacements in CRF-related cognitive impairment such as a depression in post-stroke [51] in the future.
Reference:
- Barra de la Tremblaye, P.; Plamondon, H. Alterations in the corticotropin-releasing hormone (CRH) neurocircuitry: Insights into post stroke functional impairments. Frontiers in neuroendocrinology 2016, 42, 53-75, doi:10.1016/j.yfrne.2016.07.001.
Reviewer 2 comment 10: “In according to the previous comment, I would ask the authors to better define a proper ‘Limitations and future directions’ section before the end of the manuscript, in which authors can describe in detail and report all the technical issues brought to the surface.”
We acknowledge that the limitation of our study is important because all of our results were obtained from the rodents. As Reviewer 2 suggested, how our results develop and reflect to the human disease in the future is important for the readers. Although we did not separate the as “Limitation” from the Discussion section, we added the sentences in the Discussion section as follow:
Discussion (page 4, line 1-page 5, line 3):
Therefore, it is crucially important to investigate the role of CRF and cortisol on the neuronal networks including the amygdala–hippocampus–ventromedial PFC for their contribution to both the acquisition of fear memories and the consolidation of imprecise fear memories, although there remains a limitation for elucidating the neuronal mechanisms using the rodent model, because humans and animals have dissimilar functional PFC neuroarchitecture.
Reviewer 2 comment 11: “References: Authors should consider revising the bibliography, as there are several incorrect citations. Indeed, according to the Journal’s guidelines, they should provide the abbreviated journal name in italics, the year of publication in bold, the volume number in italics for all the references.”
We thank the Reviewer 2 suggestions. We corrected the style of References. Please see the Reference section in the template manuscript.

Reviewer 2 Report
Kasama and colleagues investigated in the present study entitled ‘Formation of false context fear memory and fear generalization are respectively regulated by distinct expression levels of hypothalamic corticotropin-releasing factor in mice’, the current status of knowledge of hypothalamic corticotropin-releasing factor (CRF) knockdown or overexpression on contextual fear memory. For this purpose, the authors exposed mice to a fear conditioning paradigm in Context A (with conditioning stimulation), then exposed them to a fear extinction phase in Context B (without conditioning stimulation) after 3 and 24 hours. Results showed that freezing levels in Context B after 24 hours were higher than the ones after 3 hours, indicating that fear generalization was formed in Context B 24 hours after mice were exposed to Context B 3 hours later. Moreover, in mice with CRF overexpression, the freezing level in B-3h was higher than that in control and in mice with CRF knockdown, while freezing level in B-24h was similar to that in B-3h. The authors concluded by stating that CRF overexpression potentiates false fear memory, while CRF knockdown enhances fear generalization.
The main strength of this original research article is that it addresses an interesting and timely question, investigating the effect of HPA-axis activity, specifically CRF activity, on the formation of the false fear memory and fear generalization. In general, I think the idea of this article is really interesting and the authors’ fascinating observations on this timely topic may be of interest to the readers of the International Journal of Molecular Sciences. However, some comments, as well as some crucial evidence that should be included to support the authors’ argumentation, needed to be addressed to improve the quality of the manuscript, its adequacy, and its readability prior to the publication in the present form, in particular reshaping parts of the Introduction and Discussion sections by adding more evidence and theoretical constructs.
Please consider the following comments:
- Abstract: Please rephrase the results and conclusion to make them clear for readers to understand.
- In general, I recommend authors to use more evidence to back their claims, especially in the Introduction of the article, which I believe is currently lacking. Thus, I recommend the authors to attempt to deepen the subject of their manuscript, as the bibliography is too concise: nonetheless, in my opinion, less than 60/70 articles for a research paper are insufficient. Indeed, currently, authors cite only 50 papers, and they are too low. Therefore, I suggest the authors to focus their efforts on researching more relevant literature: I believe that adding more studies and reviews will help them to provide better and more accurate background to this study.
- Introduction: The authors decided to focus specifically on animal models to discuss the role of hypothalamic CRF in the induction of false context fear memory and fear generalization, and, in my opinion, this is too limiting. Thus, I suggest reshaping the Introduction section, which seems not enough extensive and it does not seem to consider, in most cases, all the available studies in the literature that have acknowledged inhomogeneous and dispersive. I think that more information about the ability to use contextual information to modulate the expression of fear would provide a better background here. I suggest the authors to make such effort to provide a brief overview of the pertinent published literature that offer a perspective on altered brain circuits underlying aberrant fear learning process in PTSD, because as it stands, this information is not highlighted in the text. In this regard, I would recommend focusing on the role that the ventromedial prefrontal cortex and hippocampus network have in context-dependent fear learning, addressing how impairments in these brain regions affect fear memories: evidence from a recent theoretical review (https://doi.org/10.1038/s41380-021-01326-4) that focused on neurobiology of fear conditioning, analyzed the role of the ventromedial prefrontal cortex (vmPFC) was analyzed in the processing of safety-threat information and their relative value, and how this region is fundamental for the evaluation and representation of stimulus-outcome’s value needed to produce sustained physiological responses. Also, I believe that a recent yet relevant perspective manuscript (https://doi.org/10.17219/acem/146756) might be of interest: here the focus was on providing a deeper understanding of human learning neural networks, particularly on human PFC crucial role, that might also contribute to the advancement of alternative, more precise and individualized treatments for psychiatric disorders.
- Introduction: Following the first point raised, I would also recommend another recent opinion manuscript which provided novel ‘functional interplay between central and autonomic nervous systems in human fear conditioning’, and highlights the crucial role of the prefrontal cortex. Finally, authors also might to consider some studies that have focused on this topic (https://doi.org/10.3390/biomedicines10010076; https://doi.org/10.3390/biomedicines9050517).
- Behavioral tests: Could the authors provide the specific number of mice that were used in the experiments?
- Contextual fear conditioning test: This paragraph that explains how mice were fear conditioned is the most important part of the study and should clearly describe all the experimental sessions in detail; therefore, this section might be improved by including further explanations, allowing the effective communication of experimental procedures.
- Results: In my opinion, this section is well organized, but it illustrates findings in an excessively broad way, without really providing full statistical details, to ensure in-depth understanding and replicability of the findings. Also, in my opinion, it is necessary for the authors to present their findings using summary tables.
- Discussion: In this final section, the authors described the results and their argumentation and captured the state of the art well; however, I would have liked to see some views on a way forward. I believe that the authors should make an effort, trying to explain the theoretical implication as well as the translational application of this research article, to adequately convey what they believe is the take-home message of their study. Discussion of theoretical and methodological avenues in need of refinement is necessary, as well as suggestions of a path forward in understanding the role of hypothalamic CRF on the induction of false context fear memory and fear generalization. In this regard, recent evidence suggests that the application of new methods in Neuroinflammatory disorders’ treatment, such as the Non-invasive brain stimulation techniques (NIBS), have shown promising results in humans (https://doi.org/10.1038/s41398-020-0851-5). Importantly, I recommend referring to recent studies that revealed that the application of NIBS induces long-lasting effects, noninvasively modulating the cortical excitability, and modulates a variety of cognitive functions altered in patients that suffer from PTSD: for example, a recent review (https://doi.org/10.1016/j.neubiorev.2021.04.036) on the potential and effectiveness of non-invasive brain stimulation (NIBS) to interfere and modulate the abnormal activity of neural circuits (i.e., amygdala-mPFC-hippocampus) involved in the acquisition and consolidation of fear memories, which are altered in many mood psychiatric disorders (i.e., anxiety disorder, specific phobias, post-traumatic stress disorder or depression), would be of interest. Accordingly, another recent manuscript (https://doi.org/10.1016/j.jad.2021.02.076) focused on the same topic, illustrating the therapeutic potential of NIBS as a valid alternative in the treatment of untypically persistent memories that characterized those patients that do not respond to psychotherapy and/or drug treatments. In addition to the previously mentioned literature, authors might also see these additional studies that have focused on the efficacy of NIBS and IBS (https://doi.org/10.3389/fpsyt.2018.00201; https://doi.org/10.3389/fnagi.2020.578339).
- Even though it is not mandatory, I believe that the ‘Conclusions’ section would be useful to adequately indicate convey what the authors believe is the take-home message of their study, and therefore provide a synthesis of the data presented in the paper as well as possible keys to advancing research and understanding of the prevalence of depression in post-stroke patients.
- In according to the previous comment, I would ask the authors to better define a proper ‘Limitations and future directions’ section before the end of the manuscript, in which authors can describe in detail and report all the technical issues brought to the surface.
- References: Authors should consider revising the bibliography, as there are several incorrect citations. Indeed, according to the Journal’s guidelines, they should provide the abbreviated journal name in italics, the year of publication in bold, the volume number in italics for all the references.
Overall, the manuscript contains 5 figures and 50 references. I believe that this manuscript may carry important value in investigating the effect of HPA-axis activity, specifically CRF activity, on the formation of the false fear memory and fear generalization.
I hope that, after these careful revisions, the paper can meet the Journal’s high standards for publication. I am available for a new round of revision of this article.
I declare no conflict of interest regarding this manuscript.
Best regards,
Reviewer
Author Response
To the Editors Fukushima, Japan
International Journal of Molecular Sciences May 22nd, 2022
Subject: Revision for Manuscript ID: ijms-1718457
With this letter we would like to submit a revised version of the article entitled “Formation of false context fear memory and fear generalization are respectively regulated by distinct expression levels of hypothalamic corticotropin-releasing factor in mice” by Kenjiro Seki submitted for publication to the International Journal of Molecular Sciences.
At first, we would like to thank the reviewers for their valuable critical comments and excellent suggestions, which have further contributed to the present improvement of the manuscript. We essentially followed the reviewer’s comments and arguments and address all points that had been raised. The manuscript was changed accordingly. We have highlighted all the changes made in the original manuscript below in response to reviewer. According to reviewer 1 suggestions we have also changed the title of the manuscript “Formation of false context fear memory is regulated by hypothalamic corticotropin-releasing factor in mice”.
The Abstract, Introduction, Results, Discussion and Methods section were modified according to the reviewer comments/suggestions and additional changes to make reading easier and clearer. References are updated accordingly.
For reference, the reviewer’s comments are shown in quotations. Please note that the referring page and line number shown in each our response to the reviewer comment corresponds to the line number of each page in the untracked-changed version of revised manuscript. We also submitted tracking version after the untracking version of template manuscript as one file.
Best Regards,
Kenjiro Seki, Ph.D.
Responses to Reviewer 2:
Reviewer 2 comment 1: “Abstract: Please rephrase the results and conclusion to make them clear for readers to understand.”
We acknowledge that our abstract contained many results and lack of the background of the present study. Also, due to the limitation of number of words, it was not clear what our conclusion is refer to. As Reviewer 1 suggested that the our interpretation of definition of “fear generalization” is wrong and the “fear generalization” we claimed in first our draft should be “enhancement of false memory” according to the classical definition of “fear generalization” which are used in the symptom of PTSD. We agreed the Reviewer 1 suggestion and extensively corrected our claim throughout the manuscript. So, we also improved the abstract to make it clear for reader to understand by adding the sentence to describe the background before the results and mention the conclusion more clearly as Reviewer 2 suggestion in the Abstract section as follow:
Abstract (page 4, line 1-page 5, line 3):
Traumatic events frequently produce false fear memories. We investigated the effect of hypothalamic corticotropin-releasing factor (CRF) knockdown (Hy-Crf-KD) or overexpression (Hy-CRF-OE) on contextual fear memory, because fear stress releases CRF and hypothamic–pituitry–adrenal axis activation affects the memory system. Mice were placed in a chamber with a footshock as a conditioning stimulation (CS) during Context A and then exposed to a novel chamber without CS as Context B 3 h (B-3h) and 24 h (B-24h). The freezing response in B-3h was intensified in these mice compared to control mice that were not exposed to CS, indicating that false fear memory was formed after 3 h. The within-group freezing level in B-24h was higher than that in B-3h, indicating that false context fear memories were enhanced in B-24h when the mice experienced B-3h. The difference in freezing levels between B-3h and B-24h in Hy-Crf-KD mice was larger than that in the control. In Hy-CRF-OE mice, the freezing level in B-3h was higher than that in control and Hy-Crf-KD mice, while the freezing level in B-24h was similar to that in B-3h. Locomotor activity before CS and freezing level during CS were similar among the groups. Therefore, we hypothesized that Hy-Crf-KD enhances the false context fear memory in a day, while Hy-CRF-OE potentiates the false fear memory within a few hours.
Reviewer 2 comment 2: “In general, I recommend authors to use more evidence to back their claims, especially in the Introduction of the article, which I believe is currently lacking. Thus, I recommend the authors to attempt to deepen the subject of their manuscript, as the bibliography is too concise: nonetheless, in my opinion, less than 60/70 articles for a research paper are insufficient. Indeed, currently, authors cite only 50 papers, and they are too low. Therefore, I suggest the authors to focus their efforts on researching more relevant literature: I believe that adding more studies and reviews will help them to provide better and more accurate background to this study.”
We acknowledge that our first manuscript lacked the background of our study and add many reference papers as the Reviewer 2 suggested. Please see our reply to the Reviewer 2 comment 3 what and where we changed the sentences as a background according to the suggestions of Reviewer 2 comment 3 in the Introduction section.
Reviewer 2 comment 3: “Introduction: The authors decided to focus specifically on animal models to discuss the role of hypothalamic CRF in the induction of false context fear memory and fear generalization, and, in my opinion, this is too limiting. Thus, I suggest reshaping the Introduction section, which seems not enough extensive and it does not seem to consider, in most cases, all the available studies in the literature that have acknowledged inhomogeneous and dispersive. I think that more information about the ability to use contextual information to modulate the expression of fear would provide a better background here. I suggest the authors to make such effort to provide a brief overview of the pertinent published literature that offer a perspective on altered brain circuits underlying aberrant fear learning process in PTSD, because as it stands, this information is not highlighted in the text. In this regard, I would recommend focusing on the role that the ventromedial prefrontal cortex and hippocampus network have in context-dependent fear learning, addressing how impairments in these brain regions affect fear memories: evidence from a recent theoretical review (https://doi.org/10.1038/s41380-021-01326-4) that focused on neurobiology of fear conditioning, analyzed the role of the ventromedial prefrontal cortex (vmPFC) was analyzed in the processing of safety-threat information and their relative value, and how this region is fundamental for the evaluation and representation of stimulus-outcome’s value needed to produce sustained physiological responses. Also, I believe that a recent yet relevant perspective manuscript (https://doi.org/10.17219/acem/146756) might be of interest: here the focus was on providing a deeper understanding of human learning neural networks, particularly on human PFC crucial role, that might also contribute to the advancement of alternative, more precise and individualized treatments for psychiatric disorders.”
We acknowledge that our first manuscript lacked the background of our study as Reviewer 2 suggested. Therefore, we added many reference papers as Reviewer 2 suggested and extensively improved the Introduction section as follow:
Introduction (page 4, line 1-page 5, line 3):
Cortisol modulates various learning and memory processes depending on the particular timing of cortisol increases relative to encoding, consolidation and retrieval [10]. Long-term dysregulation of cortisol systems after activation of the HPA-axis has been suggested to have lasting effects on vulnerable areas of the hippocampus, amygdala, and medial prefrontal cortex [11]. The acquisition of fear-associative memory requires various HPA-axis-related brain processes of coordinated neural activity within the amygdala [10,12,13], prefrontal cortex (PFC) [14,15], and hippocampus [14-17]. The amygdala plays a key role in the acquisition of fear learning, while the PFC and hippocampus are two other crucial neural structures that contribute to this process, collectively representing the neural network of fear conditioning [18]. The ventral part of the medial PFC may also play a major role in fear conditioning [19-21]. Subsequent traumatic events within a few hours raise the corticotropin-releasing factor (CRF) and the activation of the HPA-axis which might affect to the memory consolidation. However, how the stress-induced activation of HPA-axis, contributes to the formation of false context fear memories remains unclear.
References:
- Merz, C.J.; Hermann, A.; Stark, R.; Wolf, O.T. Cortisol modifies extinction learning of recently acquired fear in men. Social cognitive and affective neuroscience 2014, 9, 1426-1434, doi:10.1093/scan/nst137.
- Bremner, J.D. Traumatic stress: effects on the brain. Dialogues in clinical neuroscience 2006, 8, 445-461, doi:10.31887/DCNS.2006.8.4/jbremner.
- Skórzewska, A.; Lehner, M.; WisÅ‚owska-Stanek, A.; TurzyÅ„ska, D.; Sobolewska, A.; KrzÄ…Å›cik, P.; Szyndler, J.; Maciejak, P.; Chmielewska, N.; KoÅ‚osowska, K.; et al. Individual susceptibility or resistance to posttraumatic stress disorder-like behaviours. Behavioural brain research 2020, 386, 112591, doi:10.1016/j.bbr.2020.112591.
- Li, G.; Wang, G.; Shi, J.; Xie, X.; Fei, N.; Chen, L.; Liu, N.; Yang, M.; Pan, J.; Huang, W.; et al. trans-Resveratrol ameliorates anxiety-like behaviors and fear memory deficits in a rat model of post-traumatic stress disorder. Neuropharmacology 2018, 133, 181-188, doi:10.1016/j.neuropharm.2017.12.035.
- Perrine, S.A.; Eagle, A.L.; George, S.A.; Mulo, K.; Kohler, R.J.; Gerard, J.; Harutyunyan, A.; Hool, S.M.; Susick, L.L.; Schneider, B.L.; et al. Severe, multimodal stress exposure induces PTSD-like characteristics in a mouse model of single prolonged stress. Behavioural brain research 2016, 303, 228-237, doi:10.1016/j.bbr.2016.01.056.
- Ribeiro, T.O.; Bueno-de-Camargo, L.M.; Waltrick, A.P.F.; de Oliveira, A.R.; Brandão, M.L.; Munhoz, C.D.; Zanoveli, J.M. Activation of mineralocorticoid receptors facilitate the acquisition of fear memory extinction and impair the generalization of fear memory in diabetic animals. Psychopharmacology 2020, 237, 529-542, doi:10.1007/s00213-019-05388-9.
- Sarabdjitsingh, R.A.; Zhou, M.; Yau, J.L.; Webster, S.P.; Walker, B.R.; Seckl, J.R.; Joëls, M.; Krugers, H.J. Inhibiting 11β-hydroxysteroid dehydrogenase type 1 prevents stress effects on hippocampal synaptic plasticity and impairs contextual fear conditioning. Neuropharmacology 2014, 81, 231-236, doi:10.1016/j.neuropharm.2014.01.042.
- Merz, C.J.; Hamacher-Dang, T.C.; Stark, R.; Wolf, O.T.; Hermann, A. Neural Underpinnings of Cortisol Effects on Fear Extinction. Neuropsychopharmacology : official publication of the American College of Neuropsychopharmacology 2018, 43, 384-392, doi:10.1038/npp.2017.227.
- Battaglia, S. Neurobiological advances of learned fear in humans. Advances in clinical and experimental medicine : official organ Wroclaw Medical University 2022, 31, 217-221, doi:10.17219/acem/146756.
- Battaglia, S.; Garofalo, S.; di Pellegrino, G.; Starita, F. Revaluing the Role of vmPFC in the Acquisition of Pavlovian Threat Conditioning in Humans. The Journal of neuroscience : the official journal of the Society for Neuroscience 2020, 40, 8491-8500, doi:10.1523/jneurosci.0304-20.2020.
- Fullana, M.A.; Harrison, B.J.; Soriano-Mas, C.; Vervliet, B.; Cardoner, N.; Àvila-Parcet, A.; Radua, J. Neural signatures of human fear conditioning: an updated and extended meta-analysis of fMRI studies. Molecular psychiatry 2016, 21, 500-508, doi:10.1038/mp.2015.88.
- Battaglia, S.; Harrison, B.J.; Fullana, M.A. Does the human ventromedial prefrontal cortex support fear learning, fear extinction or both? A commentary on subregional contributions. Molecular psychiatry 2022, 27, 784-786, doi:10.1038/s41380-021-01326-4.
Reviewer 2 comment 4: “Introduction: Following the first point raised, I would also recommend another recent opinion manuscript which provided novel ‘functional interplay between central and autonomic nervous systems in human fear conditioning’, and highlights the crucial role of the prefrontal cortex. Finally, authors also might to consider some studies that have focused on this topic (https://doi.org/10.3390/biomedicines10010076; https://doi.org/10.3390/biomedicines9050517).”
We acknowledge that our first manuscript lacked the background of HPA-axis activity related symptom. Therefore, we added the following sentence in the Introduction section as Reviewer 2 suggested;
Introduction (page 4, line 1-page 5, line 3):
Moreover, HPA-axis activity links the varius symptom regarding the autonomic function, anxiety, locomotor activity or cognitive functions [23].
References:
- Tanaka, M.; Vécsei, L. Editorial of Special Issue "Crosstalk between Depression, Anxiety, and Dementia: Comorbidity in Behavioral Neurology and Neuropsychiatry". Biomedicines 2021, 9, doi:10.3390/biomedicines9050517.
Reviewer 2 comment 5: “Behavioral tests: Could the authors provide the specific number of mice that were used in the experiments?”
We acknowledge that our first manuscript lacked the total number of mice were used in the Methods section as follow: The number of mice we used in each group of each experiment was shown in Figure legends. Also, as Reviewer suggested, we put two summary Table1 and 2 in the revised version.
Methods (page 4, line 1-page 5, line 3):
Total 15 mice were used for measuring the plasma corticosterone concentration (5 mice for each group). For the behavioral test, in non-AAV infected mouse test (Figure 2 and Figure S2), 54 mice (24 mice for Figure 2 and 30 mice for Figure 1S) and mice were used and AAV infected mouse test (Figure 4), 57 mice were used for behavioral tests. Total 126 mice were used in the present study. In the test using a group of A-B (CS-) mice, A-B (CS+) mice, ABB (CS-) mice, ABB (CS+) mice in Figure 2 were performed at the same time respectively. Also, all of AAV injected mice with CS group (Hy-Control (CS+), Hy-Crf-KD (CS+) and Hy-CRF-OE (CS+)) were tested at the same time and all of AAV injected mice without the CS group (Hy-Control (CS-), Hy-Crf-KD (CS-) and Hy-CRF-OE (CS-)) were tested at the same time. Every day, 3 to 4 mice were tested and repeated these tests until the number of mice were enough to analyzing.
Reviewer 2 comment 6: “Contextual fear conditioning test: This paragraph that explains how mice were fear conditioned is the most important part of the study and should clearly describe all the experimental sessions in detail; therefore, this section might be improved by including further explanations, allowing the effective communication of experimental procedures.”
We acknowledge that our first manuscript lacked the detail of the method of conditioning test in the present study. Because we described some sentences about the conditioning test in the first paragraph of our first submission in the “Contextual fear conditioning” which also describing the apparatus and it might make a confusion for reader. Therefore, we separate the part of “Apparatus for the contextual fear conditioning test” and “Contextual fear conditioning” in the Methods section as follow:
Methods (page 4, line 1-page 5, line 3):
The mice were handled for 1 min for three consecutive days before the behavioral test. All mice were habituated to Box A 60 min before Context A for 15 min. Mice were placed in Box A again for 3 min, and then electrical shocks (1.0 mA for 2 s was delivered three times at 100-s intervals) were delivered 3 times at 100-s intervals as a fear CS (Context A) in Box A 3 min after the mice were placed on the grid floor, designated as Context A. All mice were returned to their home cage until the next context after Context A for fear conditioning in Box A.
Methods (page 4, line 1-page 5, line 3):
In all test in the present study, the chambers were thoroughly cleaned with a 70% ethanol solution between mice every times.
Reviewer 2 comment 7: “Results: In my opinion, this section is well organized, but it illustrates findings in an excessively broad way, without really providing full statistical details, to ensure in-depth understanding and replicability of the findings. Also, in my opinion, it is necessary for the authors to present their findings using summary tables.”
We acknowledge that the statistics were not sufficient for describing all results in our first manuscript. As Reviewer 2 suggested, we put two summary Table 1 and 2 for detail information about the results of Figure 2 and 4 including statistics in our revised version of our manuscript. Please see the two tables in the template manuscript.
Reviewer 2 comment 8: “Discussion: In this final section, the authors described the results and their argumentation and captured the state of the art well; however, I would have liked to see some views on a way forward. I believe that the authors should make an effort, trying to explain the theoretical implication as well as the translational application of this research article, to adequately convey what they believe is the take-home message of their study. Discussion of theoretical and methodological avenues in need of refinement is necessary, as well as suggestions of a path forward in understanding the role of hypothalamic CRF on the induction of false context fear memory and fear generalization. In this regard, recent evidence suggests that the application of new methods in Neuroinflammatory disorders’ treatment, such as the Non-invasive brain stimulation techniques (NIBS), have shown promising results in humans (https://doi.org/10.1038/s41398-020-0851-5). Importantly, I recommend referring to recent studies that revealed that the application of NIBS induces long-lasting effects, noninvasively modulating the cortical excitability, and modulates a variety of cognitive functions altered in patients that suffer from PTSD: for example, a recent review (https://doi.org/10.1016/j.neubiorev.2021.04.036) on the potential and effectiveness of non-invasive brain stimulation (NIBS) to interfere and modulate the abnormal activity of neural circuits (i.e., amygdala-mPFC-hippocampus) involved in the acquisition and consolidation of fear memories, which are altered in many mood psychiatric disorders (i.e., anxiety disorder, specific phobias, post-traumatic stress disorder or depression), would be of interest. Accordingly, another recent manuscript (https://doi.org/10.1016/j.jad.2021.02.076) focused on the same topic, illustrating the therapeutic potential of NIBS as a valid alternative in the treatment of untypically persistent memories that characterized those patients that do not respond to psychotherapy and/or drug treatments. In addition to the previously mentioned literature, authors might also see these additional studies that have focused on the efficacy of NIBS and IBS (https://doi.org/10.3389/fpsyt.2018.00201; https://doi.org/10.3389/fnagi.2020.578339).”
We appreciate the Reviewer 2 suggestion to discuss the important background and possibility raised from our results in the present study. As related to the Reviewer 2 comment 2 and 3, we discussed about the amygdala-mPFC-hippocampus in Discussion section as follow:
Discussion (page 4, line 1-page 5, line 3):
Functional alterations of the neural network underlying fear conditioning might contribute to the etiology of fear-related psychiatric disease, including PTSD [18]. Fear associative memory acquisition of fear learning requires coordinated neural activity within the amygdala, prefrontal cortex (PFC), and hippocampus [17-21,48,49]. In addition, cortisol exerted a critical impact on the amygdala–hippocampus–ventromedial PFC network underlying fear and extinction memories [17]. Dysregulation of negative feedback by cortisol suppressing the CRF release after the long-term activation of activation of the HPA-axis affects the hippocampus, amygdala, and medial PFC [11]. Inactivation of prefrontal inputs into the nucleus reuniens or direct silencing of nucleus reuniens projections enhances fear memory generalization [27]. Xu and Südhof demonstrated the generalization of memory attributes for a particular context by processing information from the medial PFC enroute to the hippocampus within a day after conditional training [27]. Decreased connectivity between the amygdala and medial PFC were shown to be related to memory intrusion and the re-experiencing of traumatic events [50]. Interestingly, repetitive transcranial magnetic stimulation of the right dorsolateral PFC appears to have a positive effect of reducing core symptoms in patients with PTSD [50]. Therefore, it is crucially important to investigate the role of CRF and cortisol on the neuronal networks including the amygdala–hippocampus–ventromedial PFC for their contribution to both the acquisition of fear memories and the consolidation of imprecise fear memories, although there remains a limitation for elucidating the neuronal mechanisms using the rodent model, because humans and animals have dissimilar functional PFC neuroarchitecture.
References:
- Bremner, J.D. Traumatic stress: effects on the brain. Dialogues in clinical neuroscience 2006, 8, 445-461, doi:10.31887/DCNS.2006.8.4/jbremner.
- Merz, C.J.; Hamacher-Dang, T.C.; Stark, R.; Wolf, O.T.; Hermann, A. Neural Underpinnings of Cortisol Effects on Fear Extinction. Neuropsychopharmacology : official publication of the American College of Neuropsychopharmacology 2018, 43, 384-392, doi:10.1038/npp.2017.227.
- Battaglia, S. Neurobiological advances of learned fear in humans. Advances in clinical and experimental medicine : official organ Wroclaw Medical University 2022, 31, 217-221, doi:10.17219/acem/146756.
- Battaglia, S.; Garofalo, S.; di Pellegrino, G.; Starita, F. Revaluing the Role of vmPFC in the Acquisition of Pavlovian Threat Conditioning in Humans. The Journal of neuroscience : the official journal of the Society for Neuroscience 2020, 40, 8491-8500, doi:10.1523/jneurosci.0304-20.2020.
- Fullana, M.A.; Harrison, B.J.; Soriano-Mas, C.; Vervliet, B.; Cardoner, N.; Àvila-Parcet, A.; Radua, J. Neural signatures of human fear conditioning: an updated and extended meta-analysis of fMRI studies. Molecular psychiatry 2016, 21, 500-508, doi:10.1038/mp.2015.88.
- Battaglia, S.; Harrison, B.J.; Fullana, M.A. Does the human ventromedial prefrontal cortex support fear learning, fear extinction or both? A commentary on subregional contributions. Molecular psychiatry 2022, 27, 784-786, doi:10.1038/s41380-021-01326-4.
- Xu, W.; Südhof, T.C. A neural circuit for memory specificity and generalization. Science (New York, N.Y.) 2013, 339, 1290-1295, doi:10.1126/science.1229534.
- Borgomaneri, S.; Battaglia, S.; Sciamanna, G.; Tortora, F.; Laricchiuta, D. Memories are not written in stone: Re-writing fear memories by means of non-invasive brain stimulation and optogenetic manipulations. Neuroscience and biobehavioral reviews 2021, 127, 334-352, doi:10.1016/j.neubiorev.2021.04.036.
- Stark, R.; Wolf, O.T.; Tabbert, K.; Kagerer, S.; Zimmermann, M.; Kirsch, P.; Schienle, A.; Vaitl, D. Influence of the stress hormone cortisol on fear conditioning in humans: evidence for sex differences in the response of the prefrontal cortex. NeuroImage 2006, 32, 1290-1298, doi:10.1016/j.neuroimage.2006.05.046.
- Kan, R.L.D.; Zhang, B.B.B.; Zhang, J.J.Q.; Kranz, G.S. Non-invasive brain stimulation for posttraumatic stress disorder: a systematic review and meta-analysis. Translational psychiatry 2020, 10, 168, doi:10.1038/s41398-020-0851-5.
Reviewer 2 comment 9: “Even though it is not mandatory, I believe that the ‘Conclusions’ section would be useful to adequately indicate convey what the authors believe is the take-home message of their study, and therefore provide a synthesis of the data presented in the paper as well as possible keys to advancing research and understanding of the prevalence of depression in post-stroke patients.”
We acknowledge that the Conclusion section should be provided for the reader of our study. Therefore, we put the “Conclusion” section after the Discussion as follow:
Conclusion (page 4, line 1-page 5, line 3):
In the present study, we found that the exposure to novel but similar contexts with the environment of traumatic fear experience within a few hours forms a false fear memory that is contextually enhanced within a day after the traumatic fear event. These results suggest that exposure to a novel context within a few hours after a traumatic fear event might interfere in the consolidation of precise fear memories of traumatic events and enhance false context fear memory within 24 h. Knockdown of hypothalamic Crf increases the freezing level and potentiates the false context fear memory at 24 h after fear conditioning. Meanwhile, overexpression of hypothalamic CRF enhances the onset of false fear memory formation within a few hours after a traumatic fear event. Cortisol has been suggested to exert a critical impact on the amygdala–hippocampus–ventromedial PFC network underlying the acquisition of fear memory, consolidating precise fear memories and fear generalization. Therefore, it is important that how the HPA-axis contributes to the formation and the potentiation of false context fear memories and to explore the clinical implications such as the treatment of psychiatric diseases such as PTSD and advacements in CRF-related cognitive impairment such as a depression in post-stroke [51] in the future.
Reference:
- Barra de la Tremblaye, P.; Plamondon, H. Alterations in the corticotropin-releasing hormone (CRH) neurocircuitry: Insights into post stroke functional impairments. Frontiers in neuroendocrinology 2016, 42, 53-75, doi:10.1016/j.yfrne.2016.07.001.
Reviewer 2 comment 10: “In according to the previous comment, I would ask the authors to better define a proper ‘Limitations and future directions’ section before the end of the manuscript, in which authors can describe in detail and report all the technical issues brought to the surface.”
We acknowledge that the limitation of our study is important because all of our results were obtained from the rodents. As Reviewer 2 suggested, how our results develop and reflect to the human disease in the future is important for the readers. Although we did not separate the as “Limitation” from the Discussion section, we added the sentences in the Discussion section as follow:
Discussion (page 4, line 1-page 5, line 3):
Therefore, it is crucially important to investigate the role of CRF and cortisol on the neuronal networks including the amygdala–hippocampus–ventromedial PFC for their contribution to both the acquisition of fear memories and the consolidation of imprecise fear memories, although there remains a limitation for elucidating the neuronal mechanisms using the rodent model, because humans and animals have dissimilar functional PFC neuroarchitecture.
Reviewer 2 comment 11: “References: Authors should consider revising the bibliography, as there are several incorrect citations. Indeed, according to the Journal’s guidelines, they should provide the abbreviated journal name in italics, the year of publication in bold, the volume number in italics for all the references.”
We thank the Reviewer 2 suggestions. We corrected the style of References. Please see the Reference section in the template manuscript.
Responses to Reviewer 1:
Reviewer 1 comment 1: “~~ However, we have some major problems with both the interpretation of the results and the fitting with previous conceptions of, for instance, what fear generalization actually is. This, together with a rather poor English grammar, makes the paper difficult to read and be put in context with previous work.”
We apologize that our poor English language makes difficult for understanding our works in the first submitted manuscript. We acknowledge that the English in our manuscript should be improved and we asked our revised version of manuscript to the Wordvice English Editing Service and MDPA English Editing Service.
Reviewer 1 comment 2: “One major concern is the concept of “generalization”. Although the authors dedicate a big part of the introduction to present these terms, the difference between “false memory” and “generalization” (and in the discussion, “memory confusion”) is not clear. Is it a matter of time? Amount of detected freezing? Are there different neurobiological mechanisms? Basically, the authors make interpretations in the discussion (page10 of 17, line 312) that previously they have assumed in their entire manuscript, and only when reading the discussion one understands what they refer to with generalization and false memory. They acknowledge the definition of generalization provided by others (Wiltgen and Silva), but disregarding that this definition is applied to memories more remote than 24h. This is quite confusing, and makes the first reading of the paper complicated for someone used to the general definition of generalization.”
We acknowledge that our interpretation against the word “Fear generalization” of first submitted manuscript was not adequate for our results according to general definition of published many papers. Particularly, the paper by Wiltgen an Silva, and Cai et al 2016 were very helpful to understand the definition of “generalization”. We strongly agreed the Reviewer 1 suggestion and the interpretation of the word “generalization” in our results of first submitted manuscript was changed to the “False fear memory” as follow:
Title (page 1, line 2-4):
Formation of false context fear memory is regulated by hypothalamic corticotropin-releasing factor in mice
Abstract (page 12, line ):
Traumatic events frequently produce false fear memories.
Abstract (page 12, line ):
The within-group freezing level in B-24h was higher than that in B-3h, indicating that false context fear memories were enhanced in B-24h when the mice experienced B-3h.
Abstract (page 12, line ):
Therefore, we hypothesized that Hy-Crf-KD enhances the false context fear memory in a day, while Hy-CRF-OE potentiates the false fear memory within a few hours.
Keywords (page 12, line ):
Traumatic stress, false context fear memory; hypothalamic corticotropin-releasing factor, Adeno-associated virus, shRNA
Introduction (page 4, line 1-page 5, line 3):
Indeed, the increase in generalization was due to a loss of detailed information about the context and not fear incubation [7]. However, in contrast to false context fear memory, inducing “fear generalization” across environments requires at least a week after exposure to traumatic events in rodents and it has been demonstrated that one month is required to induce freezing in novel environments [8]. According to the hypothesis that a shared neural ensemble linking distinct memories that encode close in time due to a temporary increase in neuronal excitability, a subsequent memory to the neuronal ensemble encodes the first memory. In rodents, it has been demonstrated that the first memory can strengthen the second memory within a day but not across a week [9]. Although, a false context fear memory of a traumatic event would be formed and subsequently lead to the formation of fear generalization, the considerable neuronal mechanism of fear generalization might not be the same as the concept of neuronal ensemble encoding the first memory that can lead to false fear memories.
Introduction (page 4, line 1-page 5, line 3):
The present study endeavored to understand the formation mechanism of false fear memories after traumatic stress events based on the concept of close in time encoding due to a temporary increase in neuronal excitability of subsequent memories to the neuronal ensemble after encoding an initial memory. Therefore, the present study focuses on the cause of spatial memory failure by source confusion after inducing false fear memory using two different contexts within three hours after conditional stimulation (SC). Then, we investigated the role of hypothalamic CRF on the induction of false context fear memories using mice with hypothalamic Crf knockdown (Hy-Crf-KD) and hypothalamic CRF overexpression (Hy-CRF-OE), respectively. Adeno-associated virus (AAV)-mediated mouse Crf small hairpin RNA (shRNA) was used to knock down Crf expression, and an AAV carrying mouse Crf cDNA was constructed for the overexpression of CRF.
Reviewer 1 comment 3: “Importantly, the authors do not cite key work related to their experiments and addressing the neurobiological mechanisms underlying memory linking (probably the equivalent to their “false memory”) such as Cai et al., 2016. In fact, most of what is discussed in the manuscript can also be interpreted under the light of the two memories (context A and B) being linked, rather than “confused” or “difficult to discriminate” as the authors suggest.
Having said this, my biggest suggestion is that the authors come with a different name for what they refer to with “generalization”, that then can be for the first time linked to false memory formation and CRF levels. This could improve the paper.”
As we mentioned above in our reply for Reviewer 1 comment 2, we acknowledge that our interpretation against the word “Fear generalization” of first submitted manuscript was not adequate for our results according to general definition of published many papers. We carefully read the paper by Cai et al 2016 many times and we agreed that the concept of the overlap between the neuronal ensembles representing two separate contextual memories within a day and over a week. Therefore, we extensively revised our claim throughout the manuscript as follow:
Introduction (page 4, line 1-page 5, line 3):
Indeed, the increase in generalization was due to a loss of detailed information about the context and not fear incubation [7]. However, in contrast to false context fear memory, inducing “fear generalization” across environments requires at least a week after exposure to traumatic events in rodents and it has been demonstrated that one month is required to induce freezing in novel environments [8]. According to the hypothesis that a shared neural ensemble linking distinct memories that encode close in time due to a temporary increase in neuronal excitability, a subsequent memory to the neuronal ensemble encodes the first memory. In rodents, it has been demonstrated that the first memory can strengthen the second memory within a day but not across a week [9]. Although, a false context fear memory of a traumatic event would be formed and subsequently lead to the formation of fear generalization, the considerable neuronal mechanism of fear generalization might not be the same as the concept of neuronal ensemble encoding the first memory that can lead to false fear memories.
Introduction (page 4, line 1-page 5, line 3):
The present study endeavored to understand the formation mechanism of false fear memories after traumatic stress events based on the concept of close in time encoding due to a temporary increase in neuronal excitability of subsequent memories to the neuronal ensemble after encoding an initial memory. Therefore, the present study focuses on the cause of spatial memory failure by source confusion after inducing false fear memory using two different contexts within three hours after conditional stimulation (SC). Then, we investigated the role of hypothalamic CRF on the induction of false context fear memories using mice with hypothalamic Crf knockdown (Hy-Crf-KD) and hypothalamic CRF overexpression (Hy-CRF-OE), respectively. Adeno-associated virus (AAV)-mediated mouse Crf small hairpin RNA (shRNA) was used to knock down Crf expression, and an AAV carrying mouse Crf cDNA was constructed for the overexpression of CRF.
Reviewer 1 comment 4: “In the results section, sub-titles should reflect statements related to the most important finding (“manipulation A results in B”) rather than a generic sentence.”
We acknowledge that the subtitles did not including our important finding. We have changed the subtitles in the Results section as follow:
Results (page 4, line 1-page 5, line 3):
Exposure of novel context in 3 h after fear conditioning formed false fear memory that was further enhanced at 24 hours after conditioning.
Results (page 4, line 1-page 5, line 3):
Hy-Crf-KD enhanced false fear memory level in 24 h and Hy-CRF-OE potentiate the false fear memory level in 3 h after fear conditioning.
Reviewer 1 comment 5: “Fig 1. The two contexts have the same shape. How do the authors think their results would look if the two contexts were more dissimilar, e.g., with a different shape?”
We acknowledge that we should discussed the possibility whether the mice were exposed to dissimilar chamber as a novel box represents the false fear memory. Therefore, we cite the adequate reference that dissimilar box after the conditioning did not lead the freezing behaviors in 3 hours after the conditioning in Discussion section as follow:
Discussion (page 4, line 1-page 5, line 3):
It has been demonstrated that the mice were habituated to exposed to a novel and highly dissimilar context with conditioning chamber exhibited a low level of freezing in dissimilar chamber on day 0 [25]. Our model of induction of false memory, we did not use the cue for conditioning. Therefore, false fear memory might not be able to form 3 h after the CS in the present study.
Reference:
[25] Fujinaka, A.; Li, R.; Hayashi, M.; Kumar, D.; Changarathil, G.; Naito, K.; Miki, K.; Nishiyama, T.; Lazarus, M.; Sakurai, T.; et al. Effect of context exposure after fear learning on memory generalization in mice. Mol Brain 2016, 9, 2, doi:10.1186/s13041-015-0184-0.
Reviewer 1 comment 6: “Fig 2. In our opinion, the false memory vs. generalization dichotomy brings problems in fig 2: Here, the authors say that “false fear memory (in context B) was induced 24h after the CS” and, below, “formed within 3h after CS” (page4, line112), after comparing freezing in A-B 24h vs. A-B 3h. We understand from this is to suggest that elevated levels of freezing both at 3h and 24h after conditioning are interpreted by the authors as similar false memory. However, 3h is similar to the time required for memory linking (Cai et al., 2016), whereas 24h could be, although still too short, more widely associated with the idea of “generalization”. In the concluding sentence, they state that, on the one hand, false memory was induced within 3h and, on the other hand, that generalization was produced if the mice had experienced the intermediate exposure to context B. Looking at the data, it is clear that “generalization” is produced regardless of the intermediate exposure (as evidenced by increased freezing in A-B 24h). What remains interesting is the fact that prompt formation of false memory (3h) is necessary for having increased freezing levels at 24h compared to those without 3h exposure to context B. This is probably the most interesting observation, that in our opinion becomes diluted in the false memory vs. generalization terminology/conceptual discussion. Calling this something different, such as “false memory enhancement” would eliminate this problem.
As related suggestion by Reviewer 1 in the comment 2 and 3, we have performed extensive revised throughout the manuscript, Title, Abstract, Introduction, Results, Discussion and Conclusion as follow:
Introduction (page 4, line 1-page 5, line 3):
Indeed, the increase in generalization was due to a loss of detailed information about the context and not fear incubation [7]. However, in contrast to false context fear memory, inducing “fear generalization” across environments requires at least a week after exposure to traumatic events in rodents and it has been demonstrated that one month is required to induce freezing in novel environments [8]. According to the hypothesis that a shared neural ensemble linking distinct memories that encode close in time due to a temporary increase in neuronal excitability, a subsequent memory to the neuronal ensemble encodes the first memory. In rodents, it has been demonstrated that the first memory can strengthen the second memory within a day but not across a week [9]. Although, a false context fear memory of a traumatic event would be formed and subsequently lead to the formation of fear generalization, the considerable neuronal mechanism of fear generalization might not be the same as the concept of neuronal ensemble encoding the first memory that can lead to false fear memories.
Introduction (page 4, line 1-page 5, line 3):
The present study endeavored to understand the formation mechanism of false fear memories after traumatic stress events based on the concept of close in time encoding due to a temporary increase in neuronal excitability of subsequent memories to the neuronal ensemble after encoding an initial memory. Therefore, the present study focuses on the cause of spatial memory failure by source confusion after inducing false fear memory using two different contexts within three hours after conditional stimulation (SC). Then, we investigated the role of hypothalamic CRF on the induction of false context fear memories using mice with hypothalamic Crf knockdown (Hy-Crf-KD) and hypothalamic CRF overexpression (Hy-CRF-OE), respectively. Adeno-associated virus (AAV)-mediated mouse Crf small hairpin RNA (shRNA) was used to knock down Crf expression, and an AAV carrying mouse Crf cDNA was constructed for the overexpression of CRF.
Reviewer 1 comment 7: “In comparing freezing after 24h in context A in A-A vs. A-B-A mice (fig. suppl. 1C vs. D), it seems like exposure to context B 3h after conditioning weakened memory expression in the context A. First, the experiment ABA comes a little bit out of the blue and should be more properly introduced. Second, this observation might be linked to the same concept as before, in the sense that intermediate exposure to different contexts affect memory expression 24h later, which is very interesting. How do the authors interpret this?”
We acknowledge that the results of freezing level 24 h in ABA mice looked decreased, compared to that in A-A mice. However, we did not these two experiments were not performed at the same time. Therefore, we cannot represent these graphs as one graph and perform the statistical analysis. We discussed the possibility that the shorter freezing level 24 h in ABA mice than that in A-A mice in Discussion section as follow:
Discussion (page 4, line 1-page 5, line 3):
In the present study, exposing mice to the novel box during Context B in Box B 3 h after the CS during Context A in Box A, the freezing level in Box A 24 h after the CS in ABA mice seemed decreased compared to the result of freezing level in A-A mice that were not exposed to Box B at 3 h. However, we did not perform these experiments simultaneously (not the same batch), so we cannot statistically compare the difference in the freezing level between ABA and A-A mice. If the freezing level in Box A at 24 h for ABA mice was decreased compared to that of A-A mice, two hypotheses are raised. One is memory extinction and another is that experience of Box B, at 3 h interfered with the memory consolidation of Box A that delivered the electrical shocks. The widely used paradigm of memory extinction is that after the CS, the mice should be exposed to the same chamber several times without the CS. However, we used a novel box after the CS. However, the memory of Box A with the CS might become less precise after interfering with the perceptual memory of Box B at 3 h. This result is also considered by the hypothesis that the shared neural ensemble leads to a subsequent memory with the neuronal ensemble encoding of the first memory during a temporary increase in neuronal excitability in which two distinct memories are encoded the close together in time.
Reviewer 1 comment 8: “Fig 3. Subfigures A-D can be moved to supplementary materials. The entire maps of the viral vectors are not necessary, a schematic representation is enough. In B, do the authors have in vitro evidence that their overexpression constructs worked? In D, what does “Nat Neurosci” and the different code numbers refer to? In F, the distance indicated by the scale bars should be indicated in the figure legend. Furthermore, a DAPI stain would be an added plus, as well as a zoomed out image of the hypothalamus. Finally, the figure legend contains a panel “M?”
We acknowledge that“Nat Neurosci” and the different code numbers in the figure are not explained in legend of Figure 3. We changed the indication of each band of western blotting and explained in figure legend refer to the“Nat Neurosci” and the code numbers. In addition, we wrote the scale bar in the figure legend and delete “M”. As for counterstaining with DAPI for CRF and fluorescence tag (GFP or RFP), we do not have a laser for detecting the DAPI, so we tried to purchase the TO-PRO-3 which can be detected by Ex 642. Unfortunately, we have to import from the Sigma-Aldrich or Thermo Fisher Scientific in the USA, so we have to wait over three weeks to get this. Therefore, we cannot try to stain the nuclear in the hypothalamic sections by deadline during revision.
Subfigures A-D have moved to the Supplementary Figure S2 and changed the Figure 3. Also, we improved the Methods section and the legend of Figure 3 and separate it to the supplementary Figure S2 as follow:
Methods (page , line ):
The shRNA sequences to knockdown Crf were based on a previous report [55], the Sigma MISSION shRNA library by the RNAi Consortium (Boston, MA, USA); (#1) AGATTATCGGGAAATGAAA [55], TRCN0000414479 (#2) TTAGCTCAGCAAGCTCACAG, TRCN0000436997 (#3): ATCTCTCTGGATCTCACCTTC. We assessed the knockdown efficiencies of shRNA candidates by co-transfection of each shRNA carrying AAV plasmid and FLAG-CRF encoding pCAGGS plasmid into HEK293FT cells. CRF expression was analyzed by immunoblot with anti-FLAG-HRP antibody (#015-22391, FUJIFILM Wako Pure Chemical Corp., Tokyo, Japan) (Fig. S2C, S2D and Fig. S3).
Figure 3 (page , line ):
Fig. 3. (A) AAV-PHP.eB virus injection maps according to the Paxinos mouse brain atlas. Green bars around the center of the brain map indicate virus injection needles and the location of virus injections. (B) Confocal images in the transduction of mouse hypothalamus with AAV-PHP.eB vector expressing GFP, RFP and detecting the mouse CRF by immunofluorescence in control (aï€d), knockdown (eï€h), and overexpression (iï€l). Scale bar = 200 μm (×10 magnification) for a, e, I and 50 μm for b, c, d, f, g, h, j, k, l (×40 magnification).
Supplementary Figure S2 (Page , line ):
Fig. S2. Construction and characterization of adeno-associated virus (AAV) PHP.eB vectors for knockdown of mouse Crf by shRNA system and overexpression of mouse CRF. (A) Illustration of the AAV vectors expressing Crf shRNAs under the control of U6 promoter. The vectors express GFP as a reporter gene under the control of Chicken β-Actin promoter (CBA pro).(B) Illustration of the AAV vector carrying CRF*-FLAG-T2A-RFP. CRF* represents the shRNA#2 resistant version. (C) Western blotting for expressing CRF*-FLAG-T2A-RFP carrying AAV plasmid that was transfected to HEK293 cells. (D) The knockdown validation of the effective shRNA candidates. #2 shRNA was the most effective and used this shRNA in further experiments.
Reviewer 1 comment 9: “Fig 4. 4C is referenced in the text before 4B.”
We acknowledge that the describing the result of 4B should be before the 4C. Therefore, we have changed the sentences as follow:
Results (Page , line ):
Freezing level was similar in the three groups of ABB-control, ABB-KD, and ABB-OE mice during the three electric shocks in Box A as a conditioning electric shock (Fig. 4B). One-way ANOVA revealed that the AAV injections affected the locomotor activity in Box A for 3 min just before the CS was delivered (F(2, 35) = 4.787, p < 0.05, Fig. 4C). Fisher's LSD test indicated that the locomotor activity in Hy-Crf-KD mice was higher than that in control mice and Hy-CRF-OE mice (control: p < 0.01 vs. Hy-Crf-KD; Hy-CRF-OE: p < 0.05 vs. Hy-Crf-KD, p = 0.277 vs. control, Fig. 4C).
Reviewer 1 comment 10: “Fig 4. Variability of freezing is high among injected animals. Was an outlier test carried out? Were injection sites checked and animals with poor injection discarded from the test? Were all the animals conditioned and tested at the same time? Otherwise, inter-group comparisons of total freezing levels might be subject to batch effects that are difficult to account for. In any case, the difference in freezing levels between 3h and 24h is interesting and adequate in trying to normalize freezing per group even if this was obtained from different batches.”
As Reviewer 2 suggested to summarize the statistics, we represented all averages and SEMs in the Table 1 and 2 refer to the Figure 2 and 4. The range of SEM in 3 h after the conditioning in non-AAV injected mice were from 1.373 to 11.01%, while the range of SEM in 3 h after the conditioning in AAV-injected mice were from 0.394 to 5.058. Therefore, we do not think the variability of freezing is not higher among AAV injected mice than non-injected mice. Please see the Table 1 and 2.
Reviewer 1 comment 11: “Page 6, line 189. This was the first time, together with the conclusion of the first part of the results, we understood that the authors use “fear generalization” for the increase in freezing observed at 24h after the pre-exposure to context at 3h post-conditioning… This is problematic because this is only clearly stated in the discussion and reflects and interpretation of the authors, it is difficult to follow the paper if this is not defined before. Again, we would suggest that the authors come with a new name for the phenomenon they observe, to distinguish it from the more traditional idea of generalization. This might help the reader to not get distracted in terminology issues.”
As related suggestion by Reviewer 1 in the comment 2 and 3, we have performed extensive revised throughout the manuscript, Title, Abstract, Introduction, Results, Discussion and Conclusion as follow:
Introduction (page 4, line 1-page 5, line 3):
Indeed, the increase in generalization was due to a loss of detailed information about the context and not fear incubation [7]. However, in contrast to false context fear memory, inducing “fear generalization” across environments requires at least a week after exposure to traumatic events in rodents and it has been demonstrated that one month is required to induce freezing in novel environments [8]. According to the hypothesis that a shared neural ensemble linking distinct memories that encode close in time due to a temporary increase in neuronal excitability, a subsequent memory to the neuronal ensemble encodes the first memory. In rodents, it has been demonstrated that the first memory can strengthen the second memory within a day but not across a week [9]. Although, a false context fear memory of a traumatic event would be formed and subsequently lead to the formation of fear generalization, the considerable neuronal mechanism of fear generalization might not be the same as the concept of neuronal ensemble encoding the first memory that can lead to false fear memories.
Introduction (page 4, line 1-page 5, line 3):
The present study endeavored to understand the formation mechanism of false fear memories after traumatic stress events based on the concept of close in time encoding due to a temporary increase in neuronal excitability of subsequent memories to the neuronal ensemble after encoding an initial memory. Therefore, the present study focuses on the cause of spatial memory failure by source confusion after inducing false fear memory using two different contexts within three hours after conditional stimulation (SC). Then, we investigated the role of hypothalamic CRF on the induction of false context fear memories using mice with hypothalamic Crf knockdown (Hy-Crf-KD) and hypothalamic CRF overexpression (Hy-CRF-OE), respectively. Adeno-associated virus (AAV)-mediated mouse Crf small hairpin RNA (shRNA) was used to knock down Crf expression, and an AAV carrying mouse Crf cDNA was constructed for the overexpression of CRF.
Reviewer 1 comment 12: “Fig 5. We appreciate the effort in trying to link anxiety leading to “false memory” to freezing the day after, but do not quite understand the logic behind this analysis and what additional information different from freezing this brings… it is obvious that the longer a mouse will spend freezing, the less distance it will move. Correlation of 3h distance vs. 24h freezing is lost in genetically manipulated mice, but correlation is not causation (statements as in page10 of 17, line 358 should be weakened). This only reinforces our opinion that it is extremely difficult to extract any conclusion about anxiety levels from assessments that are no de-coupled from fear conditioning. Assessing distance travelled in context A before CS, as done in fig4C, is the closest and most appropriate way.”
We acknowledge that this part is not adequate and awkward phrase. Also, we should not discuss the relationship from the result between the center preference in the box and the anxiety level. However, the findings the central preference in the Box B 3h and the enhancement of false fear memory in 24 h after the conditioning is very important. We are currently the possibility that the center preference is a kind of active coping against novel environment at 3 h and decrease the enhancement of false fear memory in 24 h. Therefore, we changed the sentences in Results and Discussion sections as follow:
Results (page 4, line 1-page 5, line 3):
Therefore, there is a possibility that the both of the facilitation of false fear memory formation and the potentiated false context fear memory level were due to the hypothalamic CRF expression level rather than adapting ability to the novel environment. In the next, we investigate whether the preference for the central zon in the Box B 3 h after the CS during the Context B affect to the false context fear memory level in 24 h. The freezing time in 24 h was negatively correlated with the distance traveled by control mice in the central zone in 3 h (control mice: Pearson’s correlation, t = -2.574, df = 11, p < 0.05, R2 = 0.376, Fig. 5C), indicating that exploratory as an active coping behavior into the central zone of a novel chamber during Context B 3 h might affect the freezing level in Context B 24 h.
Discussion (page 4, line 1-page 5, line 3):
Finally, in the present study, we found that the preference for the central zone in Box B at 3 h—as a novel environment after the CS—affected the freezing level during Context B 24 h in control mice. While hypothalamic CRF did not affect the preference for the central zone during Context B 3 h, the significant correlation between the preference for the central zone in a novel chamber (Box B 3 h) and the freezing level during Context B 24 h was abolished in Hy-Crf-KD and Hy-CRF-OE mice. Therefore, we also suggest that adequate active coping behavior when in a novel environment may decrease and subsequently induce false context fear memory formation.
Reviewer 1 comment 12: “- Line 30, evidence is not plural”
We thank your suggestion for our grammatical error in the manuscript. We corrected from “influence” to “influences”.
Reviewer 1 comment 13: “- Line 64: The statement “in patients with PTSD” is grossly misleading, as mice are being studied here.”
We acknowledge that the statement “in patients with PTSD” is not adequate in our mouse study. According to the correction of our focus in this study, we focus on the false fear memory, but not fear generalization. Therefore, we excluded this phrase “in patients with PTSD”.
Reviewer 1 comment 14: “- Line 143-147. This is not a proper sentence, grammatically speaking.”
We acknowledge that this part of our English was very poor. Therefore, we corrected this sentence in the Results section as follow:
Results (page 4, line 1-page 5, line 3):
To investigate the effects of HPA-axis activity on the formation of false fear memory using mice that were injected with AAV-PHP.eB-produced virus which can knockdown the Crf gene by the shRNA system and the overexpression of CRF in the hypothalamus (Fig. 3 and Fig. S2).

Round 2
Reviewer 1 Report
- We acknowledge that they do address the problem of generalisation, which was my main conceptual concern.
- An important problem is still with the English. I still have a hard time to understand many of the modifications they added or some of the replies to our comments. - It's surprising to hear that no blue filters are available for their confocal to visualize DAPI??? - If the authors now include the concept of conjunctive encoding and neuronal excitability, as stated in the introduction, they need to link it to stress or CRF at some point. This part should be removed. - Statements about correlation/causation related to time in the centre zone of a novel environment as a coping behaviour that affects freezing 24h later should still be weakened, this is highly speculative and only based on a correlation.Author Response
To the Editors       Fukushima, Japan
International Journal of Molecular Sciences May 29th, 2022
Subject: 2nd round of Revision for Manuscript ID: ijms-1718457
With this letter we would like to submit a 2nd round of revised version of the article entitled “Formation of false context fear memory is regulated by hypothalamic corticotropin-releasing factor in mice” by Kenjiro Seki submitted for publication to the International Journal of Molecular Sciences.
At first, we would like to thank the Reviewer 1 for their valuable critical comments and excellent suggestions, which have further contributed to the present improvement of the manuscript. Also, we apologize that our poor English and we blanked the both of page and line number in the 1st round of Response to Reviewers. We extensively and carefully corrected the sentences using the template of ijms and put the page and line number in this Response to Reviewer 1 including 1st round after the 2nd round of Response to Reviewer 1. We essentially followed the reviewer’s comments and arguments and address all points that had been raised. The manuscript was extensively changed accordingly. We have highlighted all the changes made in the original manuscript below in response to reviewer.
We apologize that our poor English in 1st round of revised version of our manuscript also makes difficult for understanding our works. We acknowledge that the English in our 1st round of revised version of manuscript should be further improved and then we asked the MDPI English Editing Service to correct and improve our 2nd round of revised version of our manuscript (Certification no. 44660).
We also apologize that we blanked the page and line number in our responses for your reference in the 1st round of “Response to Reviewers”. We did not put the page and line number in ijms template of tracking version. After the submission of 1st our revised version, we learned how we submit the tracking version in ijms template and how we put the page and line number when the template manuscript was used. We also recorrected the 1st round of our “Response to Reviewers” after this 2nd round of “Response to Reviewer”. Please see our 1st round of “Response to Reviewer”.
The Abstract, Introduction, Results, Discussion and Methods section were modified according to the reviewer comments/suggestions and additional changes to make reading easier and clearer. References are updated accordingly.
For reference, the reviewer’s comments are shown in quotations. Please note that the referring page and line number shown in each our response to the reviewer comment corresponds to the line number of each page in the untracked-changed version in ijms’s template of revised manuscript. When we showed the deleted parts in this 2nd round of revised version in the “Response to Reviewer”, the corresponding page and line number were shown in tracked-changed version in some cases.
Best Regards,
Kenjiro Seki, Ph.D.
Responses to Reviewer 1 (2nd round of revision):
Reviewer 1 comment 1: “- An important problem is still with the English. I still have a hard time to understand many of the modifications they added or some of the replies to our comments.”
We apologize that our poor English in 1st round of revised version of our manuscript also makes difficult for understanding our works. We acknowledge that the English in our 1st round of revised version of manuscript should be further improved and then we asked the MDPI English Editing Service to correct and improve our 2nd round of revised version of our manuscript (Certification no. 44660).
We also apologize that we blanked the page and line number in our responses for your reference in the 1st round of “Response to Reviewers”. We did not put the page and line number in ijms template of tracking version. After the submission of 1st our revised version, we learned how we submit the tracking version in ijms template and how we put the page and line number when the template manuscript was used. We also recorrected the 1st round of our “Response to Reviewers” after this 2nd round of “Response to Reviewer”. Please see our 1st round of “Response to Reviewer”.
Reviewer 1 comment 2: “- It's surprising to hear that no blue filters are available for their confocal to visualize DAPI???”
We thank the reviewer’s comment about our confocal images. Our confocal system, Zeiss LSM510, does not equip the UV laser. Thus, we unfortunately could not insert the nuclei images on the Figure 3 by visualizing with DAPI. Also, we mentioned in the previous “Response to Reviewer”, we tried to purchase the Topro-3 for detecting the nuclear in the cells by 633 nm excitation, however, there is no stock in all agencies in Japan where we are always asking and if we ordered the Topro-3 after the 1st round of reviewer’s suggestion, we had to wait at least 3 to 4 more weeks. I heard that the number of frights which bound for Japan are still restricted and decreased after the pandemic and currently the war between Russian and Ukraine effect the import reagents, it is very hard to get any reagent from the foreign countries.
However, instead, we carefully selected the hypothalamus area and indicated the typical part such as third ventricle (3V) and paraventricular nucleus of hypothalamus (PVN) in all images, and the dotted lines were shown for PVN. The images from i-l in Figure 3B for overexpression of CRF in the hypothalamus (AAV-CRF-T2A-RFP) were changed by other images because the 3V in the previous images were not clearly shown. Therefore, we appreciate the Reviewer 1 that we realized the importance of indication such as 3V and PVN in the images in Figure 3, although we could not show the images with DAPI.
Reviewer 1 comment 3: “- If the authors now include the concept of conjunctive encoding and neuronal excitability, as stated in the introduction, they need to link it to stress or CRF at some point. This part should be removed.”
We agree the suggestion by Reviewer 1. We should delete the sentences from last paragraph of Introduction section, because we did not any shown the link between the stress and CRF in the present study. Therefore, we corrected the sentences in the last paragraph in the Introduction as follow:
Introduction (Page 2, line 75 – 83 in untracked-changed version):
For this study, we endeavored to understand the mechanism of formation of false fear memories after traumatic stress events. In the present study, we focus on the cause of spatial memory failure due to source confusion in inducing false fear memory, using two different contexts within 3 hours after conditional stimulation (CS). Then, we investigated the role of hypothalamic CRF on the induction of false context fear memory using mice with hypothalamic Crf knockdown (Hy-Crf-KD) and hypothalamic CRF overexpression (Hy-CRF-OE). Adeno-associated virus (AAV)-mediated mouse Crf small hairpin RNA (shRNA) was used to knock down Crf expression, while an AAV carrying mouse Crf cDNA was constructed for the overexpression of CRF.
Reviewer 1 comment 4: “- Statements about correlation/causation related to time in the centre zone of a novel environment as a coping behaviour that affects freezing 24h later should still be weakened, this is highly speculative and only based on a correlation.”
We acknowledge that it was highly speculative and weakened to suggest that the preference of central zone during Box B at 3 h is a key point for potentiation of the freezing level in Box B at 24 h and the KD or OE of Hypothalamic CRF disrupt this correlation by enhancing and potentiating the false fear memory. We would carefully investigate and analyze this correlation in our future study. Therefore, we deleted the following paragraph from the Results and Discussion, and the statistic in Methods sections as follow:
Result (page 9, line 289 – 310 in tracked-changed version):
Discussion (page 16, line 547 – 557 in tracked-changed version):
Methods (page 21, line 773 – 775 in tracked-changed version):
Methods (page 21, line 778 in tracked-changed version):
.
Responses to Reviewer 1 (1st round of revision):
Reviewer 1 comment 1: “~~ However, we have some major problems with both the interpretation of the results and the fitting with previous conceptions of, for instance, what fear generalization actually is. This, together with a rather poor English grammar, makes the paper difficult to read and be put in context with previous work.”
We apologize that our poor English language makes difficult for understanding our works in the first submitted manuscript. We acknowledge that the English in our manuscript should be improved and we asked our revised version of manuscript to the Wordvice English Editing Service and MDPI English Editing Service.
Reviewer 1 comment 2: “One major concern is the concept of “generalization”. Although the authors dedicate a big part of the introduction to present these terms, the difference between “false memory” and “generalization” (and in the discussion, “memory confusion”) is not clear. Is it a matter of time? Amount of detected freezing? Are there different neurobiological mechanisms? Basically, the authors make interpretations in the discussion (page10 of 17, line 312) that previously they have assumed in their entire manuscript, and only when reading the discussion one understands what they refer to with generalization and false memory. They acknowledge the definition of generalization provided by others (Wiltgen and Silva), but disregarding that this definition is applied to memories more remote than 24h. This is quite confusing, and makes the first reading of the paper complicated for someone used to the general definition of generalization.”
We acknowledge that our interpretation against the word “Fear generalization” of first submitted manuscript was not adequate for our results according to general definition of published many papers. Particularly, the paper by Wiltgen an Silva, and Cai et al 2016 were very helpful to understand the definition of “generalization”. We strongly agreed the Reviewer 1 suggestion and the interpretation of the word “generalization” in our results of first submitted manuscript was changed to the “False fear memory” as follow:
Title (page 1, line 2 – 3 in untracked-changed version):
Formation of false context fear memory is regulated by hypo-thalamic corticotropin-releasing factor in mice
Abstract (page 1, line 11 in untracked-changed version):
Traumatic events frequently produce false fear memories.
Abstract (page 1, line 18 – 19 in untracked-changed version):
The within-group freezing level at B-24h was higher than that at B-3h, indicating that false context fear memory was enhanced at B-24h.
Abstract (page 1, line 23 – 24 in untracked-changed version):
Therefore, we hypothesized that Hy-Crf-KD potentiates the induction of false context fear memory, while Hy-CRF-OE enhances the onset of false fear memory formation.
Keywords (page 1, line 25 – 26 in untracked-changed version):
Traumatic stress, false context fear memory; hypothalamic corticotropin-releasing factor, Adeno-associated virus, shRNA
Introduction (page 1, line 37 – page 2. line 51 in untracked-changed version):
Indeed, this increase in generalization is due to a loss of detailed information about the context, and not fear incubation [7]. However, in contrast to false context fear memory, inducing of “fear generalization” across environments requires at least a week after exposure to a traumatic event in rodents, and it has been demonstrated that one month is required to induce freezing in novel environments [8]. According to the hypothesis that a shared neural ensemble linking distinct memories that encode close in time due to a temporary increase in neuronal excitability, a subsequent memory to the neuronal ensemble encodes the first memory. It has been hypothesized that a shared neural ensemble links distinct memories which are encoded within a close timeframe, due to a temporary increase in neuronal excitability. Accordingly, in rodents, it has been demonstrated that the first memory can strengthen a second memory within a day, but not across a week [9]. Although, a false context fear memory of the traumatic event may form and subsequently lead to fear generalization, the neuronal mechanism of fear generalization may differ from the concept of first memory neuronal ensemble encoding, which may lead to the false fear memories.
References:
[7] Anagnostaras, S.G.; Gale, G.D.; Fanselow, M.S. Hippocampus and contextual fear conditioning: recent controversies and advances. Hippocampus 2001, 11, 8-17, doi:10.1002/1098-1063(2001)11:1<8::aid-hipo1015>3.0.co;2-7.
[8] Wiltgen, B.J.; Silva, A.J. Memory for context becomes less specific with time. Learning & memory (Cold Spring Harbor, N.Y.) 2007, 14, 313-317, doi:10.1101/lm.430907.
[9] Cai, D.J.; Aharoni, D.; Shuman, T.; Shobe, J.; Biane, J.; Song, W.; Wei, B.; Veshkini, M.; La-Vu, M.; Lou, J.; et al. A shared neural ensemble links distinct contextual memories encoded close in time. Nature 2016, 534, 115-118, doi:10.1038/nature17955.
Introduction (page 2, line 75 – 83 in untracked-changed version):
For this study, we endeavored to understand the mechanism of formation of false fear memories after traumatic stress events. In the present study, we focus on the cause of spatial memory failure due to source confusion in inducing false fear memory, using two different contexts within 3 hours after conditional stimulation (CS). Then, we investigated the role of hypothalamic CRF on the induction of false context fear memory using mice with hypothalamic Crf knockdown (Hy-Crf-KD) and hypothalamic CRF overexpression (Hy-CRF-OE). Adeno-associated virus (AAV)-mediated mouse Crf small hairpin RNA (shRNA) was used to knock down Crf expression, while an AAV carrying mouse Crf cDNA was constructed for the overexpression of CRF.
Results (page 4, line 131 – 134 in untracked-changed version):
These results suggest that the false fear memory of Context A was induced within 3 h after the CS, and the false fear memory was potentiated in Box B at 24 h if the mice had experienced a similar environment to Context A with the CS after formation of the false fear memory.
Results (page 5, line 160 – 161 in tracked-changed version):
- 2. Hy-Crf-KD enhanced false fear memory level in 24 h and Hy-CRF-OE potentiate the false fear memory level in 3 h after fear conditioning.
Results (page 5, line 162 in tracked-changed version):
To investigate the effects of HPA axis activity on false fear memory,
Results (page 7, line 201 – 206 in tracked-changed version):
On the other hand, a paired t-test also indicated that the freezing levels of control and Hy-Crf-KD mice, but not Hy-CRF-OE mice, at 24 h after the CS were higher than those at 3 h (control, 24 h: t = 3.022, df = 12, p < 0.05 vs. 3 h; Hy-Crf-KD mice, 24 h: t = 4.467, df = 10, p < 0.01 vs. 3 h; Fig. 4D, Table 2), indicating that the false fear memory was po-tentiated in Box B at 24 h in control and Hy-Crf-KD mouse groups when the mice were experienced Box B 3 h after the CS.
Results (page 13, line 370 – 373 in tracked-changed version):
Therefore, we discussed the relationship between false context fear memory formation and hypothalamic CRF.
Discussion (page 10, line 277 – page 11, line 293 in untracked-changed version):
Fear generalization to innocuous stimuli acting as reminders of the trauma, even in safe place or environment, is one of the central problems and a hallmark of PTSD [29]. It has been further posited that the generalization will be enhanced if the new context is similar, but not completely different, from the training context, due to memory source confusion [30]. False memories can be spontaneously produced after an extreme fear experience but, classically, the term “false memory” applies to memory formed without actual experience of the event. If “fear generalization” is defined as the actual memory of a fear experience that transfer a conditioned response to stimuli that perceptually differ from the original conditioned stimulus [30], fear generalization may also be the same as false fear memory. In rodents, fear generalization has been shown to be induced when mice were exposed to contexts soon after training [26]. From these reports, the definition of “memory generalization” is never only with respect to a re-mote memory. In the present study, we applied a environment 3 h and 24 h after the CS, which the mice may not have been able to precisely discriminate from the box in which the conditioned was carried out. As such, this paradigm is not a test of the re-mote memories, but, instead, experiencing the novel box 3 h after the CS led to the source confusion of the traumatic event and potentiated the induction of false context fear memory.
References:
[26] Fujinaka, A.; Li, R.; Hayashi, M.; Kumar, D.; Changarathil, G.; Naito, K.; Miki, K.; Nishiyama, T.; Lazarus, M.; Sakurai, T.; et al. Effect of context exposure after fear learning on memory generalization in mice. Mol Brain 2016, 9, 2, doi:10.1186/s13041-015-0184-0.
[29] Jovanovic, T.; Norrholm, S.D.; Blanding, N.Q.; Davis, M.; Duncan, E.; Bradley, B.; Ressler, K.J. Impaired fear inhibition is a biomarker of PTSD but not depression. Depress Anxiety 2010, 27, 244-251, doi:10.1002/da.20663.
[30] Bergstrom, H.C. Assaying Fear Memory Discrimination and Generalization: Methods and Concepts. Curr Protoc Neurosci 2020, 91, e89, doi:10.1002/cpns.89.
Discussion (page 11, line 316 – 328 in untracked-changed version):
It has been found that, when separated by a week, independent populations of neurons encoded two distinct contexts; meanwhile, while the two contexts were separated only within a day, shared neuronal ensembles between the two contexts overlapped in the CA1 region of the hippocampus [9]. Neuronal excitability can lead to increases in memory strength, and neural ensemble sharing can strengthen the memory for a secondary context within 5 h [9]. Therefore, the neuronal mechanism of generalized re-mote fear memories in patients with PTSD is different from false memories produced within a day. In the present study, freezing behavior was observed in Context B 3 h after the conditioning, and the freezing response at 24 h was facilitated. Therefore, our observation that experiencing a novel environment enhanced false context fear memory 24 h after a traumatic fear event may be explained by the hypothesis of shared neuronal ensembles between Box A and Box B strengthening the false memory within 24 h.
Reference:
[9] Cai, D.J.; Aharoni, D.; Shuman, T.; Shobe, J.; Biane, J.; Song, W.; Wei, B.; Veshkini, M.; La-Vu, M.; Lou, J.; et al. A shared neural ensemble links distinct contextual memories encoded close in time. Nature 2016, 534, 115-118, doi:10.1038/nature17955.
Conclusion (page 13, line 389 – 394 in untracked-changed version):
In the present study, we found that exposure to a novel but similar contexts to the environment of traumatic fear experience within a few hours induced the formation of false fear memories, which were contextually enhanced within a day after the traumatic fear event. These results suggest that exposure to a novel context within a few hours after a traumatic fear event may interfere with the consolidation of precise fear memory regarding a traumatic event, thus enhancing false context fear memory formation within 24 h.
Reviewer 1 comment 3: “Importantly, the authors do not cite key work related to their experiments and addressing the neurobiological mechanisms underlying memory linking (probably the equivalent to their “false memory”) such as Cai et al., 2016. In fact, most of what is discussed in the manuscript can also be interpreted under the light of the two memories (context A and B) being linked, rather than “confused” or “difficult to discriminate” as the authors suggest.
Having said this, my biggest suggestion is that the authors come with a different name for what they refer to with “generalization”, that then can be for the first time linked to false memory formation and CRF levels. This could improve the paper.”
As we mentioned above in our response to Reviewer 1 comment 2, according to the general definition of published many papers, we acknowledge that our interpretation for the term of “Fear generalization” in first submitted manuscript was not adequate for our results. We carefully read the paper by Cai et al 2016 many times and we agreed that the concept of the overlapping between the neuronal ensembles representing two separate contextual memories within a day and over a week. Therefore, we extensively revised our suggestion throughout the manuscript. Please see our response to Reviewer 1 comment 2 as above.
Reviewer 1 comment 4: “In the results section, sub-titles should reflect statements related to the most important finding (“manipulation A results in B”) rather than a generic sentence.”
We acknowledge that the sub-titles did not include our important finding. We have changed two sub-titles in the Results section as follow:
Results (page 2, line 85 – 86 in untracked-changed version):
- 1. Exposure of novel context in 3 h after fear conditioning formed false fear memory that was further enhanced at 24 hours after conditioning.
Results (page 5, line 160 – 161 in untracked-changed version):
- 2. Hy-Crf-KD enhanced false fear memory level in 24 h and Hy-CRF-OE potentiate the false fear memory level in 3 h after fear conditioning.
Reviewer 1 comment 5: “Fig 1. The two contexts have the same shape. How do the authors think their results would look if the two contexts were more dissimilar, e.g., with a different shape?”
We acknowledge that we should discussed about the possibility when the mice were exposed to a dissimilar chamber as a novel box, whether the mice represent the false fear memory or not. Therefore, we cite the adequate reference that the mice expose to the dissimilar box after the conditioning, mice did not exhibit the freezing behaviors in 3 hours after the conditioning in Discussion section as follow:
Discussion (page 10, line 260 – 263 untracked-changed version):
It has been demonstrated that the mice were habituated to exposed to a novel and highly dissimilar context with conditioning chamber exhibited a low level of freezing in dissimilar chamber on day 0 [26].
Reference:
[26] Fujinaka, A.; Li, R.; Hayashi, M.; Kumar, D.; Changarathil, G.; Naito, K.; Miki, K.; Nishiyama, T.; Lazarus, M.; Sakurai, T.; et al. Effect of context exposure after fear learning on memory generalization in mice. Mol Brain 2016, 9, 2, doi:10.1186/s13041-015-0184-0.
Reviewer 1 comment 6: “Fig 2. In our opinion, the false memory vs. generalization dichotomy brings problems in fig 2: Here, the authors say that “false fear memory (in context B) was induced 24h after the CS” and, below, “formed within 3h after CS” (page4, line112), after comparing freezing in A-B 24h vs. A-B 3h. We understand from this is to suggest that elevated levels of freezing both at 3h and 24h after conditioning are interpreted by the authors as similar false memory. However, 3h is similar to the time required for memory linking (Cai et al., 2016), whereas 24h could be, although still too short, more widely associated with the idea of “generalization”. In the concluding sentence, they state that, on the one hand, false memory was induced within 3h and, on the other hand, that generalization was produced if the mice had experienced the intermediate exposure to context B. Looking at the data, it is clear that “generalization” is produced regardless of the intermediate exposure (as evidenced by increased freezing in A-B 24h). What remains interesting is the fact that prompt formation of false memory (3h) is necessary for having increased freezing levels at 24h compared to those without 3h exposure to context B. This is probably the most interesting observation, that in our opinion becomes diluted in the false memory vs. generalization terminology/conceptual discussion. Calling this something different, such as “false memory enhancement” would eliminate this problem.
As related the suggestion by Reviewer 1 in the comment 2 and 3, we have performed extensive revised throughout the manuscript and the Title, Abstract, Introduction, Results, Discussion and Conclusion were corrected as above. Please see our response to Reviewer 1 comment 2.
Reviewer 1 comment 7: “In comparing freezing after 24h in context A in A-A vs. A-B-A mice (fig. suppl. 1C vs. D), it seems like exposure to context B 3h after conditioning weakened memory expression in the context A. First, the experiment ABA comes a little bit out of the blue and should be more properly introduced. Second, this observation might be linked to the same concept as before, in the sense that intermediate exposure to different contexts affect memory expression 24h later, which is very interesting. How do the authors interpret this?”
We acknowledge that the results of freezing level 24 h in ABA mice looked decreased, compared to that in A-A mice. However, we did not these two experiments were not performed at the same time (not same batch). Therefore, we cannot represent these results as one graph and perform the statistical analysis. We discussed the possibility that the shorter freezing level 24 h in ABA mice than that in A-A mice in Discussion section as follow:
Discussion (page 11, line 293 – 309 in untracked-changed version):
In the present study, exposing the mice to the novel box (i.e., Context B in Box B) 3 h after the CS (i.e., Context A in Box A), caused the freezing level of ABA mice in Box A at 24 h after the CS to decrease, compared with that of mice those were not exposed to Box B at 3 h (i.e., A-A mice). However, we did not perform these experiments at the same time (i.e., not in the same batch) and, so, we cannot statistically compare the difference in freezing level between ABA and A-A mice. If the freezing level of ABA mice in Box A at 24 h was decreased, compared to that of the A-A mice, two hypotheses can be posed. One is memory extinction, and the other is that the experience in the novel Box B at 3 h interfered with the memory consolidation regarding Box A, which delivered the electrical shocks. The widely used paradigm of memory extinction is generally that, after the CS, the mice are exposed to the same chamber several times without the CS; however, we used a novel box after the CS. On the other hand, the memory of Box A, with the CS, might have become less precise by interfering with the perceptual memory though exposure to Box B at 3 h. This result can be also considered in terms of the hypothesis that a shared neural ensemble may link distinct memories that are encoded within a close timeframe due to a temporary increase in neuronal excitability.
Reviewer 1 comment 8: “Fig 3. Subfigures A-D can be moved to supplementary materials. The entire maps of the viral vectors are not necessary, a schematic representation is enough. In B, do the authors have in vitro evidence that their overexpression constructs worked? In D, what does “Nat Neurosci” and the different code numbers refer to? In F, the distance indicated by the scale bars should be indicated in the figure legend. Furthermore, a DAPI stain would be an added plus, as well as a zoomed out image of the hypothalamus. Finally, the figure legend contains a panel “M?”
We acknowledge that“Nat Neurosci” and the different code numbers in the figure are not explained in legend of Figure 3. We changed the indication of each band of western blotting and explained in figure legend refer to the“Nat Neurosci” and the code numbers. In addition, we wrote the scale bar in the figure legend and delete “M”. As for counterstaining with DAPI for CRF and fluorescence tag (GFP or RFP), we do not have a laser for detecting the DAPI, so we tried to purchase the TO-PRO-3 which can be detected by Ex 642. Unfortunately, we have to import from the Sigma-Aldrich or Thermo Fisher Scientific in the USA, so we have to wait over three weeks to get this. Therefore, we cannot try to stain the nuclear in the hypothalamic sections by deadline during revision.
Subfigures A-D in Figure 3 have moved to the Supplementary Figure S2 and changed the Figure 3. Also, we improved the Methods section and the legend of Figure 3 and separate it to the supplementary Figure S2 as follow:
Methods (page 15, line 495 – 503 in untracked-changed version):
The shRNA sequences used to knockdown Crf were based on a previous report [56], the Sigma MISSION shRNA library by the RNAi Consortium (Boston, MA, USA): (#1) AGATTATCGGGAAATGAAA [56], TRCN0000414479 (#2) TTAGCTCAGCAA-GCTCACAG, TRCN0000436997: and (#3): ATCTCTCTGGATCTCACCTTC. We assessed the knockdown efficiencies of shRNA candidates by co-transfection of each shRNA-carrying AAV plasmid and FLAG-CRF-encoding pCAGGS plasmid into HEK293FT cells. CRF expression was analyzed by immunoblot with anti-FLAG-HRP antibody (#015-22391, FUJIFILM Wako Pure Chemical Corp., Tokyo, Japan) (Fig. S2C, S2D and Fig. S3).
Reference:
[56] Elliott, E.; Ezra-NevoL, G.; Regev, L.; Neufeld-Cohen, A.; Chen, A. Resilience to social stress coincides with functional DNA methylation of the Crf gene in adult mice. Nat Neurosci 2010, 13, 1351-1353, doi:10.1038/nn.2642.
Figure 3 legend (page 6, line 167 – 177 in untracked-changed version):
Figure 3. (A) AAV-PHP.eB virus injection maps according to the Paxinos mouse brain atlas (anterior: -0.7 mm, lateral: 0.25 mm, Depth: 4.40 mm). Green bars around the center of the brain map indicate virus injection needles and the location of virus injections into the hypothalamus including the PVN (0.1 μL of virus injected by 32-gauge Neurosyringe). In the images, "3V” represents the third ventricle and PVN represents the paraventricular nucleus of hypothalamus in the red square, in the hypothalamus. (B) Transduction of mouse hypothalamus with AAV-PHP.eB vector expressing GFP or RFP and detection of mouse CRF by immunofluorescence in control (a-d; upper lane), knockdown (e-h; middle lane; Hy-Crf-KD), and overexpression (i-l; lower lane, Hy-CRF-OE) mice. In the images, “3V” represents the third ventricle and “PVN” represents the paraventricular nucleus of hypothalamus. Scale bar = 200 μm (×10 magnification) for a, e, I and 50 μm for b, c, d, f, g, h, j, k, l (×40 magnification).
Supplementary Fig. S2 (Page 3, line 56 in untracked-changed version):
Construction and characterization of adeno-associated virus (AAV) PHP.eB vectors for knockdown of mouse Crf by shRNA system and overexpression of mouse CRF. (A) Illustration of the AAV vectors expressing Crf shRNAs under the control of U6 promoter. The vectors express GFP as a reporter gene under the control of Chicken β-Actin promoter (CBA pro). (B) Illustration of the AAV vector carrying CRF*-FLAG-T2A-RFP. CRF* represents the shRNA#2 resistant version. (C) Western blotting for expression of CRF*-FLAG-T2A-RFP carrying AAV plasmid that was transfected to HEK293 cells; (D) The knockdown validation of the effective shRNA candidates. #2 shRNA was the most effective, and so was used in further experiments.
Reviewer 1 comment 9: “Fig 4. 4C is referenced in the text before 4B.”
We acknowledge that the describing the result of 4B should be before the 4C. Therefore, we have changed the sentences as follow:
Results (Page 7, line 186 – 193 in untracked-changed version):
The freezing level was similar in the three groups of ABB-control, ABB-KD, and ABB-OE mice during the three electric shocks in Box A as a conditioning stimulus (Fig. 4B). One-way ANOVA revealed that the AAV injections affected the locomotor activity in Box A for 3 min just before the CS was delivered (F(2, 35) = 4.787, p < 0.05, Fig. 4C). Fisher's LSD test indicated that the locomotor activity in Hy-Crf-KD mice was higher than that in control mice and Hy-CRF-OE mice (control: p < 0.01 vs. Hy-Crf-KD; Hy-CRF-OE: p < 0.05 vs. Hy-Crf-KD, p = 0.277 vs. control, Fig. 4C).
Reviewer 1 comment 10: “Fig 4. Variability of freezing is high among injected animals. Was an outlier test carried out? Were injection sites checked and animals with poor injection discarded from the test? Were all the animals conditioned and tested at the same time? Otherwise, inter-group comparisons of total freezing levels might be subject to batch effects that are difficult to account for. In any case, the difference in freezing levels between 3h and 24h is interesting and adequate in trying to normalize freezing per group even if this was obtained from different batches.”
As Reviewer 2 suggested to summarize the statistics, we represented all averages and SEMs in the Table 1 and 2 refer to the Figure 2 and 4. The range of SEM in 3 h after the conditioning in non-AAV injected mice were from 1.373 to 11.01%, while the range of SEM in 3 h after the conditioning in AAV-injected mice were from 0.394 to 5.058. Therefore, we do not think the variability of freezing is not higher among AAV injected mice than non-injected mice. Please see the Table 1: page 5, line 156 and Table 2: page 9, line 246 – page 10, line 248 in untracked-changed version.
Reviewer 1 comment 11: “Page 6, line 189. This was the first time, together with the conclusion of the first part of the results, we understood that the authors use “fear generalization” for the increase in freezing observed at 24h after the pre-exposure to context at 3h post-conditioning… This is problematic because this is only clearly stated in the discussion and reflects and interpretation of the authors, it is difficult to follow the paper if this is not defined before. Again, we would suggest that the authors come with a new name for the phenomenon they observe, to distinguish it from the more traditional idea of generalization. This might help the reader to not get distracted in terminology issues.”
As related suggestion by Reviewer 1 in the comment 2 and 3, we have performed extensive revised throughout the manuscript and the Title, Abstract, Introduction, Results, Discussion and Conclusion were corrected. Please see our response to Reviewer 1 comment 2 as above.
Reviewer 1 comment 12: “Fig 5. We appreciate the effort in trying to link anxiety leading to “false memory” to freezing the day after, but do not quite understand the logic behind this analysis and what additional information different from freezing this brings… it is obvious that the longer a mouse will spend freezing, the less distance it will move. Correlation of 3h distance vs. 24h freezing is lost in genetically manipulated mice, but correlation is not causation (statements as in page10 of 17, line 358 should be weakened). This only reinforces our opinion that it is extremely difficult to extract any conclusion about anxiety levels from assessments that are no de-coupled from fear conditioning. Assessing distance travelled in context A before CS, as done in fig4C, is the closest and most appropriate way.”
We acknowledge that this part is not adequate and awkward phrase. Also, we should not discuss the relationship from the result between the center preference in the box and the anxiety level. However, the findings the central preference in the Box B 3h and the enhancement of false fear memory in 24 h after the conditioning is very important. We are currently the possibility that the center preference is a kind of active coping against novel environment at 3 h and decrease the enhancement of false fear memory in 24 h. Therefore, we changed the sentences in Results and Discussion sections as follow:
Results (page 9, line 289 – 310 in tracked-changed version):
To confirm the anxiety levelbehavioral patterns in mice with genetic manipulation of CRF expression in the hypothalamus, we measured the percentage of the distance traveled and the time spent in the central zone in Box B 3 and 24 h after the CS during Context A. Two-way (time × CRF) repeated measures ANOVA revealed that only “time”, but not “CRF” affected the preference for the central zone in Box B (% distance traveled in central zone: two-way repeated measures ANOVA: F(4, 72) = 0.4528, p = 0.770; factor (time): F(2, 72) = 17.7202, p < 0.01; factor (CRF): F(2, 36) = 0.91767, p = 0.4089409; % time spent in central zone: two-way repeated measures ANOVA: F(4, 72) = 0.6403, p = 0.6355636; factor (time): F(2, 72) = 18.262, p < 0.01; factor (CRF): F(2, 36) = 0.5075508, p = 0.6062, Fig. 5A and B). However, one-way ANOVA revealed that the preference for the central zone in either Hy-Crf-KD or Hy-CRF-OE mice at each time point was similar to that of control mice (distance traveled: 0 h: F(2, 38) = 0.0132, p = 0.9869987; 3 h: F(2, 38) = 1.7432, p = 0.1894; 24 h: F(2, 38) = 0.5693, p = 0.5720, Fig. 5A; time spent: 0 h: F(2, 38) = 0.0974, p = 0.9074; 3 h: F(2, 38) = 1.1780, p = 0.3195320; 24 h: F(2, 38) = 0.9160, p = 0.4092, Fig. 5B). Conversely, there was no significant correlation between the freezing level in 24 h and the distance traveled by Hy-Crf-KD and Hy-CRF-OE mice in Box B in 3 h (Hy-Crf-KD mice: Pearson’s correlation, t = -0.5064, df = 9, p = 0.3124, R2 = 0.0277, Fig. 5D; Hy-CRF-OE mice: Pearson’s correlation, t = -0.37617, df = 13, p = 0.3564, R2 = 0.011; Fig. 5E). Considering that the hypothalamic CRF did not affect the preference for the central zone during Context B 3 h, the disrup-tion of the correlation between the preference for the central zone and the freezing level during Context B 24 h in Hy-Crf-KD and Hy-CRF-OE mice might be due to their discrimination level toward Box A and Box B in 3 h after the CS.
Discussion (page 16, line 547 – 557 in tracked-changed version):
Finally, in the present study, we found that the preference for the central zone in Box B at 3 h—as a novel environment after the CS—affected the freezing level during Context B 24 h in control mice. While hypothalamic CRF did not affect the preference for the central zone during Context B 3 h, the significant correlation between the preference for the central zone in a novel chamber (Box B 3 h) and the freezing level during Context B 24 h was abolished in Hy-Crf-KD and Hy-CRF-OE mice. Therefore, we also suggest that adequate active coping behavior when in a novel environment may decrease and subsequently induce false context fear memory formation.
Reviewer 1 comment 12: “- Line 30, evidence is not plural”
We thank your suggestion for our grammatical error in the manuscript. We corrected from “influence” to “influences”.
Reviewer 1 comment 13: “- Line 64: The statement “in patients with PTSD” is grossly misleading, as mice are being studied here.”
We acknowledge that the statement “in patients with PTSD” is not adequate in our mouse study. According to the correction of our focus in this study, we focus on the false fear memory, but not fear generalization. Therefore, we excluded this phrase “in patients with PTSD”.
Reviewer 1 comment 14: “- Line 143-147. This is not a proper sentence, grammatically speaking.”
We acknowledge that this part of our English was very poor. Therefore, we corrected this sentence in the Results section as follow:
Results (page 5, line 162 – 165 in untracked-changed version):
To investigate the effects of HPA axis activity on false fear memory, we induced mice with Hy-Crf-KD or Hy-CRF-OE through AAV-PHP.eB-produced virus injection into the hypothalamus, for comparison with mice injected with AAV-PHP.eB GFP (green fluorescence protein) into the hypothalamus as a control (Fig. 3 and Fig. S2).
Responses to Reviewer 2 (2nd round of revision):
Reviewer 2 comment 1: “The authors did an excellent job clarifying the questions I have raised in my previous round of review. Currently, this paper entitled ‘Formation of false context fear memory and fear generalization are respectively regulated by distinct expression levels of hypo-thalamic corticotropin-releasing factor in mice’, is a well-written, timely piece of research and provides an useful summary of possible effect of hypothalamic corticotropin-releasing factor (CRF) knockdown or overexpression on contextual fear memory.
I believe that this paper does not need a further revision, therefore the manuscript meets the Journal’s high standards for publication.
I am always available for other reviews of such interesting and important articles.
Thank You for your work.”
We all appreciate the Reviewer 2 for the valuable critical comments and excellent suggestions, which have further contributed to the present improvement of the manuscript. Also, we really happy to hear that your comment against our 1st round of revised version of our manuscript.
We also apologize that we blanked the page and line number in our responses for your reference in the 1st round of “Response to Reviewers”. We did not put the page and line number in ijms template of tracking version. After the submission of 1st our revised version, we learned how we submit the tracking version in ijms template and how we put the page and line number when the template manuscript was used. We also recorrected the 1st round of our “Response to Reviewers” after this 2nd round of “Response to Reviewer”. Please see our 1st round of “Response to Reviewer” as follow:
Responses to Reviewer 2 (1st round of revision):
Reviewer 2 comment 1: “Abstract: Please rephrase the results and conclusion to make them clear for readers to understand.”
We acknowledge that our abstract contained many results and lack of the background of the present study. Also, due to the limitation of number of words, it was not clear what our conclusion is refer to. As Reviewer 1 suggested that our interpretation of definition of “fear generalization” is wrong and the “fear generalization” we claimed in first our draft should be “enhancement of false memory” according to the classical definition of “fear generalization” which are used in the symptom of PTSD. We agreed the Reviewer 1 suggestion and extensively corrected our claim throughout the manuscript. So, we also improved the abstract to make it clear for reader to understand by adding the sentence to describe the background before the results and mention the conclusion more clearly as Reviewer 2 suggestion in the Abstract section as follow:
Abstract (page 1, line 11 – 24 in untracked-changed version):
Traumatic events frequently produce false fear memories. We investigated the effect of hypothalamic corticotropin-releasing factor (CRF) knockdown (Hy-Crf-KD) or overexpression (Hy-CRF-OE) on contextual fear memory, as fear stress-released CRF and hypothalamic–pituitary–adrenal axis activation affects the memory system. Mice were placed in a chamber with an electric footshock as a conditioning stimulus (CS) in Context A, then exposed to a novel chamber without CS, as Context B, at 3 h (B-3h) or 24 h (B-24h). The freezing response in B-3h was intensified in the experimental mice, compared to control mice not exposed to CS, indicating that a false fear memory was formed at 3 h. The within-group freezing level at B-24h was higher than that at B-3h, indicating that false context fear memory was enhanced at B-24h. The difference in freezing levels between B-3h and B-24h in Hy-Crf-KD mice was larger than that of controls. In Hy-CRF-OE mice, the freezing level at B-3h was higher than that in control and Hy-Crf-KD mice, while the freezing level in B-24h was similar to that in B-3h. Locomotor activity before CS and freezing level during CS were similar among the groups. Therefore, we hypothesized that Hy-Crf-KD potentiates the induction of false context fear memory, while Hy-CRF-OE enhances the onset of false fear memory formation.
Reviewer 2 comment 2: “In general, I recommend authors to use more evidence to back their claims, especially in the Introduction of the article, which I believe is currently lacking. Thus, I recommend the authors to attempt to deepen the subject of their manuscript, as the bibliography is too concise: nonetheless, in my opinion, less than 60/70 articles for a research paper are insufficient. Indeed, currently, authors cite only 50 papers, and they are too low. Therefore, I suggest the authors to focus their efforts on researching more relevant literature: I believe that adding more studies and reviews will help them to provide better and more accurate background to this study.”
We acknowledge that our first manuscript lacked the background of our study and add many reference papers as the Reviewer 2 suggested. Please see our reply to the Reviewer 2 comment 3 what and where we changed the sentences as a background according to the suggestions of Reviewer 2 comment 3 in the Introduction section.
Reviewer 2 comment 3: “Introduction: The authors decided to focus specifically on animal models to discuss the role of hypothalamic CRF in the induction of false context fear memory and fear generalization, and, in my opinion, this is too limiting. Thus, I suggest reshaping the Introduction section, which seems not enough extensive and it does not seem to consider, in most cases, all the available studies in the literature that have acknowledged inhomogeneous and dispersive. I think that more information about the ability to use contextual information to modulate the expression of fear would provide a better background here. I suggest the authors to make such effort to provide a brief overview of the pertinent published literature that offer a perspective on altered brain circuits underlying aberrant fear learning process in PTSD, because as it stands, this information is not highlighted in the text. In this regard, I would recommend focusing on the role that the ventromedial prefrontal cortex and hippocampus network have in context-dependent fear learning, addressing how impairments in these brain regions affect fear memories: evidence from a recent theoretical review (https://doi.org/10.1038/s41380-021-01326-4) that focused on neurobiology of fear conditioning, analyzed the role of the ventromedial prefrontal cortex (vmPFC) was analyzed in the processing of safety-threat information and their relative value, and how this region is fundamental for the evaluation and representation of stimulus-outcome’s value needed to produce sustained physiological responses. Also, I believe that a recent yet relevant perspective manuscript (https://doi.org/10.17219/acem/146756) might be of interest: here the focus was on providing a deeper understanding of human learning neural networks, particularly on human PFC crucial role, that might also contribute to the advancement of alternative, more precise and individualized treatments for psychiatric disorders.”
We acknowledge that our first manuscript lacked the background of our study as Reviewer 2 suggested. Therefore, we added many reference papers as Reviewer 2 suggested and extensively improved the Introduction section as follow:
Introduction (page 2, line 52 – 65 in untracked-changed version):
Cortisol modulates various learning and memory processes, depending on the particular timing of cortisol increases relative to encoding, consolidation and retrieval [10]. Long-term dysregulation of cortisol systems after activation of the HPA-axis has been suggested to have lasting effects on the vulnerable areas of the hippocampus, amygdala, and medial prefrontal cortex [11]. The acquisition of fear associative memory requires HPA-axis-related various brain processes involving coordinated neural activity within the amygdala [10,12,13], prefrontal cortex (PFC) [14,15], and hippocampus [14-17]. The amygdala plays a key role in the acquisition of fear learning, while the PFC and hippocampus are two other crucial neural structures that contribute to this process, together representing the neural network of fear conditioning [18]. The ventral part of the medial PFC may also play a major role in fear conditioning [19-21]. Traumatic events within a few hours, increase corticotropin-releasing factor (CRF) levels and activation of the HPA-axis, which may affect the memory consolidation. However, how the stress-induced activation of the HPA-axis, contributes to the formation of false context fear memory remains unclear.
References:
[10] Merz, C.J.; Hermann, A.; Stark, R.; Wolf, O.T. Cortisol modifies extinction learning of recently acquired fear in men. Social cognitive and affective neuroscience 2014, 9, 1426-1434, doi:10.1093/scan/nst137.
[11] Bremner, J.D. Traumatic stress: effects on the brain. Dialogues in clinical neuroscience 2006, 8, 445-461, doi:10.31887/DCNS.2006.8.4/jbremner.
[12] Skórzewska, A.; Lehner, M.; WisÅ‚owska-Stanek, A.; TurzyÅ„ska, D.; Sobolewska, A.; KrzÄ…Å›cik, P.; Szyndler, J.; Maciejak, P.; Chmielewska, N.; KoÅ‚osowska, K.; et al. Individual susceptibility or resistance to posttraumatic stress disorder-like behaviours. Behavioural brain research 2020, 386, 112591, doi:10.1016/j.bbr.2020.112591.
[13] Li, G.; Wang, G.; Shi, J.; Xie, X.; Fei, N.; Chen, L.; Liu, N.; Yang, M.; Pan, J.; Huang, W.; et al. trans-Resveratrol ameliorates anxiety-like behaviors and fear memory deficits in a rat model of post-traumatic stress disorder. Neuropharmacology 2018, 133, 181-188, doi:10.1016/j.neuropharm.2017.12.035.
[14] Perrine, S.A.; Eagle, A.L.; George, S.A.; Mulo, K.; Kohler, R.J.; Gerard, J.; Harutyunyan, A.; Hool, S.M.; Susick, L.L.; Schneider, B.L.; et al. Severe, multimodal stress exposure induces PTSD-like characteristics in a mouse model of single prolonged stress. Behavioural brain research 2016, 303, 228-237, doi:10.1016/j.bbr.2016.01.056.
[15] Ribeiro, T.O.; Bueno-de-Camargo, L.M.; Waltrick, A.P.F.; de Oliveira, A.R.; Brandão, M.L.; Munhoz, C.D.; Zanoveli, J.M. Activation of mineralocorticoid receptors facilitate the acquisition of fear memory extinction and impair the generalization of fear memory in diabetic animals. Psychopharmacology 2020, 237, 529-542, doi:10.1007/s00213-019-05388-9.
[16] Sarabdjitsingh, R.A.; Zhou, M.; Yau, J.L.; Webster, S.P.; Walker, B.R.; Seckl, J.R.; Joëls, M.; Krugers, H.J. Inhibiting 11β-hydroxysteroid dehydrogenase type 1 prevents stress effects on hippocampal synaptic plasticity and impairs contextual fear conditioning. Neuropharmacology 2014, 81, 231-236, doi:10.1016/j.neuropharm.2014.01.042.
[17] Merz, C.J.; Hamacher-Dang, T.C.; Stark, R.; Wolf, O.T.; Hermann, A. Neural Underpinnings of Cortisol Effects on Fear Extinction. Neuropsychopharmacology 2018, 43, 384-392, doi:10.1038/npp.2017.227.
[18] Battaglia, S. Neurobiological advances of learned fear in humans. Advances in clinical and experimental medicine : official organ Wroclaw Medical University 2022, 31, 217-221, doi:10.17219/acem/146756.
[19] Battaglia, S.; Garofalo, S.; di Pellegrino, G.; Starita, F. Revaluing the Role of vmPFC in the Acquisition of Pavlovian Threat Conditioning in Humans. The Journal of neuroscience : the official journal of the Society for Neuroscience 2020, 40, 8491-8500, doi:10.1523/jneurosci.0304-20.2020.
[20] Fullana, M.A.; Harrison, B.J.; Soriano-Mas, C.; Vervliet, B.; Cardoner, N.; Àvila-Parcet, A.; Radua, J. Neural signatures of human fear conditioning: an updated and extended meta-analysis of fMRI studies. Molecular psychiatry 2016, 21, 500-508, doi:10.1038/mp.2015.88.
[21] Battaglia, S.; Harrison, B.J.; Fullana, M.A. Does the human ventromedial prefrontal cortex support fear learning, fear extinction or both? A commentary on subregional contributions. Molecular psychiatry 2022, 27, 784-786, doi:10.1038/s41380-021-01326-4.
Reviewer 2 comment 4: “Introduction: Following the first point raised, I would also recommend another recent opinion manuscript which provided novel ‘functional interplay between central and autonomic nervous systems in human fear conditioning’, and highlights the crucial role of the prefrontal cortex. Finally, authors also might to consider some studies that have focused on this topic (https://doi.org/10.3390/biomedicines10010076; https://doi.org/10.3390/biomedicines9050517).”
We acknowledge that our first manuscript lacked the background of HPA-axis activity related symptom. Therefore, we added the following sentence in the Introduction section as Reviewer 2 suggested;
Introduction (page 2, line 69 – 71 in untracked-changed version):
Moreover, HPA-axis activity links the various symptom regarding the autonomic function, anxiety, locomotor activity or cognitive functions [23].
References:
[23] Tanaka, M.; Vécsei, L. Editorial of Special Issue "Crosstalk between Depression, Anxiety, and Dementia: Comorbidity in Behavioral Neurology and Neuropsychiatry". Biomedicines 2021, 9, doi:10.3390/biomedicines9050517.
Reviewer 2 comment 5: “Behavioral tests: Could the authors provide the specific number of mice that were used in the experiments?”
We acknowledge that our first manuscript lacked the total number of mice were used in the Methods section as follow: The number of mice we used in each group of each experiment was shown in Figure legends. Also, as Reviewer suggested, we put two summary Table1 and 2 in the revised version.
Methods (page 13, line 417 – 427 in untracked-changed version):
Total 15 mice were used for measuring the plasma corticosterone concentration (5 mice for each group). For the behavioral test, in non-AAV infected mouse test (Figure 2 and Figure S2), 54 mice (24 mice for Figure 2 and 30 mice for Figure 1S) and mice were used and AAV infected mouse test (Figure 4), 57 mice were used for behavioral tests. Total 126 mice were used in the present study. In the test using a group of A-B (CS-) mice, A-B (CS+) mice, ABB (CS-) mice, ABB (CS+) mice in Figure 2 were performed at the same time respectively. Also, all of AAV injected mice with CS group (Hy-Control (CS+), Hy-Crf-KD (CS+) and Hy-CRF-OE (CS+)) were tested at the same time and all of AAV injected mice without the CS group (Hy-Control (CS-), Hy-Crf-KD (CS-) and Hy-CRF-OE (CS-)) were tested at the same time. Every day, 3 to 4 mice were tested and repeated these tests until the number of mice were enough to analyzing.
Reviewer 2 comment 6: “Contextual fear conditioning test: This paragraph that explains how mice were fear conditioned is the most important part of the study and should clearly describe all the experimental sessions in detail; therefore, this section might be improved by including further explanations, allowing the effective communication of experimental procedures.”
We acknowledge that our first manuscript lacked the detail of the method of conditioning test in the present study. Because we described some sentences about the conditioning test in the first paragraph of our first submission in the “Contextual fear conditioning” which also describing the apparatus and it might make a confusion for reader. Therefore, we separate the part of “Apparatus for the contextual fear conditioning test” and “Contextual fear conditioning” in the Methods section as follow:
Methods (page 14, line 457 – 462 in untracked-changed version):
The mice were handled for 1 min for three consecutive days before the behavioral test. All mice were habituated to Box A 60 min before Context A for 15 min. Mice were placed in Box A again for 3 min, and then electrical shocks (1.0 mA for 2 s, delivered three times at 100 s intervals) in Box A 3 min after the mice were placed on the grid floor, designated as Context A. After exposure to Context A for fear conditioning in Box A, all mice were returned to their home cage until the next context.
Methods (page 14, line 477 – 478 in untracked-changed version):
In all test in the present study, the chambers were thoroughly cleaned with a 70% ethanol solution between mice every times.
Reviewer 2 comment 7: “Results: In my opinion, this section is well organized, but it illustrates findings in an excessively broad way, without really providing full statistical details, to ensure in-depth understanding and replicability of the findings. Also, in my opinion, it is necessary for the authors to present their findings using summary tables.”
We acknowledge that the statistics were not sufficient for describing all results in our first manuscript. As Reviewer 2 suggested, we put two summary Table 1 and 2 for detail information about the results of Figure 2 and 4 including statistics in our revised version of our manuscript. Please see the two tables in the template manuscript.
Reviewer 2 comment 8: “Discussion: In this final section, the authors described the results and their argumentation and captured the state of the art well; however, I would have liked to see some views on a way forward. I believe that the authors should make an effort, trying to explain the theoretical implication as well as the translational application of this research article, to adequately convey what they believe is the take-home message of their study. Discussion of theoretical and methodological avenues in need of refinement is necessary, as well as suggestions of a path forward in understanding the role of hypothalamic CRF on the induction of false context fear memory and fear generalization. In this regard, recent evidence suggests that the application of new methods in Neuroinflammatory disorders’ treatment, such as the Non-invasive brain stimulation techniques (NIBS), have shown promising results in humans (https://doi.org/10.1038/s41398-020-0851-5). Importantly, I recommend referring to recent studies that revealed that the application of NIBS induces long-lasting effects, noninvasively modulating the cortical excitability, and modulates a variety of cognitive functions altered in patients that suffer from PTSD: for example, a recent review (https://doi.org/10.1016/j.neubiorev.2021.04.036) on the potential and effectiveness of non-invasive brain stimulation (NIBS) to interfere and modulate the abnormal activity of neural circuits (i.e., amygdala-mPFC-hippocampus) involved in the acquisition and consolidation of fear memories, which are altered in many mood psychiatric disorders (i.e., anxiety disorder, specific phobias, post-traumatic stress disorder or depression), would be of interest. Accordingly, another recent manuscript (https://doi.org/10.1016/j.jad.2021.02.076) focused on the same topic, illustrating the therapeutic potential of NIBS as a valid alternative in the treatment of untypically persistent memories that characterized those patients that do not respond to psychotherapy and/or drug treatments. In addition to the previously mentioned literature, authors might also see these additional studies that have focused on the efficacy of NIBS and IBS (https://doi.org/10.3389/fpsyt.2018.00201; https://doi.org/10.3389/fnagi.2020.578339).”
We appreciate the Reviewer 2 suggestion to discuss the important background and possibility raised from our results in the present study. As related to the Reviewer 2 comment 2 and 3, we discussed about the amygdala-mPFC-hippocampus in Discussion section as follow:
Discussion (page 12, line 366 – page 13, line 387 in untracked-changed version):
Functional alterations of the neural network underlying fear conditioning might contribute to the etiology of fear-related psychiatric disease, including PTSD [18]. Fear-associative memory acquisition of fear learning requires coordinated neural activity within the amygdala, prefrontal cortex (PFC), and hippocampus [17-21,48,49]. In addition, cortisol exerts a critical impact on the amygdala–hippocampus–ventromedial PFC network, which underpins fear and memory extinction [17]. Dysregulation of negative feedback, through cortisol suppressing the release of CRF after the long-term activation of the HPA-axis, affects the hippocampus, amygdala, and medial PFC [11]. Inactivation of prefrontal inputs into the nucleus reuniens or direct silencing of nucleus reuniens projections enhances fear memory generalization [27]. Xu and Südhof have demonstrated the generalization of memory attributes for a particular context by processing information from the medial PFC en route to the hippocampus within a day after conditional training [27]. Decreased connectivity be-tween the amygdala and medial PFC has been shown to be related to memory intrusion and the re-experiencing of traumatic events [50]. Interestingly, repetitive transcranial magnetic stimulation of the right dorsolateral PFC appears to have a positive effect in reducing core symptoms in patients with PTSD [50]. Therefore, it is of crucially importance to investigate the role of CRF and cortisol in various neuronal net-works, including the amygdala–hippocampus–ventromedial PFC, with respect to their contribution to both the acquisition of fear memories and the consolidation of imprecise fear memories, although there are still limitations to the elucidation of neuronal mechanisms when using rodent models, as humans and animals differ in terms of the functional neuroarchitecture of the PFC.
References:
- Bremner, J.D. Traumatic stress: effects on the brain. Dialogues in clinical neuroscience 2006, 8, 445-461, doi:10.31887/DCNS.2006.8.4/jbremner.
- Merz, C.J.; Hamacher-Dang, T.C.; Stark, R.; Wolf, O.T.; Hermann, A. Neural Underpinnings of Cortisol Effects on Fear Extinction. Neuropsychopharmacology : official publication of the American College of Neuropsychopharmacology 2018, 43, 384-392, doi:10.1038/npp.2017.227.
- Battaglia, S. Neurobiological advances of learned fear in humans. Advances in clinical and experimental medicine : official organ Wroclaw Medical University 2022, 31, 217-221, doi:10.17219/acem/146756.
- Battaglia, S.; Garofalo, S.; di Pellegrino, G.; Starita, F. Revaluing the Role of vmPFC in the Acquisition of Pavlovian Threat Conditioning in Humans. The Journal of neuroscience : the official journal of the Society for Neuroscience 2020, 40, 8491-8500, doi:10.1523/jneurosci.0304-20.2020.
- Fullana, M.A.; Harrison, B.J.; Soriano-Mas, C.; Vervliet, B.; Cardoner, N.; Àvila-Parcet, A.; Radua, J. Neural signatures of human fear conditioning: an updated and extended meta-analysis of fMRI studies. Molecular psychiatry 2016, 21, 500-508, doi:10.1038/mp.2015.88.
- Battaglia, S.; Harrison, B.J.; Fullana, M.A. Does the human ventromedial prefrontal cortex support fear learning, fear extinction or both? A commentary on subregional contributions. Molecular psychiatry 2022, 27, 784-786, doi:10.1038/s41380-021-01326-4.
- Xu, W.; Südhof, T.C. A neural circuit for memory specificity and generalization. Science (New York, N.Y.) 2013, 339, 1290-1295, doi:10.1126/science.1229534.
- Borgomaneri, S.; Battaglia, S.; Sciamanna, G.; Tortora, F.; Laricchiuta, D. Memories are not written in stone: Re-writing fear memories by means of non-invasive brain stimulation and optogenetic manipulations. Neuroscience and biobehavioral reviews 2021, 127, 334-352, doi:10.1016/j.neubiorev.2021.04.036.
- Stark, R.; Wolf, O.T.; Tabbert, K.; Kagerer, S.; Zimmermann, M.; Kirsch, P.; Schienle, A.; Vaitl, D. Influence of the stress hormone cortisol on fear conditioning in humans: evidence for sex differences in the response of the prefrontal cortex. NeuroImage 2006, 32, 1290-1298, doi:10.1016/j.neuroimage.2006.05.046.
- Kan, R.L.D.; Zhang, B.B.B.; Zhang, J.J.Q.; Kranz, G.S. Non-invasive brain stimulation for posttraumatic stress disorder: a systematic review and meta-analysis. Translational psychiatry 2020, 10, 168, doi:10.1038/s41398-020-0851-5.
Reviewer 2 comment 9: “Even though it is not mandatory, I believe that the ‘Conclusions’ section would be useful to adequately indicate convey what the authors believe is the take-home message of their study, and therefore provide a synthesis of the data presented in the paper as well as possible keys to advancing research and understanding of the prevalence of depression in post-stroke patients.”
We acknowledge that the Conclusion section should be provided for the reader of our study. Therefore, we put the “Conclusion” section after the Discussion as follow:
Conclusion (page 13, line 388 – 405 in untracked-changed version):
In the present study, we found that exposure to a novel but similar contexts to the environment of traumatic fear experience within a few hours induced the formation of false fear memories, which were contextually enhanced within a day after the traumatic fear event. These results suggest that exposure to a novel context within a few hours after a traumatic fear event may interfere with the consolidation of precise fear memory regarding a traumatic event, thus enhancing false context fear memory formation within 24 h. Knockdown of hypothalamic Crf increased the freezing level and potentiated the false context fear memory at 24 h after fear conditioning. On the other hand, over-expression of hypothalamic CRF enhanced the onset of false fear memory formation within a few hours after the traumatic fear event. Cortisol has been suggested to exert a critical impact on the amygdala–hippocampus–ventromedial PFC network, thus underpinning the acquisition of fear memories, as well as the consolidation of precise fear memories and fear generalization. Therefore, it is important to understand how the HPA axis contributes to the formation and potentiation of false con-text fear memories, as well as to explore the associated clinical implications for the treatment of psychiatric diseases, such as PTSD and advance knowledge related to CRF-related cognitive impairment such as post-stroke depression [51], in the future.
Reference:
- Barra de la Tremblaye, P.; Plamondon, H. Alterations in the corticotropin-releasing hormone (CRH) neurocircuitry: Insights into post stroke functional impairments. Frontiers in neuroendocrinology 2016, 42, 53-75, doi:10.1016/j.yfrne.2016.07.001.
Reviewer 2 comment 10: “In according to the previous comment, I would ask the authors to better define a proper ‘Limitations and future directions’ section before the end of the manuscript, in which authors can describe in detail and report all the technical issues brought to the surface.”
We acknowledge that the limitation of our study is important because all of our results were obtained from the rodents. As Reviewer 2 suggested, how our results develop and reflect to the human disease in the future is important for the readers. Although we did not separate the as “Limitation” from the Discussion section, we added the sentences in the Discussion section as follow:
Discussion (page 12, line 381 – page 13, line 387 in untracked-changed version):
Therefore, it is of crucially importance to investigate the role of CRF and cortisol in various neuronal networks, including the amygdala–hippocampus–ventromedial PFC, with respect to their contribution to both the acquisition of fear memories and the consolidation of imprecise fear memories, although there are still limitations to the elucidation of neuronal mechanisms when using rodent models, as humans and animals differ in terms of the functional neuroarchitecture of the PFC.
Reviewer 2 comment 11: “References: Authors should consider revising the bibliography, as there are several incorrect citations. Indeed, according to the Journal’s guidelines, they should provide the abbreviated journal name in italics, the year of publication in bold, the volume number in italics for all the references.”
We thank the Reviewer 2 suggestions. We corrected the style of References. Please see the Reference section in the template manuscript.
Reviewer 2 Report
The authors did an excellent job clarifying the questions I have raised in my previous round of review. Currently, this paper entitled ‘Formation of false context fear memory and fear generalization are respectively regulated by distinct expression levels of hypo-thalamic corticotropin-releasing factor in mice’, is a well-written, timely piece of research and provides an useful summary of possible effect of hypothalamic corticotropin-releasing factor (CRF) knockdown or overexpression on contextual fear memory.
I believe that this paper does not need a further revision, therefore the manuscript meets the Journal’s high standards for publication.
I am always available for other reviews of such interesting and important articles.
Thank You for your work.
Author Response
To the Editors       Fukushima, Japan
International Journal of Molecular Sciences May 29th, 2022
Subject: 2nd round of Revision for Manuscript ID: ijms-1718457
With this letter we would like to submit a 2nd round of revised version of the article entitled “Formation of false context fear memory is regulated by hypothalamic corticotropin-releasing factor in mice” by Kenjiro Seki submitted for publication to the International Journal of Molecular Sciences.
At first, we would like to thank the Reviewer 1 for their valuable critical comments and excellent suggestions, which have further contributed to the present improvement of the manuscript. Also, we apologize that our poor English and we blanked the both of page and line number in the 1st round of Response to Reviewers. We extensively and carefully corrected the sentences using the template of ijms and put the page and line number in this Response to Reviewer 1 including 1st round after the 2nd round of Response to Reviewer 1. We essentially followed the reviewer’s comments and arguments and address all points that had been raised. The manuscript was extensively changed accordingly. We have highlighted all the changes made in the original manuscript below in response to reviewer.
We apologize that our poor English in 1st round of revised version of our manuscript also makes difficult for understanding our works. We acknowledge that the English in our 1st round of revised version of manuscript should be further improved and then we asked the MDPI English Editing Service to correct and improve our 2nd round of revised version of our manuscript (Certification no. 44660).
We also apologize that we blanked the page and line number in our responses for your reference in the 1st round of “Response to Reviewers”. We did not put the page and line number in ijms template of tracking version. After the submission of 1st our revised version, we learned how we submit the tracking version in ijms template and how we put the page and line number when the template manuscript was used. We also recorrected the 1st round of our “Response to Reviewers” after this 2nd round of “Response to Reviewer”. Please see our 1st round of “Response to Reviewer”.
The Abstract, Introduction, Results, Discussion and Methods section were modified according to the reviewer comments/suggestions and additional changes to make reading easier and clearer. References are updated accordingly.
For reference, the reviewer’s comments are shown in quotations. Please note that the referring page and line number shown in each our response to the reviewer comment corresponds to the line number of each page in the untracked-changed version in ijms’s template of revised manuscript. When we showed the deleted parts in this 2nd round of revised version in the “Response to Reviewer”, the corresponding page and line number were shown in tracked-changed version in some cases.
Best Regards,
Kenjiro Seki, Ph.D.
Responses to Reviewer 2 (2nd round of revision):
Reviewer 2 comment 1: “The authors did an excellent job clarifying the questions I have raised in my previous round of review. Currently, this paper entitled ‘Formation of false context fear memory and fear generalization are respectively regulated by distinct expression levels of hypo-thalamic corticotropin-releasing factor in mice’, is a well-written, timely piece of research and provides an useful summary of possible effect of hypothalamic corticotropin-releasing factor (CRF) knockdown or overexpression on contextual fear memory.
I believe that this paper does not need a further revision, therefore the manuscript meets the Journal’s high standards for publication.
I am always available for other reviews of such interesting and important articles.
Thank You for your work.”
We all appreciate the Reviewer 2 for the valuable critical comments and excellent suggestions, which have further contributed to the present improvement of the manuscript. Also, we really happy to hear that your comment against our 1st round of revised version of our manuscript.
We also apologize that we blanked the page and line number in our responses for your reference in the 1st round of “Response to Reviewers”. We did not put the page and line number in ijms template of tracking version. After the submission of 1st our revised version, we learned how we submit the tracking version in ijms template and how we put the page and line number when the template manuscript was used. We also recorrected the 1st round of our “Response to Reviewers” after this 2nd round of “Response to Reviewer”. Please see our 1st round of “Response to Reviewer” as follow:
Responses to Reviewer 2 (1st round of revision):
Reviewer 2 comment 1: “Abstract: Please rephrase the results and conclusion to make them clear for readers to understand.”
We acknowledge that our abstract contained many results and lack of the background of the present study. Also, due to the limitation of number of words, it was not clear what our conclusion is refer to. As Reviewer 1 suggested that our interpretation of definition of “fear generalization” is wrong and the “fear generalization” we claimed in first our draft should be “enhancement of false memory” according to the classical definition of “fear generalization” which are used in the symptom of PTSD. We agreed the Reviewer 1 suggestion and extensively corrected our claim throughout the manuscript. So, we also improved the abstract to make it clear for reader to understand by adding the sentence to describe the background before the results and mention the conclusion more clearly as Reviewer 2 suggestion in the Abstract section as follow:
Abstract (page 1, line 11 – 24 in untracked-changed version):
Traumatic events frequently produce false fear memories. We investigated the effect of hypothalamic corticotropin-releasing factor (CRF) knockdown (Hy-Crf-KD) or overexpression (Hy-CRF-OE) on contextual fear memory, as fear stress-released CRF and hypothalamic–pituitary–adrenal axis activation affects the memory system. Mice were placed in a chamber with an electric footshock as a conditioning stimulus (CS) in Context A, then exposed to a novel chamber without CS, as Context B, at 3 h (B-3h) or 24 h (B-24h). The freezing response in B-3h was intensified in the experimental mice, compared to control mice not exposed to CS, indicating that a false fear memory was formed at 3 h. The within-group freezing level at B-24h was higher than that at B-3h, indicating that false context fear memory was enhanced at B-24h. The difference in freezing levels between B-3h and B-24h in Hy-Crf-KD mice was larger than that of controls. In Hy-CRF-OE mice, the freezing level at B-3h was higher than that in control and Hy-Crf-KD mice, while the freezing level in B-24h was similar to that in B-3h. Locomotor activity before CS and freezing level during CS were similar among the groups. Therefore, we hypothesized that Hy-Crf-KD potentiates the induction of false context fear memory, while Hy-CRF-OE enhances the onset of false fear memory formation.
Reviewer 2 comment 2: “In general, I recommend authors to use more evidence to back their claims, especially in the Introduction of the article, which I believe is currently lacking. Thus, I recommend the authors to attempt to deepen the subject of their manuscript, as the bibliography is too concise: nonetheless, in my opinion, less than 60/70 articles for a research paper are insufficient. Indeed, currently, authors cite only 50 papers, and they are too low. Therefore, I suggest the authors to focus their efforts on researching more relevant literature: I believe that adding more studies and reviews will help them to provide better and more accurate background to this study.”
We acknowledge that our first manuscript lacked the background of our study and add many reference papers as the Reviewer 2 suggested. Please see our reply to the Reviewer 2 comment 3 what and where we changed the sentences as a background according to the suggestions of Reviewer 2 comment 3 in the Introduction section.
Reviewer 2 comment 3: “Introduction: The authors decided to focus specifically on animal models to discuss the role of hypothalamic CRF in the induction of false context fear memory and fear generalization, and, in my opinion, this is too limiting. Thus, I suggest reshaping the Introduction section, which seems not enough extensive and it does not seem to consider, in most cases, all the available studies in the literature that have acknowledged inhomogeneous and dispersive. I think that more information about the ability to use contextual information to modulate the expression of fear would provide a better background here. I suggest the authors to make such effort to provide a brief overview of the pertinent published literature that offer a perspective on altered brain circuits underlying aberrant fear learning process in PTSD, because as it stands, this information is not highlighted in the text. In this regard, I would recommend focusing on the role that the ventromedial prefrontal cortex and hippocampus network have in context-dependent fear learning, addressing how impairments in these brain regions affect fear memories: evidence from a recent theoretical review (https://doi.org/10.1038/s41380-021-01326-4) that focused on neurobiology of fear conditioning, analyzed the role of the ventromedial prefrontal cortex (vmPFC) was analyzed in the processing of safety-threat information and their relative value, and how this region is fundamental for the evaluation and representation of stimulus-outcome’s value needed to produce sustained physiological responses. Also, I believe that a recent yet relevant perspective manuscript (https://doi.org/10.17219/acem/146756) might be of interest: here the focus was on providing a deeper understanding of human learning neural networks, particularly on human PFC crucial role, that might also contribute to the advancement of alternative, more precise and individualized treatments for psychiatric disorders.”
We acknowledge that our first manuscript lacked the background of our study as Reviewer 2 suggested. Therefore, we added many reference papers as Reviewer 2 suggested and extensively improved the Introduction section as follow:
Introduction (page 2, line 52 – 65 in untracked-changed version):
Cortisol modulates various learning and memory processes, depending on the particular timing of cortisol increases relative to encoding, consolidation and retrieval [10]. Long-term dysregulation of cortisol systems after activation of the HPA-axis has been suggested to have lasting effects on the vulnerable areas of the hippocampus, amygdala, and medial prefrontal cortex [11]. The acquisition of fear associative memory requires HPA-axis-related various brain processes involving coordinated neural activity within the amygdala [10,12,13], prefrontal cortex (PFC) [14,15], and hippocampus [14-17]. The amygdala plays a key role in the acquisition of fear learning, while the PFC and hippocampus are two other crucial neural structures that contribute to this process, together representing the neural network of fear conditioning [18]. The ventral part of the medial PFC may also play a major role in fear conditioning [19-21]. Traumatic events within a few hours, increase corticotropin-releasing factor (CRF) levels and activation of the HPA-axis, which may affect the memory consolidation. However, how the stress-induced activation of the HPA-axis, contributes to the formation of false context fear memory remains unclear.
References:
[10] Merz, C.J.; Hermann, A.; Stark, R.; Wolf, O.T. Cortisol modifies extinction learning of recently acquired fear in men. Social cognitive and affective neuroscience 2014, 9, 1426-1434, doi:10.1093/scan/nst137.
[11] Bremner, J.D. Traumatic stress: effects on the brain. Dialogues in clinical neuroscience 2006, 8, 445-461, doi:10.31887/DCNS.2006.8.4/jbremner.
[12] Skórzewska, A.; Lehner, M.; WisÅ‚owska-Stanek, A.; TurzyÅ„ska, D.; Sobolewska, A.; KrzÄ…Å›cik, P.; Szyndler, J.; Maciejak, P.; Chmielewska, N.; KoÅ‚osowska, K.; et al. Individual susceptibility or resistance to posttraumatic stress disorder-like behaviours. Behavioural brain research 2020, 386, 112591, doi:10.1016/j.bbr.2020.112591.
[13] Li, G.; Wang, G.; Shi, J.; Xie, X.; Fei, N.; Chen, L.; Liu, N.; Yang, M.; Pan, J.; Huang, W.; et al. trans-Resveratrol ameliorates anxiety-like behaviors and fear memory deficits in a rat model of post-traumatic stress disorder. Neuropharmacology 2018, 133, 181-188, doi:10.1016/j.neuropharm.2017.12.035.
[14] Perrine, S.A.; Eagle, A.L.; George, S.A.; Mulo, K.; Kohler, R.J.; Gerard, J.; Harutyunyan, A.; Hool, S.M.; Susick, L.L.; Schneider, B.L.; et al. Severe, multimodal stress exposure induces PTSD-like characteristics in a mouse model of single prolonged stress. Behavioural brain research 2016, 303, 228-237, doi:10.1016/j.bbr.2016.01.056.
[15] Ribeiro, T.O.; Bueno-de-Camargo, L.M.; Waltrick, A.P.F.; de Oliveira, A.R.; Brandão, M.L.; Munhoz, C.D.; Zanoveli, J.M. Activation of mineralocorticoid receptors facilitate the acquisition of fear memory extinction and impair the generalization of fear memory in diabetic animals. Psychopharmacology 2020, 237, 529-542, doi:10.1007/s00213-019-05388-9.
[16] Sarabdjitsingh, R.A.; Zhou, M.; Yau, J.L.; Webster, S.P.; Walker, B.R.; Seckl, J.R.; Joëls, M.; Krugers, H.J. Inhibiting 11β-hydroxysteroid dehydrogenase type 1 prevents stress effects on hippocampal synaptic plasticity and impairs contextual fear conditioning. Neuropharmacology 2014, 81, 231-236, doi:10.1016/j.neuropharm.2014.01.042.
[17] Merz, C.J.; Hamacher-Dang, T.C.; Stark, R.; Wolf, O.T.; Hermann, A. Neural Underpinnings of Cortisol Effects on Fear Extinction. Neuropsychopharmacology 2018, 43, 384-392, doi:10.1038/npp.2017.227.
[18] Battaglia, S. Neurobiological advances of learned fear in humans. Advances in clinical and experimental medicine : official organ Wroclaw Medical University 2022, 31, 217-221, doi:10.17219/acem/146756.
[19] Battaglia, S.; Garofalo, S.; di Pellegrino, G.; Starita, F. Revaluing the Role of vmPFC in the Acquisition of Pavlovian Threat Conditioning in Humans. The Journal of neuroscience : the official journal of the Society for Neuroscience 2020, 40, 8491-8500, doi:10.1523/jneurosci.0304-20.2020.
[20] Fullana, M.A.; Harrison, B.J.; Soriano-Mas, C.; Vervliet, B.; Cardoner, N.; Àvila-Parcet, A.; Radua, J. Neural signatures of human fear conditioning: an updated and extended meta-analysis of fMRI studies. Molecular psychiatry 2016, 21, 500-508, doi:10.1038/mp.2015.88.
[21] Battaglia, S.; Harrison, B.J.; Fullana, M.A. Does the human ventromedial prefrontal cortex support fear learning, fear extinction or both? A commentary on subregional contributions. Molecular psychiatry 2022, 27, 784-786, doi:10.1038/s41380-021-01326-4.
Reviewer 2 comment 4: “Introduction: Following the first point raised, I would also recommend another recent opinion manuscript which provided novel ‘functional interplay between central and autonomic nervous systems in human fear conditioning’, and highlights the crucial role of the prefrontal cortex. Finally, authors also might to consider some studies that have focused on this topic (https://doi.org/10.3390/biomedicines10010076; https://doi.org/10.3390/biomedicines9050517).”
We acknowledge that our first manuscript lacked the background of HPA-axis activity related symptom. Therefore, we added the following sentence in the Introduction section as Reviewer 2 suggested;
Introduction (page 2, line 69 – 71 in untracked-changed version):
Moreover, HPA-axis activity links the various symptom regarding the autonomic function, anxiety, locomotor activity or cognitive functions [23].
References:
[23] Tanaka, M.; Vécsei, L. Editorial of Special Issue "Crosstalk between Depression, Anxiety, and Dementia: Comorbidity in Behavioral Neurology and Neuropsychiatry". Biomedicines 2021, 9, doi:10.3390/biomedicines9050517.
Reviewer 2 comment 5: “Behavioral tests: Could the authors provide the specific number of mice that were used in the experiments?”
We acknowledge that our first manuscript lacked the total number of mice were used in the Methods section as follow: The number of mice we used in each group of each experiment was shown in Figure legends. Also, as Reviewer suggested, we put two summary Table1 and 2 in the revised version.
Methods (page 13, line 417 – 427 in untracked-changed version):
Total 15 mice were used for measuring the plasma corticosterone concentration (5 mice for each group). For the behavioral test, in non-AAV infected mouse test (Figure 2 and Figure S2), 54 mice (24 mice for Figure 2 and 30 mice for Figure 1S) and mice were used and AAV infected mouse test (Figure 4), 57 mice were used for behavioral tests. Total 126 mice were used in the present study. In the test using a group of A-B (CS-) mice, A-B (CS+) mice, ABB (CS-) mice, ABB (CS+) mice in Figure 2 were performed at the same time respectively. Also, all of AAV injected mice with CS group (Hy-Control (CS+), Hy-Crf-KD (CS+) and Hy-CRF-OE (CS+)) were tested at the same time and all of AAV injected mice without the CS group (Hy-Control (CS-), Hy-Crf-KD (CS-) and Hy-CRF-OE (CS-)) were tested at the same time. Every day, 3 to 4 mice were tested and repeated these tests until the number of mice were enough to analyzing.
Reviewer 2 comment 6: “Contextual fear conditioning test: This paragraph that explains how mice were fear conditioned is the most important part of the study and should clearly describe all the experimental sessions in detail; therefore, this section might be improved by including further explanations, allowing the effective communication of experimental procedures.”
We acknowledge that our first manuscript lacked the detail of the method of conditioning test in the present study. Because we described some sentences about the conditioning test in the first paragraph of our first submission in the “Contextual fear conditioning” which also describing the apparatus and it might make a confusion for reader. Therefore, we separate the part of “Apparatus for the contextual fear conditioning test” and “Contextual fear conditioning” in the Methods section as follow:
Methods (page 14, line 457 – 462 in untracked-changed version):
The mice were handled for 1 min for three consecutive days before the behavioral test. All mice were habituated to Box A 60 min before Context A for 15 min. Mice were placed in Box A again for 3 min, and then electrical shocks (1.0 mA for 2 s, delivered three times at 100 s intervals) in Box A 3 min after the mice were placed on the grid floor, designated as Context A. After exposure to Context A for fear conditioning in Box A, all mice were returned to their home cage until the next context.
Methods (page 14, line 477 – 478 in untracked-changed version):
In all test in the present study, the chambers were thoroughly cleaned with a 70% ethanol solution between mice every times.
Reviewer 2 comment 7: “Results: In my opinion, this section is well organized, but it illustrates findings in an excessively broad way, without really providing full statistical details, to ensure in-depth understanding and replicability of the findings. Also, in my opinion, it is necessary for the authors to present their findings using summary tables.”
We acknowledge that the statistics were not sufficient for describing all results in our first manuscript. As Reviewer 2 suggested, we put two summary Table 1 and 2 for detail information about the results of Figure 2 and 4 including statistics in our revised version of our manuscript. Please see the two tables in the template manuscript.
Reviewer 2 comment 8: “Discussion: In this final section, the authors described the results and their argumentation and captured the state of the art well; however, I would have liked to see some views on a way forward. I believe that the authors should make an effort, trying to explain the theoretical implication as well as the translational application of this research article, to adequately convey what they believe is the take-home message of their study. Discussion of theoretical and methodological avenues in need of refinement is necessary, as well as suggestions of a path forward in understanding the role of hypothalamic CRF on the induction of false context fear memory and fear generalization. In this regard, recent evidence suggests that the application of new methods in Neuroinflammatory disorders’ treatment, such as the Non-invasive brain stimulation techniques (NIBS), have shown promising results in humans (https://doi.org/10.1038/s41398-020-0851-5). Importantly, I recommend referring to recent studies that revealed that the application of NIBS induces long-lasting effects, noninvasively modulating the cortical excitability, and modulates a variety of cognitive functions altered in patients that suffer from PTSD: for example, a recent review (https://doi.org/10.1016/j.neubiorev.2021.04.036) on the potential and effectiveness of non-invasive brain stimulation (NIBS) to interfere and modulate the abnormal activity of neural circuits (i.e., amygdala-mPFC-hippocampus) involved in the acquisition and consolidation of fear memories, which are altered in many mood psychiatric disorders (i.e., anxiety disorder, specific phobias, post-traumatic stress disorder or depression), would be of interest. Accordingly, another recent manuscript (https://doi.org/10.1016/j.jad.2021.02.076) focused on the same topic, illustrating the therapeutic potential of NIBS as a valid alternative in the treatment of untypically persistent memories that characterized those patients that do not respond to psychotherapy and/or drug treatments. In addition to the previously mentioned literature, authors might also see these additional studies that have focused on the efficacy of NIBS and IBS (https://doi.org/10.3389/fpsyt.2018.00201; https://doi.org/10.3389/fnagi.2020.578339).”
We appreciate the Reviewer 2 suggestion to discuss the important background and possibility raised from our results in the present study. As related to the Reviewer 2 comment 2 and 3, we discussed about the amygdala-mPFC-hippocampus in Discussion section as follow:
Discussion (page 12, line 366 – page 13, line 387 in untracked-changed version):
Functional alterations of the neural network underlying fear conditioning might contribute to the etiology of fear-related psychiatric disease, including PTSD [18]. Fear-associative memory acquisition of fear learning requires coordinated neural activity within the amygdala, prefrontal cortex (PFC), and hippocampus [17-21,48,49]. In addition, cortisol exerts a critical impact on the amygdala–hippocampus–ventromedial PFC network, which underpins fear and memory extinction [17]. Dysregulation of negative feedback, through cortisol suppressing the release of CRF after the long-term activation of the HPA-axis, affects the hippocampus, amygdala, and medial PFC [11]. Inactivation of prefrontal inputs into the nucleus reuniens or direct silencing of nucleus reuniens projections enhances fear memory generalization [27]. Xu and Südhof have demonstrated the generalization of memory attributes for a particular context by processing information from the medial PFC en route to the hippocampus within a day after conditional training [27]. Decreased connectivity be-tween the amygdala and medial PFC has been shown to be related to memory intrusion and the re-experiencing of traumatic events [50]. Interestingly, repetitive transcranial magnetic stimulation of the right dorsolateral PFC appears to have a positive effect in reducing core symptoms in patients with PTSD [50]. Therefore, it is of crucially importance to investigate the role of CRF and cortisol in various neuronal net-works, including the amygdala–hippocampus–ventromedial PFC, with respect to their contribution to both the acquisition of fear memories and the consolidation of imprecise fear memories, although there are still limitations to the elucidation of neuronal mechanisms when using rodent models, as humans and animals differ in terms of the functional neuroarchitecture of the PFC.
References:
- Bremner, J.D. Traumatic stress: effects on the brain. Dialogues in clinical neuroscience 2006, 8, 445-461, doi:10.31887/DCNS.2006.8.4/jbremner.
- Merz, C.J.; Hamacher-Dang, T.C.; Stark, R.; Wolf, O.T.; Hermann, A. Neural Underpinnings of Cortisol Effects on Fear Extinction. Neuropsychopharmacology : official publication of the American College of Neuropsychopharmacology 2018, 43, 384-392, doi:10.1038/npp.2017.227.
- Battaglia, S. Neurobiological advances of learned fear in humans. Advances in clinical and experimental medicine : official organ Wroclaw Medical University 2022, 31, 217-221, doi:10.17219/acem/146756.
- Battaglia, S.; Garofalo, S.; di Pellegrino, G.; Starita, F. Revaluing the Role of vmPFC in the Acquisition of Pavlovian Threat Conditioning in Humans. The Journal of neuroscience : the official journal of the Society for Neuroscience 2020, 40, 8491-8500, doi:10.1523/jneurosci.0304-20.2020.
- Fullana, M.A.; Harrison, B.J.; Soriano-Mas, C.; Vervliet, B.; Cardoner, N.; Àvila-Parcet, A.; Radua, J. Neural signatures of human fear conditioning: an updated and extended meta-analysis of fMRI studies. Molecular psychiatry 2016, 21, 500-508, doi:10.1038/mp.2015.88.
- Battaglia, S.; Harrison, B.J.; Fullana, M.A. Does the human ventromedial prefrontal cortex support fear learning, fear extinction or both? A commentary on subregional contributions. Molecular psychiatry 2022, 27, 784-786, doi:10.1038/s41380-021-01326-4.
- Xu, W.; Südhof, T.C. A neural circuit for memory specificity and generalization. Science (New York, N.Y.) 2013, 339, 1290-1295, doi:10.1126/science.1229534.
- Borgomaneri, S.; Battaglia, S.; Sciamanna, G.; Tortora, F.; Laricchiuta, D. Memories are not written in stone: Re-writing fear memories by means of non-invasive brain stimulation and optogenetic manipulations. Neuroscience and biobehavioral reviews 2021, 127, 334-352, doi:10.1016/j.neubiorev.2021.04.036.
- Stark, R.; Wolf, O.T.; Tabbert, K.; Kagerer, S.; Zimmermann, M.; Kirsch, P.; Schienle, A.; Vaitl, D. Influence of the stress hormone cortisol on fear conditioning in humans: evidence for sex differences in the response of the prefrontal cortex. NeuroImage 2006, 32, 1290-1298, doi:10.1016/j.neuroimage.2006.05.046.
- Kan, R.L.D.; Zhang, B.B.B.; Zhang, J.J.Q.; Kranz, G.S. Non-invasive brain stimulation for posttraumatic stress disorder: a systematic review and meta-analysis. Translational psychiatry 2020, 10, 168, doi:10.1038/s41398-020-0851-5.
Reviewer 2 comment 9: “Even though it is not mandatory, I believe that the ‘Conclusions’ section would be useful to adequately indicate convey what the authors believe is the take-home message of their study, and therefore provide a synthesis of the data presented in the paper as well as possible keys to advancing research and understanding of the prevalence of depression in post-stroke patients.”
We acknowledge that the Conclusion section should be provided for the reader of our study. Therefore, we put the “Conclusion” section after the Discussion as follow:
Conclusion (page 13, line 388 – 405 in untracked-changed version):
In the present study, we found that exposure to a novel but similar contexts to the environment of traumatic fear experience within a few hours induced the formation of false fear memories, which were contextually enhanced within a day after the traumatic fear event. These results suggest that exposure to a novel context within a few hours after a traumatic fear event may interfere with the consolidation of precise fear memory regarding a traumatic event, thus enhancing false context fear memory formation within 24 h. Knockdown of hypothalamic Crf increased the freezing level and potentiated the false context fear memory at 24 h after fear conditioning. On the other hand, over-expression of hypothalamic CRF enhanced the onset of false fear memory formation within a few hours after the traumatic fear event. Cortisol has been suggested to exert a critical impact on the amygdala–hippocampus–ventromedial PFC network, thus underpinning the acquisition of fear memories, as well as the consolidation of precise fear memories and fear generalization. Therefore, it is important to understand how the HPA axis contributes to the formation and potentiation of false con-text fear memories, as well as to explore the associated clinical implications for the treatment of psychiatric diseases, such as PTSD and advance knowledge related to CRF-related cognitive impairment such as post-stroke depression [51], in the future.
Reference:
- Barra de la Tremblaye, P.; Plamondon, H. Alterations in the corticotropin-releasing hormone (CRH) neurocircuitry: Insights into post stroke functional impairments. Frontiers in neuroendocrinology 2016, 42, 53-75, doi:10.1016/j.yfrne.2016.07.001.
Reviewer 2 comment 10: “In according to the previous comment, I would ask the authors to better define a proper ‘Limitations and future directions’ section before the end of the manuscript, in which authors can describe in detail and report all the technical issues brought to the surface.”
We acknowledge that the limitation of our study is important because all of our results were obtained from the rodents. As Reviewer 2 suggested, how our results develop and reflect to the human disease in the future is important for the readers. Although we did not separate the as “Limitation” from the Discussion section, we added the sentences in the Discussion section as follow:
Discussion (page 12, line 381 – page 13, line 387 in untracked-changed version):
Therefore, it is of crucially importance to investigate the role of CRF and cortisol in various neuronal networks, including the amygdala–hippocampus–ventromedial PFC, with respect to their contribution to both the acquisition of fear memories and the consolidation of imprecise fear memories, although there are still limitations to the elucidation of neuronal mechanisms when using rodent models, as humans and animals differ in terms of the functional neuroarchitecture of the PFC.
Reviewer 2 comment 11: “References: Authors should consider revising the bibliography, as there are several incorrect citations. Indeed, according to the Journal’s guidelines, they should provide the abbreviated journal name in italics, the year of publication in bold, the volume number in italics for all the references.”
We thank the Reviewer 2 suggestions. We corrected the style of References. Please see the Reference section in the template manuscript.
Responses to Reviewer 1 (2nd round of revision):
Reviewer 1 comment 1: “- An important problem is still with the English. I still have a hard time to understand many of the modifications they added or some of the replies to our comments.”
We apologize that our poor English in 1st round of revised version of our manuscript also makes difficult for understanding our works. We acknowledge that the English in our 1st round of revised version of manuscript should be further improved and then we asked the MDPI English Editing Service to correct and improve our 2nd round of revised version of our manuscript (Certification no. 44660).
We also apologize that we blanked the page and line number in our responses for your reference in the 1st round of “Response to Reviewers”. We did not put the page and line number in ijms template of tracking version. After the submission of 1st our revised version, we learned how we submit the tracking version in ijms template and how we put the page and line number when the template manuscript was used. We also recorrected the 1st round of our “Response to Reviewers” after this 2nd round of “Response to Reviewer”. Please see our 1st round of “Response to Reviewer”.
Reviewer 1 comment 2: “- It's surprising to hear that no blue filters are available for their confocal to visualize DAPI???”
We thank the reviewer’s comment about our confocal images. Our confocal system, Zeiss LSM510, does not equip the UV laser. Thus, we unfortunately could not insert the nuclei images on the Figure 3 by visualizing with DAPI. Also, we mentioned in the previous “Response to Reviewer”, we tried to purchase the Topro-3 for detecting the nuclear in the cells by 633 nm excitation, however, there is no stock in all agencies in Japan where we are always asking and if we ordered the Topro-3 after the 1st round of reviewer’s suggestion, we had to wait at least 3 to 4 more weeks. I heard that the number of frights which bound for Japan are still restricted and decreased after the pandemic and currently the war between Russian and Ukraine effect the import reagents, it is very hard to get any reagent from the foreign countries.
However, instead, we carefully selected the hypothalamus area and indicated the typical part such as third ventricle (3V) and paraventricular nucleus of hypothalamus (PVN) in all images, and the dotted lines were shown for PVN. The images from i-l in Figure 3B for overexpression of CRF in the hypothalamus (AAV-CRF-T2A-RFP) were changed by other images because the 3V in the previous images were not clearly shown. Therefore, we appreciate the Reviewer 1 that we realized the importance of indication such as 3V and PVN in the images in Figure 3, although we could not show the images with DAPI.
Reviewer 1 comment 3: “- If the authors now include the concept of conjunctive encoding and neuronal excitability, as stated in the introduction, they need to link it to stress or CRF at some point. This part should be removed.”
We agree the suggestion by Reviewer 1. We should delete the sentences from last paragraph of Introduction section, because we did not any shown the link between the stress and CRF in the present study. Therefore, we corrected the sentences in the last paragraph in the Introduction as follow:
Introduction (Page 2, line 75 – 83 in untracked-changed version):
For this study, we endeavored to understand the mechanism of formation of false fear memories after traumatic stress events. In the present study, we focus on the cause of spatial memory failure due to source confusion in inducing false fear memory, using two different contexts within 3 hours after conditional stimulation (CS). Then, we investigated the role of hypothalamic CRF on the induction of false context fear memory using mice with hypothalamic Crf knockdown (Hy-Crf-KD) and hypothalamic CRF overexpression (Hy-CRF-OE). Adeno-associated virus (AAV)-mediated mouse Crf small hairpin RNA (shRNA) was used to knock down Crf expression, while an AAV carrying mouse Crf cDNA was constructed for the overexpression of CRF.
Reviewer 1 comment 4: “- Statements about correlation/causation related to time in the centre zone of a novel environment as a coping behaviour that affects freezing 24h later should still be weakened, this is highly speculative and only based on a correlation.”
We acknowledge that it was highly speculative and weakened to suggest that the preference of central zone during Box B at 3 h is a key point for potentiation of the freezing level in Box B at 24 h and the KD or OE of Hypothalamic CRF disrupt this correlation by enhancing and potentiating the false fear memory. We would carefully investigate and analyze this correlation in our future study. Therefore, we deleted the following paragraph from the Results and Discussion, and the statistic in Methods sections as follow:
Result (page 9, line 289 – 310 in tracked-changed version):
Discussion (page 16, line 547 – 557 in tracked-changed version):
Methods (page 21, line 773 – 775 in tracked-changed version):
Methods (page 21, line 778 in tracked-changed version):
.
Responses to Reviewer 1 (1st round of revision):
Reviewer 1 comment 1: “~~ However, we have some major problems with both the interpretation of the results and the fitting with previous conceptions of, for instance, what fear generalization actually is. This, together with a rather poor English grammar, makes the paper difficult to read and be put in context with previous work.”
We apologize that our poor English language makes difficult for understanding our works in the first submitted manuscript. We acknowledge that the English in our manuscript should be improved and we asked our revised version of manuscript to the Wordvice English Editing Service and MDPI English Editing Service.
Reviewer 1 comment 2: “One major concern is the concept of “generalization”. Although the authors dedicate a big part of the introduction to present these terms, the difference between “false memory” and “generalization” (and in the discussion, “memory confusion”) is not clear. Is it a matter of time? Amount of detected freezing? Are there different neurobiological mechanisms? Basically, the authors make interpretations in the discussion (page10 of 17, line 312) that previously they have assumed in their entire manuscript, and only when reading the discussion one understands what they refer to with generalization and false memory. They acknowledge the definition of generalization provided by others (Wiltgen and Silva), but disregarding that this definition is applied to memories more remote than 24h. This is quite confusing, and makes the first reading of the paper complicated for someone used to the general definition of generalization.”
We acknowledge that our interpretation against the word “Fear generalization” of first submitted manuscript was not adequate for our results according to general definition of published many papers. Particularly, the paper by Wiltgen an Silva, and Cai et al 2016 were very helpful to understand the definition of “generalization”. We strongly agreed the Reviewer 1 suggestion and the interpretation of the word “generalization” in our results of first submitted manuscript was changed to the “False fear memory” as follow:
Title (page 1, line 2 – 3 in untracked-changed version):
Formation of false context fear memory is regulated by hypo-thalamic corticotropin-releasing factor in mice
Abstract (page 1, line 11 in untracked-changed version):
Traumatic events frequently produce false fear memories.
Abstract (page 1, line 18 – 19 in untracked-changed version):
The within-group freezing level at B-24h was higher than that at B-3h, indicating that false context fear memory was enhanced at B-24h.
Abstract (page 1, line 23 – 24 in untracked-changed version):
Therefore, we hypothesized that Hy-Crf-KD potentiates the induction of false context fear memory, while Hy-CRF-OE enhances the onset of false fear memory formation.
Keywords (page 1, line 25 – 26 in untracked-changed version):
Traumatic stress, false context fear memory; hypothalamic corticotropin-releasing factor, Adeno-associated virus, shRNA
Introduction (page 1, line 37 – page 2. line 51 in untracked-changed version):
Indeed, this increase in generalization is due to a loss of detailed information about the context, and not fear incubation [7]. However, in contrast to false context fear memory, inducing of “fear generalization” across environments requires at least a week after exposure to a traumatic event in rodents, and it has been demonstrated that one month is required to induce freezing in novel environments [8]. According to the hypothesis that a shared neural ensemble linking distinct memories that encode close in time due to a temporary increase in neuronal excitability, a subsequent memory to the neuronal ensemble encodes the first memory. It has been hypothesized that a shared neural ensemble links distinct memories which are encoded within a close timeframe, due to a temporary increase in neuronal excitability. Accordingly, in rodents, it has been demonstrated that the first memory can strengthen a second memory within a day, but not across a week [9]. Although, a false context fear memory of the traumatic event may form and subsequently lead to fear generalization, the neuronal mechanism of fear generalization may differ from the concept of first memory neuronal ensemble encoding, which may lead to the false fear memories.
References:
[7] Anagnostaras, S.G.; Gale, G.D.; Fanselow, M.S. Hippocampus and contextual fear conditioning: recent controversies and advances. Hippocampus 2001, 11, 8-17, doi:10.1002/1098-1063(2001)11:1<8::aid-hipo1015>3.0.co;2-7.
[8] Wiltgen, B.J.; Silva, A.J. Memory for context becomes less specific with time. Learning & memory (Cold Spring Harbor, N.Y.) 2007, 14, 313-317, doi:10.1101/lm.430907.
[9] Cai, D.J.; Aharoni, D.; Shuman, T.; Shobe, J.; Biane, J.; Song, W.; Wei, B.; Veshkini, M.; La-Vu, M.; Lou, J.; et al. A shared neural ensemble links distinct contextual memories encoded close in time. Nature 2016, 534, 115-118, doi:10.1038/nature17955.
Introduction (page 2, line 75 – 83 in untracked-changed version):
For this study, we endeavored to understand the mechanism of formation of false fear memories after traumatic stress events. In the present study, we focus on the cause of spatial memory failure due to source confusion in inducing false fear memory, using two different contexts within 3 hours after conditional stimulation (CS). Then, we investigated the role of hypothalamic CRF on the induction of false context fear memory using mice with hypothalamic Crf knockdown (Hy-Crf-KD) and hypothalamic CRF overexpression (Hy-CRF-OE). Adeno-associated virus (AAV)-mediated mouse Crf small hairpin RNA (shRNA) was used to knock down Crf expression, while an AAV carrying mouse Crf cDNA was constructed for the overexpression of CRF.
Results (page 4, line 131 – 134 in untracked-changed version):
These results suggest that the false fear memory of Context A was induced within 3 h after the CS, and the false fear memory was potentiated in Box B at 24 h if the mice had experienced a similar environment to Context A with the CS after formation of the false fear memory.
Results (page 5, line 160 – 161 in tracked-changed version):
- 2. Hy-Crf-KD enhanced false fear memory level in 24 h and Hy-CRF-OE potentiate the false fear memory level in 3 h after fear conditioning.
Results (page 5, line 162 in tracked-changed version):
To investigate the effects of HPA axis activity on false fear memory,
Results (page 7, line 201 – 206 in tracked-changed version):
On the other hand, a paired t-test also indicated that the freezing levels of control and Hy-Crf-KD mice, but not Hy-CRF-OE mice, at 24 h after the CS were higher than those at 3 h (control, 24 h: t = 3.022, df = 12, p < 0.05 vs. 3 h; Hy-Crf-KD mice, 24 h: t = 4.467, df = 10, p < 0.01 vs. 3 h; Fig. 4D, Table 2), indicating that the false fear memory was po-tentiated in Box B at 24 h in control and Hy-Crf-KD mouse groups when the mice were experienced Box B 3 h after the CS.
Results (page 13, line 370 – 373 in tracked-changed version):
Therefore, we discussed the relationship between false context fear memory formation and hypothalamic CRF.
Discussion (page 10, line 277 – page 11, line 293 in untracked-changed version):
Fear generalization to innocuous stimuli acting as reminders of the trauma, even in safe place or environment, is one of the central problems and a hallmark of PTSD [29]. It has been further posited that the generalization will be enhanced if the new context is similar, but not completely different, from the training context, due to memory source confusion [30]. False memories can be spontaneously produced after an extreme fear experience but, classically, the term “false memory” applies to memory formed without actual experience of the event. If “fear generalization” is defined as the actual memory of a fear experience that transfer a conditioned response to stimuli that perceptually differ from the original conditioned stimulus [30], fear generalization may also be the same as false fear memory. In rodents, fear generalization has been shown to be induced when mice were exposed to contexts soon after training [26]. From these reports, the definition of “memory generalization” is never only with respect to a re-mote memory. In the present study, we applied a environment 3 h and 24 h after the CS, which the mice may not have been able to precisely discriminate from the box in which the conditioned was carried out. As such, this paradigm is not a test of the re-mote memories, but, instead, experiencing the novel box 3 h after the CS led to the source confusion of the traumatic event and potentiated the induction of false context fear memory.
References:
[26] Fujinaka, A.; Li, R.; Hayashi, M.; Kumar, D.; Changarathil, G.; Naito, K.; Miki, K.; Nishiyama, T.; Lazarus, M.; Sakurai, T.; et al. Effect of context exposure after fear learning on memory generalization in mice. Mol Brain 2016, 9, 2, doi:10.1186/s13041-015-0184-0.
[29] Jovanovic, T.; Norrholm, S.D.; Blanding, N.Q.; Davis, M.; Duncan, E.; Bradley, B.; Ressler, K.J. Impaired fear inhibition is a biomarker of PTSD but not depression. Depress Anxiety 2010, 27, 244-251, doi:10.1002/da.20663.
[30] Bergstrom, H.C. Assaying Fear Memory Discrimination and Generalization: Methods and Concepts. Curr Protoc Neurosci 2020, 91, e89, doi:10.1002/cpns.89.
Discussion (page 11, line 316 – 328 in untracked-changed version):
It has been found that, when separated by a week, independent populations of neurons encoded two distinct contexts; meanwhile, while the two contexts were separated only within a day, shared neuronal ensembles between the two contexts overlapped in the CA1 region of the hippocampus [9]. Neuronal excitability can lead to increases in memory strength, and neural ensemble sharing can strengthen the memory for a secondary context within 5 h [9]. Therefore, the neuronal mechanism of generalized re-mote fear memories in patients with PTSD is different from false memories produced within a day. In the present study, freezing behavior was observed in Context B 3 h after the conditioning, and the freezing response at 24 h was facilitated. Therefore, our observation that experiencing a novel environment enhanced false context fear memory 24 h after a traumatic fear event may be explained by the hypothesis of shared neuronal ensembles between Box A and Box B strengthening the false memory within 24 h.
Reference:
[9] Cai, D.J.; Aharoni, D.; Shuman, T.; Shobe, J.; Biane, J.; Song, W.; Wei, B.; Veshkini, M.; La-Vu, M.; Lou, J.; et al. A shared neural ensemble links distinct contextual memories encoded close in time. Nature 2016, 534, 115-118, doi:10.1038/nature17955.
Conclusion (page 13, line 389 – 394 in untracked-changed version):
In the present study, we found that exposure to a novel but similar contexts to the environment of traumatic fear experience within a few hours induced the formation of false fear memories, which were contextually enhanced within a day after the traumatic fear event. These results suggest that exposure to a novel context within a few hours after a traumatic fear event may interfere with the consolidation of precise fear memory regarding a traumatic event, thus enhancing false context fear memory formation within 24 h.
Reviewer 1 comment 3: “Importantly, the authors do not cite key work related to their experiments and addressing the neurobiological mechanisms underlying memory linking (probably the equivalent to their “false memory”) such as Cai et al., 2016. In fact, most of what is discussed in the manuscript can also be interpreted under the light of the two memories (context A and B) being linked, rather than “confused” or “difficult to discriminate” as the authors suggest.
Having said this, my biggest suggestion is that the authors come with a different name for what they refer to with “generalization”, that then can be for the first time linked to false memory formation and CRF levels. This could improve the paper.”
As we mentioned above in our response to Reviewer 1 comment 2, according to the general definition of published many papers, we acknowledge that our interpretation for the term of “Fear generalization” in first submitted manuscript was not adequate for our results. We carefully read the paper by Cai et al 2016 many times and we agreed that the concept of the overlapping between the neuronal ensembles representing two separate contextual memories within a day and over a week. Therefore, we extensively revised our suggestion throughout the manuscript. Please see our response to Reviewer 1 comment 2 as above.
Reviewer 1 comment 4: “In the results section, sub-titles should reflect statements related to the most important finding (“manipulation A results in B”) rather than a generic sentence.”
We acknowledge that the sub-titles did not include our important finding. We have changed two sub-titles in the Results section as follow:
Results (page 2, line 85 – 86 in untracked-changed version):
- 1. Exposure of novel context in 3 h after fear conditioning formed false fear memory that was further enhanced at 24 hours after conditioning.
Results (page 5, line 160 – 161 in untracked-changed version):
- 2. Hy-Crf-KD enhanced false fear memory level in 24 h and Hy-CRF-OE potentiate the false fear memory level in 3 h after fear conditioning.
Reviewer 1 comment 5: “Fig 1. The two contexts have the same shape. How do the authors think their results would look if the two contexts were more dissimilar, e.g., with a different shape?”
We acknowledge that we should discussed about the possibility when the mice were exposed to a dissimilar chamber as a novel box, whether the mice represent the false fear memory or not. Therefore, we cite the adequate reference that the mice expose to the dissimilar box after the conditioning, mice did not exhibit the freezing behaviors in 3 hours after the conditioning in Discussion section as follow:
Discussion (page 10, line 260 – 263 untracked-changed version):
It has been demonstrated that the mice were habituated to exposed to a novel and highly dissimilar context with conditioning chamber exhibited a low level of freezing in dissimilar chamber on day 0 [26].
Reference:
[26] Fujinaka, A.; Li, R.; Hayashi, M.; Kumar, D.; Changarathil, G.; Naito, K.; Miki, K.; Nishiyama, T.; Lazarus, M.; Sakurai, T.; et al. Effect of context exposure after fear learning on memory generalization in mice. Mol Brain 2016, 9, 2, doi:10.1186/s13041-015-0184-0.
Reviewer 1 comment 6: “Fig 2. In our opinion, the false memory vs. generalization dichotomy brings problems in fig 2: Here, the authors say that “false fear memory (in context B) was induced 24h after the CS” and, below, “formed within 3h after CS” (page4, line112), after comparing freezing in A-B 24h vs. A-B 3h. We understand from this is to suggest that elevated levels of freezing both at 3h and 24h after conditioning are interpreted by the authors as similar false memory. However, 3h is similar to the time required for memory linking (Cai et al., 2016), whereas 24h could be, although still too short, more widely associated with the idea of “generalization”. In the concluding sentence, they state that, on the one hand, false memory was induced within 3h and, on the other hand, that generalization was produced if the mice had experienced the intermediate exposure to context B. Looking at the data, it is clear that “generalization” is produced regardless of the intermediate exposure (as evidenced by increased freezing in A-B 24h). What remains interesting is the fact that prompt formation of false memory (3h) is necessary for having increased freezing levels at 24h compared to those without 3h exposure to context B. This is probably the most interesting observation, that in our opinion becomes diluted in the false memory vs. generalization terminology/conceptual discussion. Calling this something different, such as “false memory enhancement” would eliminate this problem.
As related the suggestion by Reviewer 1 in the comment 2 and 3, we have performed extensive revised throughout the manuscript and the Title, Abstract, Introduction, Results, Discussion and Conclusion were corrected as above. Please see our response to Reviewer 1 comment 2.
Reviewer 1 comment 7: “In comparing freezing after 24h in context A in A-A vs. A-B-A mice (fig. suppl. 1C vs. D), it seems like exposure to context B 3h after conditioning weakened memory expression in the context A. First, the experiment ABA comes a little bit out of the blue and should be more properly introduced. Second, this observation might be linked to the same concept as before, in the sense that intermediate exposure to different contexts affect memory expression 24h later, which is very interesting. How do the authors interpret this?”
We acknowledge that the results of freezing level 24 h in ABA mice looked decreased, compared to that in A-A mice. However, we did not these two experiments were not performed at the same time (not same batch). Therefore, we cannot represent these results as one graph and perform the statistical analysis. We discussed the possibility that the shorter freezing level 24 h in ABA mice than that in A-A mice in Discussion section as follow:
Discussion (page 11, line 293 – 309 in untracked-changed version):
In the present study, exposing the mice to the novel box (i.e., Context B in Box B) 3 h after the CS (i.e., Context A in Box A), caused the freezing level of ABA mice in Box A at 24 h after the CS to decrease, compared with that of mice those were not exposed to Box B at 3 h (i.e., A-A mice). However, we did not perform these experiments at the same time (i.e., not in the same batch) and, so, we cannot statistically compare the difference in freezing level between ABA and A-A mice. If the freezing level of ABA mice in Box A at 24 h was decreased, compared to that of the A-A mice, two hypotheses can be posed. One is memory extinction, and the other is that the experience in the novel Box B at 3 h interfered with the memory consolidation regarding Box A, which delivered the electrical shocks. The widely used paradigm of memory extinction is generally that, after the CS, the mice are exposed to the same chamber several times without the CS; however, we used a novel box after the CS. On the other hand, the memory of Box A, with the CS, might have become less precise by interfering with the perceptual memory though exposure to Box B at 3 h. This result can be also considered in terms of the hypothesis that a shared neural ensemble may link distinct memories that are encoded within a close timeframe due to a temporary increase in neuronal excitability.
Reviewer 1 comment 8: “Fig 3. Subfigures A-D can be moved to supplementary materials. The entire maps of the viral vectors are not necessary, a schematic representation is enough. In B, do the authors have in vitro evidence that their overexpression constructs worked? In D, what does “Nat Neurosci” and the different code numbers refer to? In F, the distance indicated by the scale bars should be indicated in the figure legend. Furthermore, a DAPI stain would be an added plus, as well as a zoomed out image of the hypothalamus. Finally, the figure legend contains a panel “M?”
We acknowledge that“Nat Neurosci” and the different code numbers in the figure are not explained in legend of Figure 3. We changed the indication of each band of western blotting and explained in figure legend refer to the“Nat Neurosci” and the code numbers. In addition, we wrote the scale bar in the figure legend and delete “M”. As for counterstaining with DAPI for CRF and fluorescence tag (GFP or RFP), we do not have a laser for detecting the DAPI, so we tried to purchase the TO-PRO-3 which can be detected by Ex 642. Unfortunately, we have to import from the Sigma-Aldrich or Thermo Fisher Scientific in the USA, so we have to wait over three weeks to get this. Therefore, we cannot try to stain the nuclear in the hypothalamic sections by deadline during revision.
Subfigures A-D in Figure 3 have moved to the Supplementary Figure S2 and changed the Figure 3. Also, we improved the Methods section and the legend of Figure 3 and separate it to the supplementary Figure S2 as follow:
Methods (page 15, line 495 – 503 in untracked-changed version):
The shRNA sequences used to knockdown Crf were based on a previous report [56], the Sigma MISSION shRNA library by the RNAi Consortium (Boston, MA, USA): (#1) AGATTATCGGGAAATGAAA [56], TRCN0000414479 (#2) TTAGCTCAGCAA-GCTCACAG, TRCN0000436997: and (#3): ATCTCTCTGGATCTCACCTTC. We assessed the knockdown efficiencies of shRNA candidates by co-transfection of each shRNA-carrying AAV plasmid and FLAG-CRF-encoding pCAGGS plasmid into HEK293FT cells. CRF expression was analyzed by immunoblot with anti-FLAG-HRP antibody (#015-22391, FUJIFILM Wako Pure Chemical Corp., Tokyo, Japan) (Fig. S2C, S2D and Fig. S3).
Reference:
[56] Elliott, E.; Ezra-NevoL, G.; Regev, L.; Neufeld-Cohen, A.; Chen, A. Resilience to social stress coincides with functional DNA methylation of the Crf gene in adult mice. Nat Neurosci 2010, 13, 1351-1353, doi:10.1038/nn.2642.
Figure 3 legend (page 6, line 167 – 177 in untracked-changed version):
Figure 3. (A) AAV-PHP.eB virus injection maps according to the Paxinos mouse brain atlas (anterior: -0.7 mm, lateral: 0.25 mm, Depth: 4.40 mm). Green bars around the center of the brain map indicate virus injection needles and the location of virus injections into the hypothalamus including the PVN (0.1 μL of virus injected by 32-gauge Neurosyringe). In the images, "3V” represents the third ventricle and PVN represents the paraventricular nucleus of hypothalamus in the red square, in the hypothalamus. (B) Transduction of mouse hypothalamus with AAV-PHP.eB vector expressing GFP or RFP and detection of mouse CRF by immunofluorescence in control (a-d; upper lane), knockdown (e-h; middle lane; Hy-Crf-KD), and overexpression (i-l; lower lane, Hy-CRF-OE) mice. In the images, “3V” represents the third ventricle and “PVN” represents the paraventricular nucleus of hypothalamus. Scale bar = 200 μm (×10 magnification) for a, e, I and 50 μm for b, c, d, f, g, h, j, k, l (×40 magnification).
Supplementary Fig. S2 (Page 3, line 56 in untracked-changed version):
Construction and characterization of adeno-associated virus (AAV) PHP.eB vectors for knockdown of mouse Crf by shRNA system and overexpression of mouse CRF. (A) Illustration of the AAV vectors expressing Crf shRNAs under the control of U6 promoter. The vectors express GFP as a reporter gene under the control of Chicken β-Actin promoter (CBA pro). (B) Illustration of the AAV vector carrying CRF*-FLAG-T2A-RFP. CRF* represents the shRNA#2 resistant version. (C) Western blotting for expression of CRF*-FLAG-T2A-RFP carrying AAV plasmid that was transfected to HEK293 cells; (D) The knockdown validation of the effective shRNA candidates. #2 shRNA was the most effective, and so was used in further experiments.
Reviewer 1 comment 9: “Fig 4. 4C is referenced in the text before 4B.”
We acknowledge that the describing the result of 4B should be before the 4C. Therefore, we have changed the sentences as follow:
Results (Page 7, line 186 – 193 in untracked-changed version):
The freezing level was similar in the three groups of ABB-control, ABB-KD, and ABB-OE mice during the three electric shocks in Box A as a conditioning stimulus (Fig. 4B). One-way ANOVA revealed that the AAV injections affected the locomotor activity in Box A for 3 min just before the CS was delivered (F(2, 35) = 4.787, p < 0.05, Fig. 4C). Fisher's LSD test indicated that the locomotor activity in Hy-Crf-KD mice was higher than that in control mice and Hy-CRF-OE mice (control: p < 0.01 vs. Hy-Crf-KD; Hy-CRF-OE: p < 0.05 vs. Hy-Crf-KD, p = 0.277 vs. control, Fig. 4C).
Reviewer 1 comment 10: “Fig 4. Variability of freezing is high among injected animals. Was an outlier test carried out? Were injection sites checked and animals with poor injection discarded from the test? Were all the animals conditioned and tested at the same time? Otherwise, inter-group comparisons of total freezing levels might be subject to batch effects that are difficult to account for. In any case, the difference in freezing levels between 3h and 24h is interesting and adequate in trying to normalize freezing per group even if this was obtained from different batches.”
As Reviewer 2 suggested to summarize the statistics, we represented all averages and SEMs in the Table 1 and 2 refer to the Figure 2 and 4. The range of SEM in 3 h after the conditioning in non-AAV injected mice were from 1.373 to 11.01%, while the range of SEM in 3 h after the conditioning in AAV-injected mice were from 0.394 to 5.058. Therefore, we do not think the variability of freezing is not higher among AAV injected mice than non-injected mice. Please see the Table 1: page 5, line 156 and Table 2: page 9, line 246 – page 10, line 248 in untracked-changed version.
Reviewer 1 comment 11: “Page 6, line 189. This was the first time, together with the conclusion of the first part of the results, we understood that the authors use “fear generalization” for the increase in freezing observed at 24h after the pre-exposure to context at 3h post-conditioning… This is problematic because this is only clearly stated in the discussion and reflects and interpretation of the authors, it is difficult to follow the paper if this is not defined before. Again, we would suggest that the authors come with a new name for the phenomenon they observe, to distinguish it from the more traditional idea of generalization. This might help the reader to not get distracted in terminology issues.”
As related suggestion by Reviewer 1 in the comment 2 and 3, we have performed extensive revised throughout the manuscript and the Title, Abstract, Introduction, Results, Discussion and Conclusion were corrected. Please see our response to Reviewer 1 comment 2 as above.
Reviewer 1 comment 12: “Fig 5. We appreciate the effort in trying to link anxiety leading to “false memory” to freezing the day after, but do not quite understand the logic behind this analysis and what additional information different from freezing this brings… it is obvious that the longer a mouse will spend freezing, the less distance it will move. Correlation of 3h distance vs. 24h freezing is lost in genetically manipulated mice, but correlation is not causation (statements as in page10 of 17, line 358 should be weakened). This only reinforces our opinion that it is extremely difficult to extract any conclusion about anxiety levels from assessments that are no de-coupled from fear conditioning. Assessing distance travelled in context A before CS, as done in fig4C, is the closest and most appropriate way.”
We acknowledge that this part is not adequate and awkward phrase. Also, we should not discuss the relationship from the result between the center preference in the box and the anxiety level. However, the findings the central preference in the Box B 3h and the enhancement of false fear memory in 24 h after the conditioning is very important. We are currently the possibility that the center preference is a kind of active coping against novel environment at 3 h and decrease the enhancement of false fear memory in 24 h. Therefore, we changed the sentences in Results and Discussion sections as follow:
Results (page 9, line 289 – 310 in tracked-changed version):
To confirm the anxiety levelbehavioral patterns in mice with genetic manipulation of CRF expression in the hypothalamus, we measured the percentage of the distance traveled and the time spent in the central zone in Box B 3 and 24 h after the CS during Context A. Two-way (time × CRF) repeated measures ANOVA revealed that only “time”, but not “CRF” affected the preference for the central zone in Box B (% distance traveled in central zone: two-way repeated measures ANOVA: F(4, 72) = 0.4528, p = 0.770; factor (time): F(2, 72) = 17.7202, p < 0.01; factor (CRF): F(2, 36) = 0.91767, p = 0.4089409; % time spent in central zone: two-way repeated measures ANOVA: F(4, 72) = 0.6403, p = 0.6355636; factor (time): F(2, 72) = 18.262, p < 0.01; factor (CRF): F(2, 36) = 0.5075508, p = 0.6062, Fig. 5A and B). However, one-way ANOVA revealed that the preference for the central zone in either Hy-Crf-KD or Hy-CRF-OE mice at each time point was similar to that of control mice (distance traveled: 0 h: F(2, 38) = 0.0132, p = 0.9869987; 3 h: F(2, 38) = 1.7432, p = 0.1894; 24 h: F(2, 38) = 0.5693, p = 0.5720, Fig. 5A; time spent: 0 h: F(2, 38) = 0.0974, p = 0.9074; 3 h: F(2, 38) = 1.1780, p = 0.3195320; 24 h: F(2, 38) = 0.9160, p = 0.4092, Fig. 5B). Conversely, there was no significant correlation between the freezing level in 24 h and the distance traveled by Hy-Crf-KD and Hy-CRF-OE mice in Box B in 3 h (Hy-Crf-KD mice: Pearson’s correlation, t = -0.5064, df = 9, p = 0.3124, R2 = 0.0277, Fig. 5D; Hy-CRF-OE mice: Pearson’s correlation, t = -0.37617, df = 13, p = 0.3564, R2 = 0.011; Fig. 5E). Considering that the hypothalamic CRF did not affect the preference for the central zone during Context B 3 h, the disrup-tion of the correlation between the preference for the central zone and the freezing level during Context B 24 h in Hy-Crf-KD and Hy-CRF-OE mice might be due to their discrimination level toward Box A and Box B in 3 h after the CS.
Discussion (page 16, line 547 – 557 in tracked-changed version):
Finally, in the present study, we found that the preference for the central zone in Box B at 3 h—as a novel environment after the CS—affected the freezing level during Context B 24 h in control mice. While hypothalamic CRF did not affect the preference for the central zone during Context B 3 h, the significant correlation between the preference for the central zone in a novel chamber (Box B 3 h) and the freezing level during Context B 24 h was abolished in Hy-Crf-KD and Hy-CRF-OE mice. Therefore, we also suggest that adequate active coping behavior when in a novel environment may decrease and subsequently induce false context fear memory formation.
Reviewer 1 comment 12: “- Line 30, evidence is not plural”
We thank your suggestion for our grammatical error in the manuscript. We corrected from “influence” to “influences”.
Reviewer 1 comment 13: “- Line 64: The statement “in patients with PTSD” is grossly misleading, as mice are being studied here.”
We acknowledge that the statement “in patients with PTSD” is not adequate in our mouse study. According to the correction of our focus in this study, we focus on the false fear memory, but not fear generalization. Therefore, we excluded this phrase “in patients with PTSD”.
Reviewer 1 comment 14: “- Line 143-147. This is not a proper sentence, grammatically speaking.”
We acknowledge that this part of our English was very poor. Therefore, we corrected this sentence in the Results section as follow:
Results (page 5, line 162 – 165 in untracked-changed version):
To investigate the effects of HPA axis activity on false fear memory, we induced mice with Hy-Crf-KD or Hy-CRF-OE through AAV-PHP.eB-produced virus injection into the hypothalamus, for comparison with mice injected with AAV-PHP.eB GFP (green fluorescence protein) into the hypothalamus as a control (Fig. 3 and Fig. S2).

Round 3
Reviewer 1 Report
The authors have addressed most of our concerns. However, the paper is still in dire need for grammar/English checking. This should be done throughout the manuscript, and not only at certain paragraphs.